# Pre-existing polymerase-specific T cells expand in abortive seronegative SARS-CoV-2

Leo Swadling[1✉], Mariana O. Diniz[1,24], Nathalie M. Schmidt[1,24], Oliver E. Amin[1,24], Aneesh Chandran[1,24], Emily Shaw[1,24], Corinna Pade[2], Joseph M. Gibbons[2], Nina Le Bert[3], Anthony T. Tan[3], Anna Jeffery-Smith[1,2], Cedric C. S. Tan[4], Christine Y. L. Tham[3], Stephanie Kucykowicz[1], Gloryanne Aidoo-Micah[1], Joshua Rosenheim[1], Jessica Davies[1], Marina Johnson[5], Melanie P. Jensen[6,7], George Joy[6,8], Laura E. McCoy[1], Ana M. Valdes[9,10], Benjamin M. Chain[1], David Goldblatt[5], Daniel M. Altmann[11], Rosemary J. Boyton[12,13], Charlotte Manisty[6,8], Thomas A. Treibel[6,8], James C. Moon[6,8], COVIDsortium Investigators*, Lucy van Dorp[4], Francois Balloux[4], Áine McKnight[2], Mahdad Noursadeghi[1,24], Antonio Bertoletti[3,14,24] & Mala K. Maini[1✉]

Individuals with potential exposure to severe acute respiratory syndrome coronavirus 2 (SARS-CoV-2) do not necessarily develop PCR or antibody positivity, suggesting that some individuals may clear subclinical infection before seroconversion. T cells can contribute to the rapid clearance of SARS-CoV-2 and other coronavirus infections[1–3]. Here we hypothesize that pre-existing memory T cell responses, with cross-protective potential against SARS-CoV-2 (refs. [4–11]), would expand in vivo to support rapid viral control, aborting infection. We measured SARS-CoV-2-reactive T cells, including those against the early transcribed replication–transcription complex (RTC)[12,13], in intensively monitored healthcare workers (HCWs) who tested repeatedly negative according to PCR, antibody binding and neutralization assays (seronegative HCWs (SN-HCWs)). SN-HCWs had stronger, more multispecific memory T cells compared with a cohort of unexposed individuals from before the pandemic (prepandemic cohort), and these cells were more frequently directed against the RTC than the structural-protein-dominated responses observed after detectable infection (matched concurrent cohort). SN-HCWs with the strongest RTC-specific T cells had an increase in *IFI27*, a robust early innate signature of SARS-CoV-2 (ref. [14]), suggesting abortive infection. RNA polymerase within RTC was the largest region of high sequence conservation across human seasonal coronaviruses (HCoV) and SARS-CoV-2 clades. RNA polymerase was preferentially targeted (among the regions tested) by T cells from prepandemic cohorts and SN-HCWs. RTC-epitope-specific T cells that cross-recognized HCoV variants were identified in SN-HCWs. Enriched pre-existing RNA-polymerase-specific T cells expanded in vivo to preferentially accumulate in the memory response after putative abortive compared to overt SARS-CoV-2 infection. Our data highlight RTC-specific T cells as targets for vaccines against endemic and emerging *Coronaviridae*.

There is wide variability in the outcome of exposure to SARS-CoV-2, ranging from severe illness to asymptomatic infection, to those individuals who remain negative according to standard diagnostic tests. Recent studies have identified SARS-CoV-2 T cell reactivity in prepandemic samples[5–11,15–18] and isolated cases of exposed individuals who have not seroconverted with single-time-point screening[4,16,19–22]. We studied an intensively monitored cohort of HCWs with potential exposure during the first UK pandemic wave (23 March 2020), comparing those with or without PCR and/or antibody evidence of SARS-CoV-2 infection. We postulated that, in HCWs for whom PCR and the most

sensitive binding and neutralizing antibody tests remained repeatedly negative (SN-HCWs), T cell assays might distinguish a subset of SN-HCWs with a subclinical, rapidly terminated (abortive) infection. We hypothesized that these individuals would exhibit pre-existing memory T cells with cross-reactive potential, obviating the time required for de novo T cell priming and clonal expansion. In SN-HCWs, and in an additionally recruited cohort of medical students and laboratory staff with stored prepandemic samples that remained seronegative after close contact with cases, we had the opportunity to compare SARS-CoV-2-specific memory T cells with those that were already

present in the same individual before, or at the time of, potential exposure.

We included an analysis of the understudied T cells directed against the core RTC within open reading frame 1ab (ORF1ab) (RNA polymerase co-factor non-structural protein 7 (NSP7), RNA polymerase NSP12 and helicase NSP13, hereafter the RTC); these are putative targets for pre-existing responses with pan-*Coronaviridae* reactivity, because they are likely to be highly conserved due to their key early roles in the viral life cycle. Consistent with this, in cases in which immunity against other viruses (including hepatitis B virus (HBV), hepatitis C virus (HCV), HIV and Japaneses encephalitis virus (JEV)) has been described in exposed seronegative individuals, T cells were more likely to target non-structural proteins, such as polymerase, compared with in individuals with a seropositive infection[23–27].

## SARS-CoV-2 T cells in seronegative HCWs

We compared T cell reactivity in intensively monitored HCWs with a laboratory-confirmed infection or SN-HCWs, matched for exposure risk and demographic factors (COVIDsortium; Fig. 1a and Extended Data Table 1). Additional control cohorts included healthy adults who were sampled in London, UK, or Singapore before SARS-CoV-2 circulation in humans (prepandemic cohort; Fig. 1a). SN-HCWs were defined by negative weekly diagnostic tests (baseline–week 16, SARS-CoV-2 PCR, nasopharyngeal swab; anti-spike-1 IgG and anti-nucleoprotein (NP) IgG/IgM seroassays[28]; Fig. 1b–d). Having previously reported a range of neutralizing antibody titres at week 16 in laboratory-confirmed infections, we examined neutralizing antibodies in SN-HCWs. Two HCWs with neutralizing antibody titres that were just above the threshold were excluded from further analyses; the remaining SN-HCWs were negative by pseudotype assay (Fig. 1e), with a subset also confirmed to be negative at three time points for authentic virus neutralization (Extended Data Fig. 1a). SN-HCWs could have become PCR negative by recruitment; however, non-seroconverters after PCR positivity were rare (2.6% of PCR-positive HCWs negative by all three seroassays[16]) and antibody responses are unlikely to have waned before recruitment[28]. Furthermore, SN-HCWs lacked detectable SARS-CoV-2 spike-specific memory B cells, which we have shown persist after waning of neutralizing antibodies[29] (Extended Data Fig. 1b; below the detection threshold). Thus, the SN-HCWs represented a cohort of intensely monitored HCWs who resisted classical laboratory-confirmed infection.

We quantified SARS-CoV-2-specific memory T cells by ELISpot using unbiased stimulation with overlapping peptides covering structural proteins and the less-well-studied non-structural proteins of the RTC (Fig. 1f). As previously described, when using sensitive assays[5–7,9,17,18] (such as 400,000 peripheral blood mononuclear cells (PBMCs) per well IFNγ ELISpot analysis used here[8,16]), some SARS-CoV-2-reactive T cells were detectable in prepandemic samples; however, their multispecificity was significantly lower compared with the week 16 group with a laboratory-confirmed infection (Fig. 1g, h; structural responses at week 16 previously reported[16]). By contrast, SN-HCWs had SARS-CoV-2-specific T cells that were comparable in breadth to infected HCWs at week 16 and significantly more multispecific than prepandemic samples (Fig. 1g, h). The SARS-CoV-2-specific T cells of SN-HCWs targeted more protein pools and had an approximately fivefold higher cumulative magnitude of responses compared with those of the prepandemic cohort, with an overall strength equivalent to the infected cohort at week 16 (Fig. 1i, j).

T cells from prepandemic samples tended to not target both halves of NP (NP1 and NP2 subpools), whereas around 50% of SN-HCWs and HCWs with a laboratory-confirmed infection did, confirming our earlier suggestion[8] that this serves as a simple proxy measure of a multispecific response (Extended Data Fig. 1c–e). Taken together, we found a higher magnitude and breadth of SARS-CoV-2-specific T cells in HCWs who

repeatedly tested PCR and antibody negative compared with individuals in the prepandemic cohort.

## RTC-specific T cells and *IFI27* in SN-HCWs

We next investigated whether T cell memory differs in SN-HCWs versus HCWs with laboratory-confirmed infection. Anti-viral T cells recognizing influenza A, Epstein–Barr virus (EBV) and cytomegalovirus (CMV) (together, FEC) were equivalent between the three cohorts (Extended Data Fig. 2a). However, the relative immunodominance of T cells against SARS-CoV-2 structural versus RTC proteins differed between the groups. The laboratory-confirmed-infection group had more responses to structural proteins (spike, membrane, NP and ORF3a) than to RTC (NSP7, NSP12, NSP13) (Fig. 2a, b). Memory T cells against structural proteins tended to positively correlate with viral load, whereas RTC responses did not show this association (Extended Data Fig. 2b). By contrast, T cells of the SN-HCWs targeted both structural and RTC regions, with significantly more RTC-specific T cells compared with either the infected or prepandemic groups (Fig. 2a and Extended Data Fig. 2c, d). Prepandemic samples had a ratio of RTC to structural responses that did not differ significantly from that in SN-HCWs (Fig. 2b), pointing to a possible influence of pre-existing responses on the pool of T cells expanding in SN-HCWs. A further small group (10%) of HCWs had PCR-confirmed infection but lacked detectable neutralizing antibodies at week 16, some of the individuals in this group also lacked binding antibodies; this subgroup was similarly enriched for RTC-reactive T cells (Extended Data Fig. 2e, f). Taken together, this suggests that the structural proteins, which are abundantly produced during active infection, are dominant T cell targets after mild infection, whereas T cells in SN-HCWs preferentially focus on the RTC.

To confirm the T cell identity of ELISpot responses in SN-HCWs at week 16, we expanded them with RTC peptides and detected both CD4+ and CD8+ SARS-CoV-2-specific T cells dividing (CellTrace violet (CTV) dilution) and producing IFNγ (Extended Data Fig. 3a and Extended Data Table 2). Their post-expansion frequencies tended to be lower than control influenza A/EBV/CMV-specific responses in the same donors but proportional to their differing ex vivo frequencies, indicating comparable proliferative potential (Extended Data Fig. 3b). In vitro-expanded RTC-specific T cells in SN-HCWs were also highly functional, producing multiple cytokines in tandem (Extended Data Fig. 3c, d). Most of the SARS-CoV-2-specific T cells expanded from SN-HCWs were CD4+; however, CD8+ T cells were also detectable in the majority of individuals (Extended Data Fig. 3e).

Our data raised the possibility that SARS-CoV-2 infection in HCWs represents a spectrum, with some SN-HCWs expanding T cells as a result of a subclinical infection that was not detectable by PCR or antibody seroconversion. To test this postulate, we measured the interferon-inducible transcript *IFI27* in the blood, which has recently been shown to detect SARS-CoV-2 infection at, or one week before, PCR positivity (specificity of 0.95 and sensitivity of 0.84)[14]. Of the 25% of SN-HCWs with the strongest post-exposure RTC-specific T cell responses (Extended Data Fig. 4a), 40% (that is, 10% of SN-HCW group) already had *IFI27* levels at recruitment that were above the threshold set on the basis of a cohort of unexposed prepandemic samples, although their levels tended to be lower than those in individuals with a laboratory-confirmed infection (Fig. 2c). To further estimate the frequency of abortive infections we tested a larger cohort of 99 unselected SN-HCW baseline samples, and found that a comparable proportion (9.1%) had *IFI27* induction above the prepandemic threshold (Fig. 2c). The *IFI27* signal peaked above the prepandemic threshold in 93.3% of those with strong RTC-specific T cells over weeks 0–5, but in none with weak or undetectable RTC-specific T cells (Fig. 2c). *IFI27* levels showed a cumulative increase, peaking at 3–5 weeks after the UK lockdown (23 March 2020) (Fig. 2d; by which time all of the first-wave laboratory-confirmed infections had occurred (Fig. 1b)). By contrast,

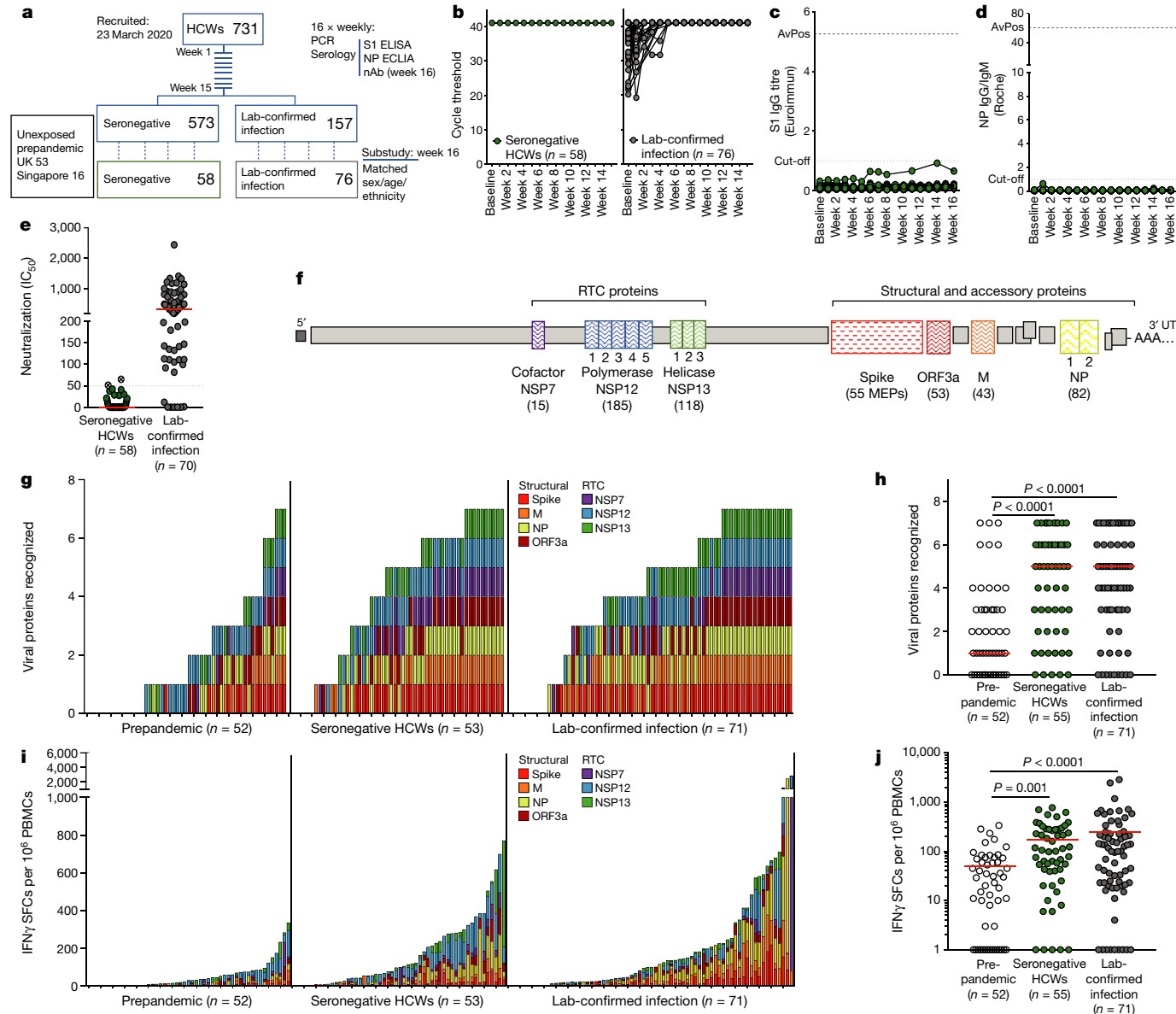

**Fig. 1 | SARS-CoV-2-specific T cells in SN-HCWs. a**, Design of the HCW and prepandemic cohorts. nAb, neutralizing antibodies. **b**, Cycle threshold values for the *E* gene PCR analysis in SN-HCWs and HCWs with a laboratory (lab)-confirmed infection (undetectable at 40 cycles was assigned 41). **c**, **d**, Anti-spike S1 (**c**) and anti-NP antibody (**d**) titres in SN-HCWs (baseline to week 16; *n* = 58; dotted lines at assay positivity cut-off and at average peak (AvPos) response in laboratory-confirmed infection). **e**, Pseudovirus neutralization at week 16. The crossed circles represent individuals who were excluded from SN-HCW group (IC$_{50}$ > 50). **f**, SARS-CoV-2 proteome highlighting RTC and structural regions assayed for T cell responses (peptide subpools are identified

by the numbered boxes) and the number of overlapping 15-mer peptides (or mapped epitope peptides (MEP) for spike). **g–j**, IFNγ ELISpot analyses. **g**, **h**, Viral proteins recognized by individuals coloured by specificity (**g**) and the number of viral proteins targeted by group (**h**). **i**, **j**, The magnitude of the T cell response coloured by viral protein (**i**) and the cumulative magnitude of the T cell response by group (**j**). The red bar shows the geometric mean. For **e**, **h**, the red bar shows the median. For **h**, **j**, statistical analysis was performed using Kruskal–Wallis tests with Dunn's correction. M, membrane; SFCs, spot-forming cells. For **b**–**e**, **g**–**j**, participants were from the COVIDsortium HCW cohort.

*IFI27* was unchanged over weeks 0–5 in SN-HCWs with weak or absent RTC-specific responses, resulting in a lower *IFI27* slope and variance (Fig. 2d and Extended Data Fig. 4b, c). The peak *IFI27* level correlated with NSP7 T cells at week 16, with the latter correlating more strongly with NSP12 and other RTC-specific responses compared with structural responses. Neither *IFI27* or T cell specificity correlated with age, sex or other demographic factors, such as exposure type, in this small cohort (Extended Data Fig. 4d and Extended Data Table 1).

In summary, during a period of high transmission at the start of the first UK pandemic wave, a low-level systemic interferon response indicative of virus exposure was detectable selectively in individuals who had

the strongest SARS-CoV-2-specific T cells after exposure, despite them lacking PCR or antibody confirmation of infection. Extrapolating from previous data showing that *IFI27* is induced at the time of incident infection and correlates with viral load[14], this is consistent with a low-level infection among SN-HCWs with stronger RTC-specific T cell responses.

## Targeting of conserved RNA polymerase

A transient/abortive infection that is not detectable by PCR or sero-conversion could conceivably result from a lower viral inoculum and/or from a more efficient innate and/or adaptive immune response. The

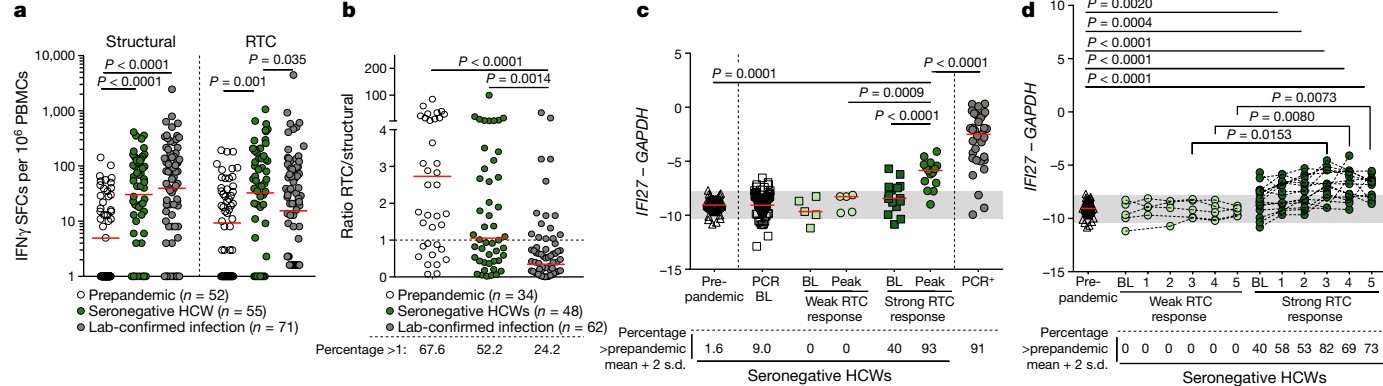

**Fig. 2 | RTC-specific T cell and *IFI27* signature in SN-HCWs. a**, **b**, IFNγ ELISpot analysis at week 16. **a**, The magnitude of T cell response to structural regions and the RTC. **b**, The ratio of the T cell response to the RTC versus structural regions. The percentage of the cohort with a ratio above 1 (RTC > structural) is shown below. For **a**, **b**, the red bar shows the geometric mean. **c**, *IFI27* transcript signal by reverse transcription PCR (RT–PCR) in unexposed prepandemic samples (*n* = 59), baseline (BL) samples in HCWs who remained PCR negative and seronegative throughout follow-up (*n* = 99), SN-HCWs with weak (*n* = 5, <50 SFCs per 10⁶ PBMCs; Extended Data Fig. 4a) or strong (*n* = 15, >50 SFCs per 10⁶

PBMCs) RTC-specific T cells (baseline and peak signal (weeks 0–5)), and HCWs at the time of PCR positivity (PCR⁺). **d**, The longitudinal *IFI27* signal in SN-HCWs with weak or strong RTC-specific T cell responses (*n* values as in **c**). For **c**, **d**, the red bar shows the median, with 2 s.d. either side of the prepandemic cohort mean highlighted in grey; the percentage with raised *IFI27* above the mean + 2 s.d. is indicated below. Statistical analysis was performed using Kruskal–Wallis analysis of variance (ANOVA) with Dunn's correction (**a**–**d**). Mann–Whitney paired *t*-test for paired BL versus peak (**c**). For **a**–**d**, participants were from the COVIDsortium HCW cohort.

latter would be favoured by pre-existing memory T cells with the potential to expand rapidly after cross-recognition of early viral products of SARS-CoV-2 replication. Early T cell proliferation and T-cell-receptor clonal expansion, even before the virus is detectable, has been observed during mild SARS-CoV-2 infection[17,30] and expansion of virus-specific T cells predates antibody induction after mRNA vaccination[2,31]. Having found that the SN-HCW group is enriched for SARS-CoV-2-specific T cells, particularly against RTC, we investigated the possibility that some of these represented expansions of pre-existing cross-reactive responses.

Probable candidates for the source of pre-existing T cells that cross-recognize SARS-CoV-2 are previous infections with closely related human endemic common cold coronaviruses (α-HCoV 229E, NL63 and β-HCoV HKU1, OC43). We bioinformatically determined the sequence homology of all possible SARS-CoV-2-derived 15-mer peptides to a curated set of HCoV sequences (Supplementary Table 1). RTC proteins, which are expressed at the first stage of the SARS-CoV-2 life cycle[13], had 15-mer sequences of high homology to HCoV[32,33] (Fig. 3a). In particular, NSP7-, NSP12- and NSP13-derived 15-mers had 6.3%, 29.9% and 31.0% higher average sequence homology to the four HCoV species, respectively, compared with structural-protein-derived 15-mers (all *P* < 0.001; Fig. 3b). NSP12, which was the largest of these proteins, represented the region with the most homology overall among human-infecting *Coronaviridae*. We further assessed the diversity across global circulating SARS-CoV-2 sequences (13,785, representative subsample of 611,893 sequences, GISAID, 27 July 2021; Extended Data Fig. 5a) using Nei's genetic diversity index and an estimate of the minimal number of independent mutational events (homoplasies) at any nucleotide. By both metrics, the RTC proteins NSP12 and NSP13 were among the most conserved across SARS-CoV-2 clades (Fig. 3c and Extended Data Fig. 5b, d) and were significantly more conserved than many structural proteins (Extended Data Table 3).

Importantly, the highly conserved RNA polymerase (NSP12) was also the region among those tested in prepandemic samples that was most commonly targeted by T cells, with the highest average magnitude and frequency of responders (Fig. 3d). Notably, the same preferential targeting of NSP12 was observed in a geographically distinct cohort of prepandemic samples from Singapore (Fig. 3d). Pre-existing T cells had the potential to recognize all of the viral antigens tested, including those with less conservation across HCoV, as previously described[5,7,17,34].

Responses against these regions were further enriched in SN-HCWs (Fig. 3d, e; Mann–Whitney *U*-test, *P* < 0.0001 for all except for ORF3a (*P* = 0.0006) and NSP13 (*P* = 0.0003)), suggesting many sources of pre-existing and de novo responses contribute to T cell memory in exposed seronegative individuals. Despite potential demographic confounding factors between cohorts (Extended Data Table 1), as with prepandemic samples, T cells of SN-HCWs preferentially targeted NSP12 (Fig. 3e). Thus, the viral protein that is most commonly targeted by pre-existing T cells is also the largest conserved region between *Coronaviridae*, suggesting exposure to HCoV is one probable source of cross-reactive T cells.

To further examine the potential for cross-reactivity due to previous infection with seasonal HCoV, we mapped new and previously described[6,8,18,35] RTC-specific CD4⁺ and CD8⁺ T cell epitopes in SN-HCWs, revealing high sequence conservation with HCoV (Extended Data Table 4 and Extended Data Fig. 6a, b). We identified cross-reactivity against the HLA-A*02:01 restricted epitope in NSP7. A subset of T cells co-stained with MHC class I pentamers loaded with SARS-CoV-2 and HKU1 sequence peptide ex vivo, and bound to SARS-CoV-2 peptide-loaded pentamer after expansion for 10 days with either peptide (Extended Data Fig. 6c). T cells from 3 out of 5 HLA-A*02:01⁺ SN-HCWs tested had stronger responses to the HKU1 sequence than to other seasonal HCoV or SARS-CoV-2 (Extended Data Fig. 6d, e). This suggested that previous HKU1 infection primed these NSP7 responses that are able to cross-recognize the SARS-CoV-2 sequence, albeit with reduced efficiency. HLA-B*35⁺ SN-HCWs also showed variable cross-recognition of seasonal HCoV variant sequences of an NSP12 epitope (Extended Data Fig. 6f).

An alternative explanation for expanded T cells with cross-reactive potential in SN-HCWs is an infection with a seasonal coronavirus during the first wave of SARS-CoV-2 infections in London. As expected, all HCWs had detectable anti-spike IgG against the four endemic HCoV and, as previously described[36], spike-specific antibodies against betacoronavirus OC43 were increased in those with PCR-detectable infection and SARS-CoV-2-specific seroconversion (Extended Data Fig. 7). However, there was no difference in endemic HCoV titres in HCWs who had strong RTC-specific T cells and raised *IFI27* compared with those with weak or absent RTC-specific responses (Extended Data Fig. 7), making it improbable that HCoV infection itself accounted for the SARS-CoV-2-reactive T cells that we detected in SN-HCWs.

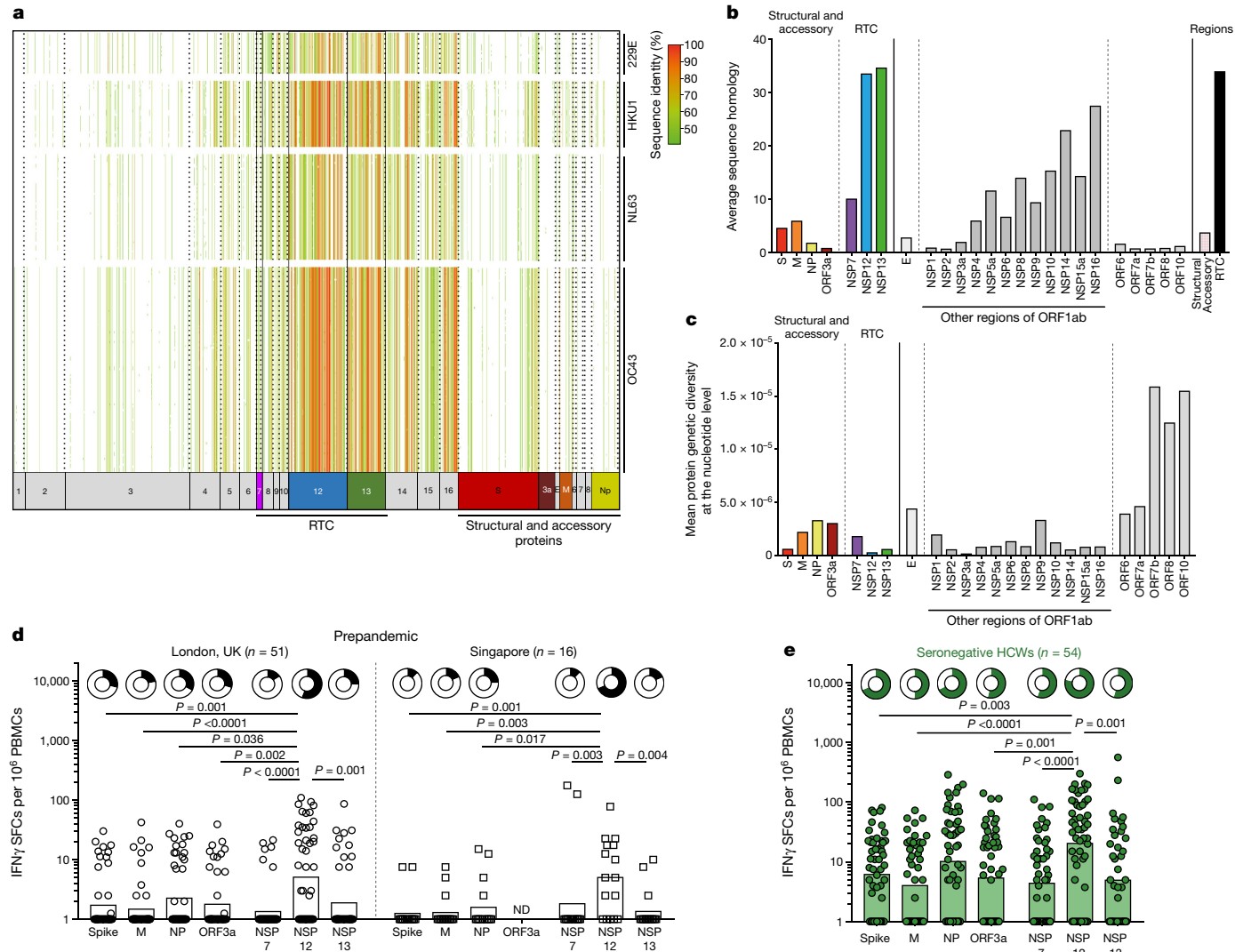

**Fig. 3 | Cross-reactive T cells targeting conserved RNA polymerase.**
**a**, Sequence homology of SARS-CoV-2-derived peptide sequences to HCoV sequences. The columns show 15-mer SARS-CoV-2-derived peptides. The rows show HCoV genome records. Cells are coloured by the level of homology of the 15-mer to a particular HCoV proteome. Cells with no fill indicate that a sequence homology of <40% was observed. **b**, The average sequence homology of 15-mers covering SARS-CoV-2 proteins, or regions (pink, structural (S, M, NP and ORF3a); black, RTC (NSP7, NSP12 and NSP13)), to HCoV sequences. Viral proteins that were not assayed for T cell responses are shown in grey. **c**, The

nucleotide diversity along the SARS-CoV-2 genome estimated with Nei's genetic diversity index across each viral protein for all SARS-CoV-2 clades (subsampling; Extended Data Fig. 5a). **d**, **e**, IFNγ ELISpot analysis of the magnitude of T cell responses to individual SARS-CoV-2 proteins in unexposed prepandemic samples (**d**) and SN-HCWs at week 16 (**e**). The frequency of responders is shown as doughnut charts above. The bar shows the geometric mean. ND, not done. Statistical analysis was performed using Kruskal–Wallis tests with Dunn's correction. Participants were from the COVIDsortium HCW cohort.

In summary, RTC regions such as polymerase that are expressed in the first stage of the viral life cycle are highly conserved among HCoV and are preferentially targeted by T cells in prepandemic and SN-HCW samples. A subset of T cells from donors who were able to abort infection could cross-recognize SARS-CoV-2 and HCoV sequences at individual RTC epitopes, pointing to previous infection with HCoV as one source of pre-existing cross-reactive T cells.

## Polymerase-specific T cells in abortive infection

To examine whether pre-existing cross-reactive and/or rapidly generated de novo RTC-specific T cells expand in vivo, we obtained paired PBMC samples before and after SARS-CoV-2 exposure. Medical students and laboratory staff (contact cohort, *n* = 23) who were sampled before the coronavirus disease 2019 (COVID-19) pandemic (winter 2018–2019), were resampled after close contact with individuals with SARS-CoV-2

infection, with or without IgG seroconversion and with or without PCR positivity (contact cohort; Extended Data Table 5). Parallel analysis of pre- and post-exposure/infection PBMCs demonstrated expansion of the RTC over structural responses in the close-contact seronegative group (Fig. 4a). By contrast, the group with serological confirmation of infection showed the expected in vivo expansion of pre-existing structural SARS-CoV-2-reactive T cells, with no significant increase in RTC-specific T cells (Fig. 4a and Extended Data Fig. 8a). We observed in vivo expansion of pre-existing NSP12 responses in 4 out of 5 individuals who remained seronegative after exposure to SARS-CoV-2, resulting in a significant increase in NSP12 but not control FEC responses (Extended Data Fig. 8b, c). Four out of five remaining seronegative close contacts had newly detected, presumed de novo, low-level responses after exposure (Extended Data Fig. 8c).

We next reverted to the SN-HCW group, in which small volume PBMC collections were available from the time of recruitment, enabling the

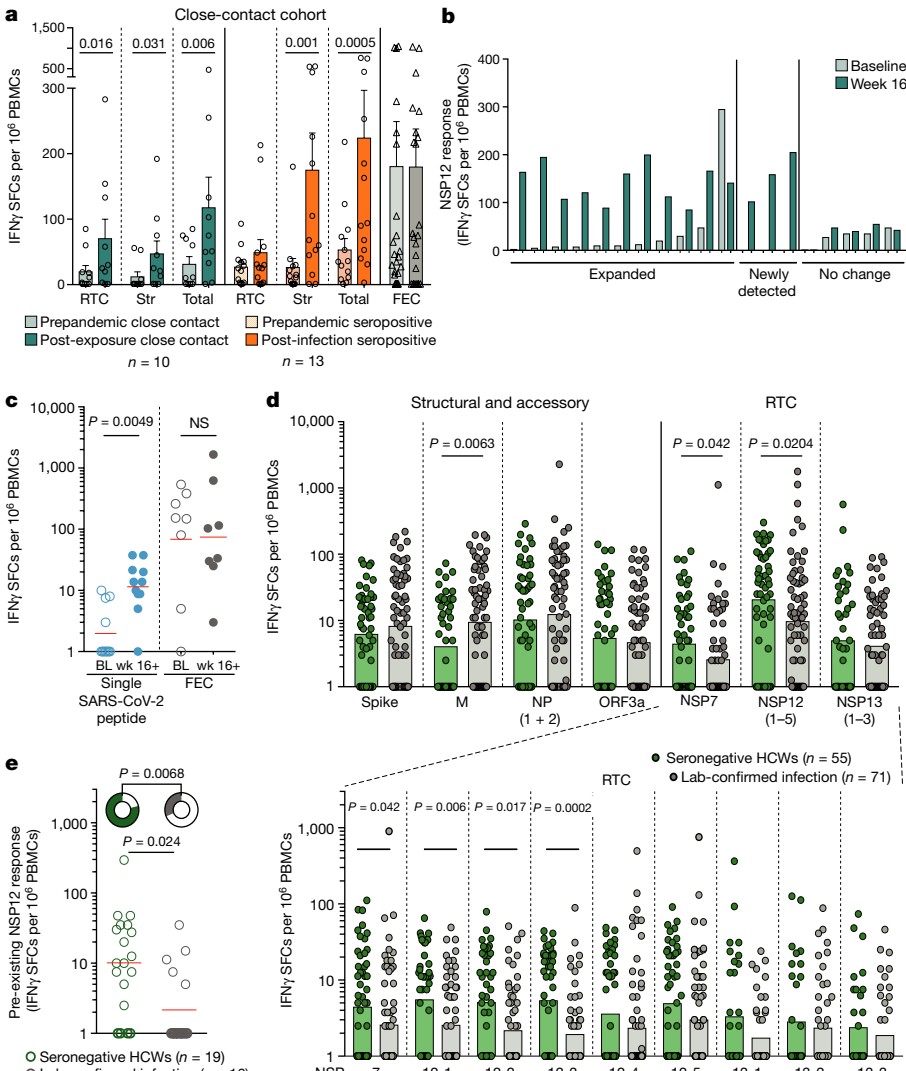

**Fig. 4 | In vivo expansion of polymerase-specific T cells in abortive infection. a**–**e**, IFNγ ELISpot analysis. **a**, The magnitude of the T cell response in seronegative individuals who had close contact with cases (green) or in seropositive individuals with infection (orange) to the RTC, structural proteins (Str), a summed total, and an influenza A, EBV and CMV (FEC) peptide pool (grey seronegative/seropositive combined), before and after exposure/ infection. Data are mean ± s.e.m. *P* values are shown at the top. **b**, The change in magnitude of NSP12 T cell response between recruitment and post-exposure in SN-HCWs (subgroup with top 19 RTC responses at week 16; Extended Data Fig. 4a). Expanded, greater than twofold change. **c**, The magnitude of paired pre- and post-exposure T cell responses to individual 9–15-mer peptides (individual responses; Extended Data Fig. 8g) from RTC or the control FEC peptide pool in SN-HCWs (weeks 16–26, 11 responses from 9 SN-HCWs). CI,

confidence intervals. **d**, The magnitude of the T cell response to individual SARS-CoV-2 proteins (top) and to subpools (~40 overlapping peptides; bottom) within the RTC at week 16 in HCWs with a laboratory-confirmed infection or SN-HCWs. **e**, Pre-existing NSP12-specific T cell responses in baseline samples from SN-HCWs and the laboratory-confirmed infection group (PCR positive after baseline or seroconversion at least 4 weeks after recruitment). The doughnut plot above shows frequency. For **c**–**e**, the red lines (**c**, **e**) and bars (**d**) show the geometric mean. Statistical analysis was performed using Wilcoxon tests (**a**, **c**), Mann–Whitney *U*-test and Fisher's exact test (**d**, **e**). For **a**, participants were from the contact cohort (Extended Data Table 5). For **b**–**e**, participants were from the COVIDsortium HCW cohort (Extended Data Table 1).

targeted analysis of baseline T cells in those with the strongest RTC responses at week 16. NSP12-specific T cells were already detectable at the baseline in 79% of those SN-HCWs with the strongest NSP12 responses after exposure (Fig. 4b). NSP12 responses expanded in vivo on average 8.4-fold between recruitment and week 16, with no corresponding change in FEC responses (Fig. 4c and Extended Data Fig. 8d). We confirmed the expansion at week 16 of pre-existing RTC-specific T cells at the subpool (Extended Data Fig. 8e, f) and individual peptide (Fig. 4c and Extended Data Fig. 8g) levels. Moreover, many T cells were newly detected after exposure (Extended Data Fig. 8g), reflecting either de novo priming or expansion of responses that were previously below the limit of assay detection (example of expanded response

undetectable by ex vivo ELISpot; Extended Data Fig. 8h). All of the HCWs with newly detected or expanding/contracting NSP12-specific T cells had both NP1- and NP2-reactive T cells after exposure (Extended Data Fig. 8i), whereas only 2 out of 5 individuals with no change in NSP12 had these specificities, suggesting that they may not have had the same level of SARS-CoV-2 exposure. The fold change in NSP12 between recruitment and the week 16 follow-up correlated with the total SARS-CoV-2 response, supporting its use to identify those seronegative individuals with expanded T cell immunity after exposure (Extended Data Fig. 8j).

Finally, we examined whether there was a preferential enrichment of RTC-specific responses in SN-HCWs compared with HCWs with a laboratory-confirmed infection at week 16. Notably, the RNA

polymerase NSP12 and its cofactor NSP7 were the only proteins that induced higher-magnitude T cell responses in seronegative individuals in whom detectable infection was not established compared with those with overt infection (Fig. 4d). T cells in SN-HCWs targeted a larger number of regions of NSP12 (subpools of about 40 overlapping 15-mers; Fig. 1g) compared with T cells in the prepandemic or seropositive cohorts (Extended Data Fig. 8k). T cells targeting several regions of NSP12 and other RTC pools were enriched in SN-HCWs compared to HCWs with a laboratory-confirmed infection (Fig. 4d (bottom)). To examine whether the reduced frequency of NSP12-specific T cells in the 16 week memory response of those with laboratory-confirmed infection was reflective of their repertoire at the time of encountering SARS-CoV-2, we obtained baseline PBMCs from a subset of individuals who were sampled before PCR positivity or more than 4 weeks before seroconversion. NSP12-specific T cells were already significantly lower at the baseline in those who went on to develop laboratory-confirmed infection compared with in SN-HCWs (Fig. 4e and Extended Data Fig. 8l, m), supporting a potential role in protection from PCR-detectable infection and seroconversion.

## Conclusions

We provide T cell and innate transcript evidence for abortive, seronegative SARS-CoV-2 infection. Longitudinal samples from SN-HCWs and an additional cohort showed that RTC-specific (particularly polymerase) T cells were enriched before exposure, expanded in vivo and preferentially accumulated in those in whom SARS-CoV-2 failed to establish infection compared with those with overt infection.

The differential biasing of T cells towards early-expressed non-structural SARS-CoV-2 proteins in HCWs without seroconversion may reflect repetitive occupational exposure to very low viral inocula, reported to drive the induction of non-structural T cells in HIV, simian immunodeficiency virus (SIV) and HBV[26,37,38]. Such repetitive exposure would be congruent with the observed protracted induction of the innate signal *IFI27* and the development of de novo T cells in some SN-HCWs.

However, we also documented the expansion of pre-existing T cells, with responses that are capable of cross-recognizing epitope variants between seasonal HCoV and SARS-CoV-2. Cross-reactive SARS-CoV-2-specifc CD8[+] T cells directed against epitopes that are highly conserved among HCoV are now well described, with pre-existing T cells frequently targeting essential viral proteins with low scope for tolerating mutational variation, such as those in ORF1ab[6,18,32]. The abundant SARS-CoV-2-specific CD4[+] T cells may also contribute to protection in SN-HCWs by antibody-independent mechanisms, such as antiviral cytokine and chemokine production. HCWs have higher frequencies of HCoV-reactive T cells compared with the general public[19], and recent HCoV infection is associated with a reduced risk of severe COVID-19 infection[39], probably in part attributable to cross-reactive neutralizing antibodies[40,41]; however, pre-existing T cells have also been implicated[15,42]. The early induction of T cells, before detectable antibodies in mild infection[30] and concurrent with mRNA vaccination efficacy, support a role for pre-existing cross-reactive memory T cells[2,31].

Pre-existing RTC-specific T cells, at a higher frequency than naive T cells and poised for immediate reactivation on antigen cross-recognition, would be expected to favour early control, explaining their enrichment after abortive compared to classical infection. However, the relative contribution of viral inoculum and cross-reactive T cells needs to be further dissected in human challenge experiments and animal models. A caveat of this work is that we analysed only peripheral immunity; it is plausible that mucosal-sequestered antibodies[43] had a role in our seronegative cohort. It also remains possible that innate immunity mediates control in abortive infections, with RTC-biased T cell responses being generated as a biomarker of low-grade infection. Interferon-independent induction of RIG-I has been proposed to abort

SARS-CoV-2 infection by restraining the viral lifecycle before sgRNA production[13]; this would favour the presentation of epitopes from ORF1ab, released into the cytoplasm in the first stage of the viral life cycle[12], while blocking the production of structural proteins from pregenomic RNA. This raises the possibility that some SARS-CoV-2-infected cells could be recognized and removed by ORF1ab-reactive T cells without widespread production of structural proteins and mature virion formation.

We have described the induction of innate and cellular immunity without seroconversion, highlighting a subset of individuals in whom the risk of SARS-CoV-2 reinfection and immunogenicity of vaccines should be specifically assessed. The HCWs who we studied were exposed to Wuhan Hu-1 and had partial protection from personal protective equipment; it remains to be seen whether abortive infections can occur after exposure to more infectious variants of concern, or in the presence of vaccine-induced immunity. However, clearance without seroconversion points to T cells that may be particularly effective vaccine targets. Cross-protection between coronaviruses is proportional to their sequence homology in mice[44], making the highly conserved NSP12 region studied here, as well as less studied NSP3/14/16, top candidates for heterologous immunity. Our data highlight the presence of pre-existing T cells in a proportion of donors that are able to expand in vivo and target a highly conserved region of SARS-CoV-2 and other *Coronaviridae*. The boosting of such T cells may offer durable pan-*Coronaviridae* reactivity against endemic and emerging viruses, arguing for their inclusion and assessment as an adjunct to spike-specific antibodies in next-generation vaccines.

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

[1]Division of Infection and Immunity, University College London, London, UK. [2]Blizard Institute, Barts and the London School of Medicine and Dentistry, Queen Mary University of London, London, UK. [3]Emerging Infectious Diseases Program, Duke-NUS Medical School, Singapore, Singapore. [4]UCL Genetics Institute, University College London, London, UK. [5]Great Ormond Street Institute of Child Health NIHR Biomedical Research Centre, University College London, London, UK. [6]Barts Heart Centre, St Bartholomew's Hospital, Barts Health NHS Trust, London, UK. [7]Department of Cellular Pathology, Northwest London Pathology, Imperial College London NHS Trust, London, UK. [8]Institute of Cardiovascular Science, University College London, London, UK. [9]Academic Rheumatology, Clinical Sciences, Nottingham City Hospital, Nottingham, UK. [10]NIHR Nottingham Biomedical Research Centre, Nottingham University Hospitals NHS Trust and University of Nottingham, Nottingham, UK. [11]Department of Immunology and Inflammation, Imperial College London, London, UK. [12]Department of Infectious Disease, Faculty of Medicine, Imperial College London, London, UK. [13]Lung Division, Royal Brompton & Harefield Hospitals, Guy's and St Thomas' NHS Foundation Trust, London, UK. [14]Singapore Immunology Network, A*STAR, Singapore, Singapore. [24]These authors contributed equally: Mariana O. Diniz, Nathalie M. Schmidt, Oliver E. Amin, Aneesh Chandran, Emily Shaw, Mahdad Noursadeghi, Antonio Bertoletti. ✉e-mail: l.swadling@ucl.ac.uk; m.maini@ucl.ac.uk

**COVIDsortium Investigators**

**Hakam Abbass**[6,8], **Aderonke Abiodun**[6,8], **Mashael Alfarih**[6], **Zoe Alldis**[6], **Daniel M. Altmann**[11], **Oliver E. Amin**[1,24], **Mervyn Andiapen**[6], **Jessica Artico**[6], **João B. Augusto**[6], **Georgina L. Baca**[6], **Sasha N. L. Bailey**[12], **Anish N. Bhuva**[6], **Alex Boulter**[6], **Ruth Bowles**[6], **Rosemary J. Boyton**[12,13], **Olivia V. Bracken**[15], **Ben O'Brien**[6,16], **Tim Brooks**[17], **Natalie Bullock**[2], **David K. Butler**[12], **Gabriella Captur**[8,18], **Nicola Champion**[6], **Carmen Chan**[6], **Aneesh Chandran**[1,24], **David Collier**[19], **Jorge Couto de Sousa**[6], **Xose Couto-Parada**[6], **Teresa Cutino-Mogue**[6], **Rhodri H. Davies**[6], **Brooke Douglas**[18], **Cecilia Di Genova**[20], **Keenan Dieobi-Anene**[6], **Mariana O. Diniz**[1,24], **Anaya Ellis**[17], **Karen Feehan**[14], **Malcolm Finlay**[6], **Marianna Fontana**[18], **Nasim Forooghi**[6], **Celia Gaier**[14], **Joseph M. Gibbons**[2], **Derek Gilroy**[21], **Matt Hamblin**[6], **Gabrielle Harker**[17], **Jacqueline Hewson**[17], **Lauren M. Hickling**[22], **Aroon D. Hingorani**[1], **Lee Howes**[8], **Alun Hughes**[8], **Gemma Hughes**[6], **Rebecca Hughes**[6,8], **Ivie Itua**[6], **Victor Jardim**[6], **Wing-Yiu Jason Lee**[6], **Melanie Petra Jensen**[6,7], **Jessica Jones**[17], **Meleri Jones**[6], **George Joy**[6,8], **Vikas Kapil**[6,19], **Hibba Kurdi**[6,8], **Jonathan Lambourne**[6], **Kai-Min Lin**[12], **Sarah Louth**[18], **Mala K. Maini**[1], **Vineela Mandadapu**[6], **Charlotte Manisty**[6,8], **Áine McKnight**[2], **Katia Menacho**[6], **Celina Mfuko**[6], **Oliver Mitchelmore**[6], **Christopher Moon**[17], **James C. Moon**[6,8], **Diana Munoz-Sandoval**[12], **Sam M. Murray**[12], **Mahdad Noursadeghi**[1,24], **Ashley Otter**[17], **Corinna Pade**[2], **Susana Palma**[6], **Ruth Parker**[23], **Kush Patel**[6], **Babita Pawarova**[18], **Steffen E. Petersen**[6], **Brian Piniera**[6], **Franziska P. Pieper**[12], **Daniel Pope**[6,19], **Mary Prossora**[6], **Lisa Rannigan**[18], **Alicja Rapala**[8], **Catherine J. Reynolds**[12], **Amy Richards**[6], **Matthew Robathan**[19], **Joshua Rosenheim**[1], **Genine Sambile**[6], **Nathalie M. Schmidt**[1,24], **Amanda Semper**[17], **Andreas Seraphim**[6], **Mihaela Simion**[18], **Angelique Smit**[18], **Michelle Sugimoto**[14], **Leo Swadling**[1], **Stephen Taylor**[17], **Nigel Temperton**[20], **Stephen Thomas**[17], **George D. Thornton**[6,8], **Thomas A. Treibel**[6,8], **Art Tucker**[6], **Jessry Veerapen**[6], **Mohit Vijayakumar**[6], **Sophie Welch**[6], **Theresa Wodehouse**[6], **Lucinda Wynne**[6] & **Dan Zahedi**[23]

[15]Division of Medicine, University College London, London, UK. [16]German Heart Centre and Charité University, Berlin, Germany. [17]National Infection Service, Public Health England, Porton Down, UK. [18]Royal Free London NHS Foundation Trust, London, UK. [19]William Harvey Research Institute, Queen Mary University of London, London, UK. [20]Viral Pseudotype Unit, , Medway School of Pharmacy, Chatham, UK. [21]Centre for Clinical Pharmacology, University College London, London, UK. [22]East London NHS Foundation Trust Unit for Social and Community Psychiatry, Newham Centre for Mental Health, London, UK. [23]School of Clinical Medicine, University of Cambridge, Cambridge, UK.

## Methods

### COVIDsortium healthcare worker participants

The COVIDsortium bioresource was approved by the ethical committee of UK National Research Ethics Service (20/SC/0149) and registered at https://ClinicalTrials.gov (NCT04318314). Full study details of the bioresource (participant screening, study design, sample collection and sample processing) have previously been described[16,45].

In this cohort and London as a whole, infections peaked for the first pandemic wave of infections during the first week of lockdown (23 March 2020)[46], and we observed approximately synchronous exposure coincident with recruitment; we therefore used this as the benchmark for assessing exposure-generated immunity. Across the main study cohort, 48 participants had positive RT–PCR results with 157 (21.5%) seropositive participants. Furthermore, 79% of positive PCR tests were within the first 2 weeks of follow-up and no HCWs tested PCR positive after week 5 of follow-up[14,46] (Fig. 1b), with seroconversion within the first 3 weeks of follow-up for most[28]. Infections were asymptomatic or mild with only two hospital admissions (none requiring intensive care admission). The cross-sectional case controlled substudy ($n$ = 129) collected samples at 16–18 weeks after the first UK lockdown (Fig. 1a). Power calculations were performed before week 16 substudy sampling to determine the sample size needed to test the hypothesis that HCWs with pre-existing T cell responses are enriched in exposed seronegative group at a range of incidence of infection, assuming 50% of the total cohort had pre-existing T cell responses. Sample sizes of 18–64 per group were estimated. An age-, sex- and ethnicity-matched nested substudy was designed within the larger ($n$ = 731) parent study and 129 attended for 16 week sampling, including high-volume PBMC isolation.

Laboratory-confirmed infection was determined by weekly nasopharyngeal RNA stabilizing swabs and RT–PCR (Roche cobas SARS-CoV-2 test, envelope ($E$) gene) and antibody assay positivity (spike protein 1 IgG Ab assay, EUROIMMUN) and anti-nucleocapsid total antibody assay (Roche) described in detail below. The seronegative HCW group was matched for demographics and exposure to the laboratory-confirmed infected group and was defined by negativity in these 3 tests at all 16 time points as well as negative for neutralizing antibodies at week 16 and at selected prior time points as indicated.

The cohort of medical students and laboratory staff was approved by UCL Ethics (project ID:13545/001) and prepandemic samples from healthy donors were collected and cryopreserved before August 2019 under ethics number 11/LO/0421. All participants provided written informed consent and the study conformed to the principles of the Helsinki Declaration.

### Isolation of PBMCs and serum

PBMCs were isolated from heparinized blood samples using Pancoll (Pan Biotech) or Histopaque-1077 Hybri-Max (Sigma-Aldrich) density-gradient centrifugation in SepMate tubes (StemCell) according to the manufacturer's specifications. Isolated PBMCs were cryopreserved in fetal calf serum (FCS) containing 10% DMSO and stored in liquid nitrogen.

Whole-blood samples were collected in SST vacutainers (Vacuette) with inert polymer gel for serum separation and clot activator coating. After centrifugation at 1,000$g$ for 10 min at room temperature, the serum layer was aliquoted and stored at −80 °C. All T cell assays reported here were performed on cryopreserved PBMCs.

### Weekly SARS-CoV-2 S1 and NP serology

Weekly Euroimmun anti-SARS-CoV-2 enzyme-linked immunosorbent assay (ELISA; anti-SARS-CoV-2 S1 antigen IgG and the Roche Elecsys anti-SARS-CoV-2 electrochemiluminescence immunoassay (ECLIA; anti-SARS-CoV-2 nucleoprotein IgG/IgM) commercial assays were performed by Public Health England as previously described[16]. S1 ELISA: A ratio of ≥1.1 was deemed to be positive. A ratio of 11 was taken to be the upper threshold as the assay saturates beyond this point. NP ECLIA: anti-NP results are expressed as a cut-off index (COI) value based on the electrochemiluminescence signal of a two-point calibration, with results COI ≥ 1.0 classified as positive.

### Neutralization assays for the pseudotype and authentic virus

SARS-CoV-2 pseudotype neutralization assays were conducted using pseudotyped lentiviral particles as previously described[16]. In brief, serum was heat-inactivated at 56 °C for 30 min. Serum dilutions in DMEM were performed in duplicate with a starting dilution of 1 in 20 and 7 consecutive twofold dilutions to a final dilution of 1/2,560 in a total volume of 100 μl. SARS-CoV-2 pseudotyped lentiviral particles ($1 \times 10^5$ RLU) were added to each well (serum dilutions and controls) and incubated at 37 °C for 1 h. Then, $4 \times 10^4$ Huh7 cells suspended in 100 μl complete medium were added per well and incubated for 72 h at 37 °C and 5% $CO_2$. Firefly luciferase activity (luminescence) was measured using the Steady-Glo Luciferase Assay System (Promega) and a CLARIOStar Plate Reader (BMG Labtech). The curves of relative infection rates (as a percentage) versus the serum dilutions ($\log_{10}$-transformed values) against a negative control of pooled sera collected before 2016 (Sigma-Aldrich) and a positive neutralizer were plotted using Prism 9 (GraphPad). A nonlinear regression method was used to determine the dilution fold that neutralized 50% ($IC_{50}$).

Authentic SARS-CoV-2 microneutralization assays were performed as previously described[47]. In brief, a mixture of serum dilutions in DMEM (1 in 20 and 11 consecutive twofold dilutions to a final dilution of 1/40,960) and $3 \times 10^4$ FFU of SARS-CoV-2 virus (Wuhan Hu-1) were incubated at 37 °C for 1 h. After initial incubation, preseeded Vero E6 cells were infected with the serum–virus samples and incubated (37 °C and 5% $CO_2$) for 72 h. Cells were then fixed with 100 μl 3.7% (v/v) formaldehyde for 1 h. Cells were washed with PBS and stained with 0.1% (w/v) crystal violet solution for 10 min. After removal of excess crystal violet and air drying, the crystal violet stain was resolubilized with 100 μl 1% (w/v) sodium dodecyl sulfate solution. Absorbance readings were taken at 570 nm using a CLARIOStar Plate Reader (BMG Labtech). Absorbance readings for each well were standardized against technical positive (virus control) and negative (cells only) controls on each plate to determine a percentage neutralization value. A nonlinear regression (curve fit) method was used to determine the dilution fold that neutralized 50% ($IC_{50}$) using Prism 9 (GraphPad). SARS-CoV-2 is classified as a hazard group 3 pathogen and therefore all authentic SARS-CoV-2 propagation and microneutralization assays were performed in a containment level 3 facility.

### Spike ELISA

Seropositivity against SARS-CoV-2 spike was determined for medical student and laboratory staff cohort between July 2020 and Jan 2021 (Extended Data Table 5) by ELISA, as validated and described previously[40,48,49]. In brief, 9 columns of 96-half-well MaxiSorp plates (Thermo Fisher Scientific) were coated overnight at 4 °C with purified S1 protein in PBS (3 μg ml⁻¹ per well in 25 μl), the remaining 3 columns were coated with goat anti-human F(ab)'2 (1:1,000) to generate in internal standard curve. The next day, plates were washed with PBS-T (0.05% Tween-20 in PBS) and blocked for 1 h at room temperature with assay buffer (5% milk powder PBS-T). Sera were diluted in blocking buffer (1:50). Serum (25 μl) was then added to S1 coated wells in duplicate and incubated for 2 h at room temperature. Serial dilutions of known concentrations of IgG were added to the F(ab)'2 IgG-coated wells in triplicate (Sigma-Aldrich). After incubation for 2 h at room temperature, the plates were washed with PBS-T and 25 μl alkaline phosphatase-conjugated goat anti-human IgG (Jackson ImmunoResearch) at a 1:1,000 dilution in assay buffer added to each well and incubated for 1 h room temperature. Plates were then washed with PBS-T, and 25 μl of alkaline phosphatase substrate (Sigma-Aldrich) added. Optical density values were measured using a MultiskanFC (Thermo Fisher Scientific) plate reader at 405 nm

and S1-specific IgG titres interpolated from the IgG standard curve using 4PL regression curve-fitting on GraphPad Prism 8.

## HCoV spike meso-scale discovery immunoassay

A multiplexed meso-scale discovery immunoassay immunoassay to measure anti-HCoV spike IgG antibodies was performed as previously described[50]. Plates were coated with 200–400 µg ml$^{-1}$ spike protein (trimers in prefusion form) from the endemic human coronaviruses HKU1, OC43, 229E and NL63. Antibody concentration is presented in arbitrary units (AU) interpolated from the ECL signal of the internal standard sample using a four-parameter logistic curve fit. Serum samples taken at week 8—the peak time point for spike S1 IgG after PCR-positive SARS-CoV-2 infection—were assayed for HCoV antibodies.

## SARS-CoV-2 spike-specific memory B cell staining

Multiparameter flow cytometry was used for ex vivo identification of spike-specific memory B cells staining as previously described[29]. Biotinylated tetrameric spike (1 µg) was fluorochrome linked by incubating with streptavidin-conjugated APC (Prozyme) and PE (Prozyme) for 30 min in the dark on ice. PBMCs were thawed and incubated with Live/Dead fixable dead cell stain (UV, Thermo Fisher Scientific) and saturating concentrations of phenotyping monoclonal antibodies were diluted in 50% 1× PBS 50% Brilliant Violet Buffer (BD Biosciences): anti-CD3 Bv510 (BioLegend, OKT3, 1:200), anti-CD11c FITC (BD Biosciences, B-ly6, 1:100), anti-CD14 Bv510 (BioLegend, M5E2, 1:200), anti-CD19 Bv786 (BD Bioscience, HIB19, 1:50), anti-CD20 AlexFluor700 (BD Biosciences, 2H7, 1:100), anti-CD21 Bv711 (BD Biosciences, B-ly4, 1:100), anti-CD27 BUV395 (BD Biosciences, L128, 1:100), anti-CD38 Pe-CF594 (BD Biosciences, HIT2, 1:200), IgD Pe-Cy7 (BD Biosciences, IA6-2, 1:100). For identification of SARS-CoV-2-antigen-specific B cells, 1 µg per 500 µl of stain each of tetrameric spike–APC and spike–PE were added to cells. Cells were incubated in the staining solution for 30 min at room temperature, washed with PBS and subsequently fixed with the FoxP3 Buffer Set (BD Biosciences) according to the manufacturer's instructions. All of the samples were acquired on the BD Fortessa-X20 flow cytometer. Data were analysed using FlowJo v.10.7 (TreeStar). Example gating and positivity cut-off have previously been reported[29]. The magnitude of the SARS-CoV-2 spike-specific memory B cell population is expressed as a percentage of memory B cells (gated as: lymphocytes, singlets, Live, CD3$^-$CD14$^-$CD19$^+$, CD20$^+$, excluding: CD38$^{hi}$, IgD$^+$ and CD27$^+$CD21$^-$) binding both PE- and APC-labelled spike.

## SARS-CoV-2 peptides

Full lists of the peptides contained in pools of overlapping peptides covering structural[16] and RTC proteins[8] have previously been described (15-mer peptides overlapping by 10 amino acids, GL Biochem Shanghai, >80% purity). A list of peptides that overlap NSP12 is provided in Supplementary Table 3. For IFNγ ELISpot assays, SARS-CoV sequence peptides were used (96.5% sequence homology with Wuhan SARS-CoV-2 consensus sequence, 34/931 amino acids differ; Supplementary Table 3). For epitope mapping, SARS-CoV-2 sequence peptides were used for NSP12-2 and NSP12-5 (GL Biochem Shanghai, >80% purity).

To limit competition for in vitro peptide presentation, we limited stimulations to a maximum of 55 peptides and have, therefore, divided large proteins such as NP into subpools: NP (NP1, NP2, 41 peptides each), M (43 peptides), ORF3a (53 peptides), NSP7 (15), NSP12 (36–37 per pool NSP12-1 to NSP12-5) and NSP13 (39–40 peptides per pool NSP13-1 to NSP13-3). Fifteen-mer peptides covering the predicted SARS-CoV-2 spike epitopes[8] were used to give a total of 55 peptides in this pool (spike). Optimal 9-mer peptides for CD8$^+$ epitopes were custom-synthesized by ThinkPeptides (>70% purity; Supplementary Table 3).

## IFNγ ELISpot assay

The IFNγ ELISpot assay was performed as previously described on cryopreserved PBMCs[8,16,51]. Unless otherwise stated, culture medium

for human PBMCs (R10) was sterile 0.22-µm-filtered RPMI medium (Thermo Fisher Scientific) supplemented with 10% by volume heat-inactivated (1 h, 64 °C) FCS (Hyclone) and 1% by volume 100× penicillin and streptomycin solution (Gibco-BRL).

ELISpot plates (Merck-Millipore, MSIP4510) were coated with human anti-IFNγ antibodies (1-D1K, Mabtech; 10 µg ml$^{-1}$) in PBS overnight at 4 °C. The plates were washed six times with sterile PBS and blocked with R10 for 2 h at 37 °C with 5% CO$_2$. PBMCs were thawed and rested in R10 for 3 h at 37 °C with 5% CO$_2$ before being counted to ensure that only viable cells were included. PBMCs (400,000 per well) were seeded in R10 and were stimulated for 16–20 h with SARS-CoV-2 peptide pools (2 µg ml$^{-1}$ per peptide) at 37 °C in a humidified atmosphere with 5% CO$_2$. In cases in which insufficient T cells were available, NSP12 pools 1, 2 and 3, and NSP13 pools 1, 2 and 3 were combined into a single well. For baseline measurements, NSP12 pools 1–5 were stimulated in a single well and, in cases in which insufficient T cells were available, a single DMSO well was included. HCWs who did not have a full complement of stimulations were excluded from analysis of total magnitude of breadth of response, resulting in slightly lower $n$ values. Internal plate controls were R10 alone (without T cells) and two DMSO wells (negative controls), concanavalin A (ConA, positive control; Sigma-Aldrich) and FEC (HLA I-restricted peptides from influenza, Epstein–Barr virus and CMV; 1 µg ml$^{-1}$ per peptide). ELISpot plates were developed with human biotinylated IFNγ detection antibodies (7-B6-1, Mabtech; 1 µg ml$^{-1}$) for 3 h at room temperature, followed by incubation with goat anti-biotin alkaline phosphatase (Vector Laboratories; 1:1,000) for 2 h at room temperature, both diluted in PBS with 0.5% BSA by volume (Sigma-Aldrich), and finally with 50 µl per well of sterile filtered BCIP/NBT Phosphatase Substrate (Thermo Fisher Scientific) for 7 min room temperature. Plates were washed in double-distilled H$_2$O and left to dry overnight before being read on the AID classic ELISpot plate reader (Autoimmun Diagnostika).

The average of two DMSO wells was subtracted from all peptide-stimulated wells for a given PBMC sample and any response that was lower in magnitude than 2 s.d. of these sample specific DMSO control wells was not considered to be a peptide-specific response (given value 0). Results were expressed as IFNγ SFCs per 10$^6$ PBMCs after background subtraction. The geometric mean of all DMSO wells was 9.571 SFCs per 10$^6$ PBMCs (3.8 spots). We excluded the results if the negative control wells had >95 SFC per 10$^6$ PBMCs or positive control wells (ConA) were negative. T cell responses to SARS-CoV-2 did not correlate with background spots in DMSO wells (for example, the SN-HCW group, Spearman $r = -0.068$, $P = 0.6141$).

## Antigen-specific T cell proliferation assay and epitope mapping

Frozen PBMCs were thawed and washed twice with sterile PBS. PBMC were resuspended in 1 ml R10 culture medium (2–10 × 10$^6$ PBMCs) and 0.5 µl of 5 mM stock CTV (Thermo Fisher Scientific) was added per sample with mixing. PBMCs were stained in the dark for 10 min at 37 °C in a humidified atmosphere with 5% CO$_2$. Ten-times volume of cold R10 was added to stop the staining reaction, and cells were incubated for 5 min on ice. Cells were washed in PBS and incubated for 5 min at 37 °C before being transferred to a new tube and were washed again in R10. CTV stained PBMC were plated in 96-well plates (2–4 × 10$^5$ PBMCs in 200 µl R10) and stimulated with peptide pools (2 µg ml$^{-1}$ per peptide) for 10 days in R10 supplemented with 0.5 µg ml$^{-1}$ soluble anti-CD28 antibodies (Thermo Fisher scientific) and 20 U ml$^{-1}$ recombinant human IL-2 (Peprotech). CTV-stained and unstained PBMCs were run to confirm efficiency of staining. Then, 100 µl medium was added on day 1, and 100 µl medium was removed and replaced with R10 supplemented with anti-CD28 and IL-2 as above on days 3 and 6. On day 9, PBMCs were restimulated with peptide pools (2 µg ml$^{-1}$ per peptide) and brefeldin A (10 µg ml$^{-1}$; Sigma-Aldrich). After 16–18 h restimulation, PBMCs were collected, washed in PBS and stained for fixable live/dead (Near infrared, Thermo Fisher Scientific, 1:1,000), washed in PBS, before being

fixed in fix/perm buffer (TF staining buffer kit, eBioscience) for 20 min room temperature. Cells were washed in PBS and incubated in perm buffer (TF staining buffer kit, diluted 1:10 in double-distilled $H_2O$) for 20 min room temperature, washed in PBS and resuspended in perm buffer with saturating concentrations of anti-human antibodies for intracellular staining: anti-IL-2 PerCp-eFluor710 (Invitrogen, MQ1-17H12, 1:50), anti-TNFα FITC (BD Bioscience, MAb11, 1:100), anti-CD8α BV785 (BioLegend, RPA-T8, 1:200), anti-IFNγ BV605 (BD Biosciences, B27, 1:100), anti-IFNγ APC (BioLegend, 4S.B3, 1:50), anti-CD3 BUV805 (BD Biosciences, UCHT1, 1:200), anti-CD4 BUV395 (BD Biosciences, SK3, 1:200), anti-CD154 (CD40L) Pe-Cy7 (BioLegend, 24-31, 1:50) and anti-MIP-1β PE (BD Biosciences, D21-1351, 1:100). Cells were washed twice in PBS and analysed using the BD LSRII flow cytometer. Cytometer voltages were consistent across batches. Fluorescence minus one (FMOs) and unstimulated samples were used to determine gates applied across samples. Data were analysed using FlowJo v.10.7 (TreeStar).

Optimization experiments showed that the use of recombinant human IL-2 increases non-peptide specific proliferation of T cells but is essential for optimal expansion of proliferating cytokine producing peptide-specific T cells. CTV dilution and staining with anti-human-IFNγ antibodies was used to identify antigen-specific T cells. An unstimulated control well (equivalent DMSO to peptide wells added) was included for each PBMC sample and the percentage of $CTV^{lo}IFN\gamma^+CD4^+$ or $CD8^+$ cells proliferating in unstimulated wells was subtracted as background cytokine release from all peptide stimulated wells. The T cell proliferation assay above was used to expand SARS-CoV-2-specific T cells and a two-dimensional matrix (Supplementary Table 2) was used such that each 15-mer peptide was represented in 2 pools, aiding the identification of individuals immunogenic 15-mer peptides. T cell responses were then confirmed by repeated expansion with individual 15-mers.

Polyfunctionality, defined as the number of cytokines co-produced by T cells after expansion for 10 days, was assessed using SPICE (v.6.0) and pestle (v.2.0), available at GitHub (https://niaid.github.io/spice/)[52]. Responses <0.1% of $CD4^+$ or $CD8^+$ T cells were excluded. Boolean gating was used to identify the percentage of T cells making the 31 possible combinations of the following cytokines: IFNγ, TNF, IL-2, CD154, MIP-1β. Pestle was used to background-subtract the percentage of cytokine-producing cells from unstimulated wells that were run in parallel and to format data for visualization in SPICE. The proportion of T cells making a specific number of cytokines in combination is presented as pie graphs (base mean) and pie arcs represent the proportion making a given cytokine. The RTC-specific T cell polyfunctionality was calculated as an average over T cell responses to NSP7, NSP12 and NSP13 and the structural-specific T cell polyfunctionality is an average of responses to spike, ORF3a, M and NP (Extended Data Fig. 3d).

## MHC class I pentamer staining

HLA-A*02-restricted pentamers (Proimmune) of the following specificities were used: SARS-CoV-2 $NSP7_{27-35}$ (KLWAQCVQL) or HCoV HKU1 $NSP7_{27-35}$ (KLWQYCSVL; ex vivo stains only). For post-expansion staining, antigen-specific T cells were expanded with a cognate 9-mer peptide of SARS-CoV-2 or HCoV HKUW sequence for 8–10 days as above (2 μg ml⁻¹ per peptide) in R10 supplemented with 0.5 μg ml⁻¹ soluble anti-CD28 antibodies and 20 U ml⁻¹ recombinant human IL-2; medium was added on days 1, 3 and 6 before pentamer staining. For ex vivo staining, PBMCs were thawed, washed twice in PBS. Pentamers were centrifuged at 13,000 rpm. for 10 min before use. PBMCs ($0.5-2 \times 10^6$) were stained with 1 μl pentamers at room temperature for 20 min in 50 μl PBS in a 96-well plate. PBMCs were further stained with Blue fixable Live/dead (Invitrogen, 1:1,000) for 20 min at 4 °C, and surface-stained with a mixture of saturating concentrations of monoclonal antibodies for 30 min at 4 °C: anti-CD3 BUV805 (BD Biosciences, UCHT1, 1:200), anti-CD4 BUV395 (BD Biosciences, SK3, 1:200), anti-CD56 Pe-Cy7 (BD Biosciences, NCAM16.2, 1:100), anti-CD8α Alexa700 (BioLegend, RPA-78, 1:200), post-expansion CD19 Bv786 (BD Biosciences, HIB19, 1:100).

PBMCs were fixed with 1% paraformaldehyde and flow cytometry was performed as above using a BD LSRII flow cytometer. Data were analysed using FlowJo v.10.7 (TreeStar). During analysis, stringent gating criteria were applied (the gating strategy is shown in Extended Data Fig. 6c) with doublet, dead cell, $CD19^+$ B cell (post-expansion) and $CD56^+$ NK/NKT exclusion to minimize non-specific binding contamination. HLA-mismatched PBMC (non-HLA-A*02) and fluorescence minus one controls for pentamers were stained in parallel to assess non-specific binding (Extended Data Fig. 6c).

## *Coronaviridae* family sequence homology analyses

The sequence homology of SARS-CoV-2-derived peptides to HCoV sequences was computed as previously described[32]. In brief, the SARS-CoV-2 proteome (NC_045512.2) was decomposed into 15-mer peptide sequences overlapping by 14 amino acids. A protein BLAST search of each 15-mer peptide was then performed against a custom sequence database comprising 2,531 *Coronaviridae* sequences[32]. Homology values of each SARS-CoV-2-derived peptide to viral accessions with '229E', 'OC43', 'NL63' or 'HKU1' included in the species name and that were isolated from human hosts were retained (Supplementary Table 1). Moreover, to determine whether the conservation of 15-mer peptides differed between the SARS-CoV-2 proteins, the average homology of peptides within each protein was computed. A permutation test was conducted to test whether the difference in average homology between the two proteins, Δh, was statistically significant. In brief, the protein membership of each 15-mer peptide was permuted (1,000 iterations). The Δh of two proteins was then calculated at each iteration, resulting in a final null distribution of Δh values. P values were computed as the number of permutations that yielded a Δh at least as extreme as the observed Δh of the two proteins. Custom scripts used to perform the homology searches, heatmap visualization and permutation testing are available at GitHub (https://github.com/cednotsed/tcell_cross_reactivity_covid.git).

For sequence alignments of immunogenic 15-mers or at described MHC class I-restricted epitopes, reference protein sequences for ORF1ab (accession numbers: QHD43415.1, NP_828849.2, YP_009047202.1, YP_009555238.1, YP_173236.1, YP_003766.2 and NP_073549.1) were downloaded from the NCBI database (https://www.ncbi.nlm.nih.gov/protein/) as previously described[8]. Sequences were aligned using the MUSCLE algorithm with the default parameters and percentage identity was calculated in Geneious Prime 2020.1.2 (www.geneious.com). Alignment figures were generated using Snapgene 5.1 (GSL Biotech).

## SARS-CoV-2 species genome diversity analyses

For genome diversity analysis, a complete masked alignment was downloaded from the GISAID[53,54] EpiCoV database on 26 July 2021 together with a GISAID Audacity phylogeny comprising 611,893 accessions (a full list and metadata are available at Figshare (https://figshare.com/s/049d53f789a8b111b87e)). The alignment was subsampled to include 800 of each defined NextStrain phylogenetic clade, as provided by GISAID metadata. For clades containing less than 800 accessions, all representatives of that clade were included, resulting in a comprehensive sampling over the global phylogeny of 13,785 accessions encompassing the genomic diversity of SARS-CoV-2 to date (Extended Data Fig. 5a). Diversity along the genome was assessed using two metrics of diversity: the number of recurrent mutational emergences (homoplasies) at any position and Nei's genetic diversity index[55]. Homoplasy counts per locus were computed through application of the HomoplasyFinder screening pipeline[56] against a maximum likelihood phylogeny constructed over the 13,785-genome alignment. Nei's genetic diversity index was computed as $H = 1 - \sum_{i=1}^{l} p_i^2$, where $l$ is the count of distinct alleles at a position, and $p_i = (i=1,...,l)$ is the frequency of allele $i$ in the studied alignment. The average homoplasy count per locus per gene region and average Nei's genetic diversity per gene region were computed by normalizing the per-locus values to gene length for all

ORFs and NSPs according to the reference annotations of GISAID reference genome EPI_ISL_402124. Significant differences between all pairwise combinations of ORF/NSP were assessed using the Wilcoxon rank-sum test implemented in compare_means() in the R package ggpubr v.0.4.0 (Extended Data Table 3).

### IFI27 qPCR

Total RNA from Tempus blood was extracted using the Tempus Spin RNA isolation kit (Applied Biosystems, 4380204). cDNA was obtained using the High-Capacity cDNA Reverse Transcription Kit (Applied Biosystems). Quantitative PCR (qPCR) was performed using the TaqMan Fast Advanced Master Mix (Applied Biosystems) on the ABI StepOnePlus Real-Time PCR machine (Applied Biosystems). The following cycling conditions were used: 95 °C for 2 min; followed by 40 cycles of 95 °C for 3 s and 60 °C for 30 s. IFI27 and GAPDH were amplified using the TaqMan Gene Expression Assay probes Hs01086373_g1 (IFI27) and Hs02786624_g1 (GAPDH). GAPDH was used as a housekeeping gene control. The unexposed prepandemic control HCW cohort for qPCR analysis was described previously[57]

### Correlogram plot

A pairwise correlation matrix between variables was calculated and visualized as a correlogram using corrplot (https://github.com/taiyun/corrplot) in R v.3.5.3 with R studio v.1.0.153. The Spearman's rank correlation coefficient $r$ is indicated by the size and colour of the circles. Only correlations with $P < 0.05$ are shown. Variables are ordered by hierarchical clustering.

### Statistics and reproducibility

Data were assumed to have a non-Gaussian distribution and nonparametric tests were used throughout. For single-paired and unpaired comparisons, Wilcoxon matched-pairs signed-rank tests and a Mann-Whitney $U$-tests were used. For multiple unpaired comparisons, Kruskal–Wallis one-way ANOVA with Dunn's correction was used. For correlations, Spearman's $r$ test was used. $P < 0.05$ was considered to be significant. Prism v.7.0e and v.8.0 for Mac was used for analysis. Details are provided in the figure legends.

### Data reporting

Power calculations were used to estimate the sample size needed for the week 16 substudy (see above). No statistical methods were used to predetermine the sample size. For all of the assays, samples from each cohort were run in parallel to reduce the impact of interbatch technical variation. IFNγ ELISpot assays were performed on HCW cohorts before unblinding of the group (laboratory-confirmed infection or SN-HCW). Other experiments were not randomized and the investigators were not blinded to allocation during experiments and outcome assessment.

### Reporting summary

Further information on research design is available in the Nature Research Reporting Summary linked to this paper.

## Data availability

All data analysed during this study are included in this published article and its Supplementary Information. Genomic data analysed was obtained from the publicly available NCBI Virus database and, after registration, from the GISAID EpiCoV repository. The datasets generated during and/or analysed during the current study are available from the corresponding authors on reasonable request. Correspondence and requests for materials should be addressed to M.K.M. or L.S. Source data are provided with this paper.

## Code availability

Custom scripts that were used to perform the homology searches, heatmap visualization and permutation testing are available at GitHub (https://github.com/cednotsed/tcell_cross_reactivity_covid.git).

**Acknowledgements** We thank all of the patients and control volunteers who participated in this study and all of the clinical staff who helped with recruitment and sample collection; J. Evans at the Rayne Building FACS facility for assistance with flow cytometry assays; and members of all of the contributing and submitting laboratories around the globe who have openly shared large numbers of UK SARS-CoV-2 assemblies. A full list of acknowledgements providing submitting and originating laboratories is provided at Figshare (https://figshare.com/s/049d53f789a8b111b87e). The COVIDsortium is supported by funding donated by individuals, charitable Trusts and corporations, including Goldman Sachs, K. C. Griffin, The Guy Foundation, GW Pharmaceuticals, Kusuma Trust and Jagclif Charitable Trust, and enabled by Barts Charity with support from UCLH Charity. Wider support is acknowledged on the COVIDsortium website. Institutional support from Barts Health NHS Trust and Royal Free NHS Foundation Trust facilitated study processes, in partnership with University College London and Queen Mary University London. This study was funded by UKRI/NIHR UK-CIC (supporting L.S. and M.K.M.). M.K.M. is also supported by Wellcome Trust Investigator Award (214191/Z/18/Z) and CRUK Immunology grant (26603), and L.S. by a Medical Research Foundation fellowship (044-0001). M.N. is supported by the Wellcome Trust (207511/Z/17/Z) and by NIHR Biomedical Research Funding to UCL and UCLH; A.B. by a Special NUHS COVID-19 Seed Grant Call, Project NUHSRO/2020/052/RO5+5/NUHS-COVID/6 (WBS R-571-000-077-733). J.C.M., C.M. and T.A.T. are directly and indirectly supported by the University College London Hospitals (UCLH) and Barts NIHR Biomedical Research Centres and through the British Heart Foundation (BHF) Accelerator Award (AA/18/6/34223). T.A.T. is funded by a BHF Intermediate Research Fellowship (FS/19/35/34374). A.M.V., Á.M., C.M. and J.C.M. were supported by the UKRI/MRC Covid-19 Rapid response grant COV0331. Á.M. and C.P. are supported by Rosetrees Trust, The John Black Charitable Foundation, and Medical College of St Bartholomew's Hospital Trust and NIHR-MRC grant MR/V027883/1. R.J.B. and D.M.A. are supported by the MRC (MR/S019553/1, MR/R02622X/1, MR/V036939/1, MR/W020610/1), NIHR Imperial Biomedical Research Centre (BRC): ITMAT, Cystic Fibrosis Trust SRC (2019SRC015), and Horizon 2020 Marie Skłodowska-Curie Innovative Training Network (ITN) European Training Network (no. 860325). Funding for the HLA imputed data was provided by UKRI/MRC COVID-19 rapid response grant (Cov-0331, MR/V027883/1). L.E.M. is supported by a Medical Research Council Career Development Award (MR/R008698/1). L.v.D. is supported by a UCL Excellence Fellowship. The funders had no role in study design data collection, data analysis, data interpretation or writing of the report.

**Author contributions** M.K.M. conceived the project and obtained funding. L.S., M.N., A.B. and M.K.M. designed experiments. C.M., T.A.T., J.C.M., M.N. and Á.M. established the HCW cohort. L.S., M.O.D., N.M.S., O.E.A., C.P., J.M.G., S.K., G.A.-M., J.R., J.D. and G.J. collected or processed HCW samples with COVIDsortium investigators. M.K.M., L.S., M.N. and E.S. established medical student/laboratory staff and prepandemic cohorts (UK). L.E.M., M.J., D.G. and COVIDsortium investigators performed serology. M.K.M. and L.S. designed T cell experiments. L.S., M.O.D., N.M.S. and O.E.A. developed, performed and analysed the T cell experiments. A.B., N.L.B., A.T.T. and C.Y.L.T. performed T cell assays and analysed data from the prepandemic cohort (Singapore). A.C. performed and analysed blood transcriptomic experiments. A.M.V. supervised HLA analysis. Á.M. supervised neutralizing antibody experiments. J.M.G. and C.P. performed and analysed neutralizing antibody experiments. C.C.S.T., L.v.D. and F.B. performed viral sequence analysis. E.S., M.P.J., G.J., R.J.B., C.M., T.A.T., J.C.M., Á.M. and M.O.D. provided or processed essential clinical data. M.J. and D.G. performed and analysed HCoV serology. L.S., M.O.D., N.M.S., O.E.A., A.C., C.P., J.M.G., N.L.B., A.T.T., A.J.-S., C.C.S.T., C.Y.L.T., A.M.V., B.M.C., D.G., L.v.D., D.M.A., R.J.B., C.M., T.A.T., L.E.M., F.B., Á.M., M.N., A.B. and M.K.M. analysed and interpreted the data. L.S. and M.K.M. prepared the manuscript. All of the authors reviewed the manuscript.

**Competing interests** A.B. is a cofounder of Lion TCR, a biotechnology company that develops T cell receptors for the treatment of virus-related diseases and cancers. R.J.B. and D.M.A. are members of the Global T-cell Expert Consortium and have consulted for Oxford Immunotec outside the submitted work. The other authors declare no competing interests.

**Additional information**
**Correspondence and requests for materials** should be addressed to Leo Swadling or Mala K. Maini.

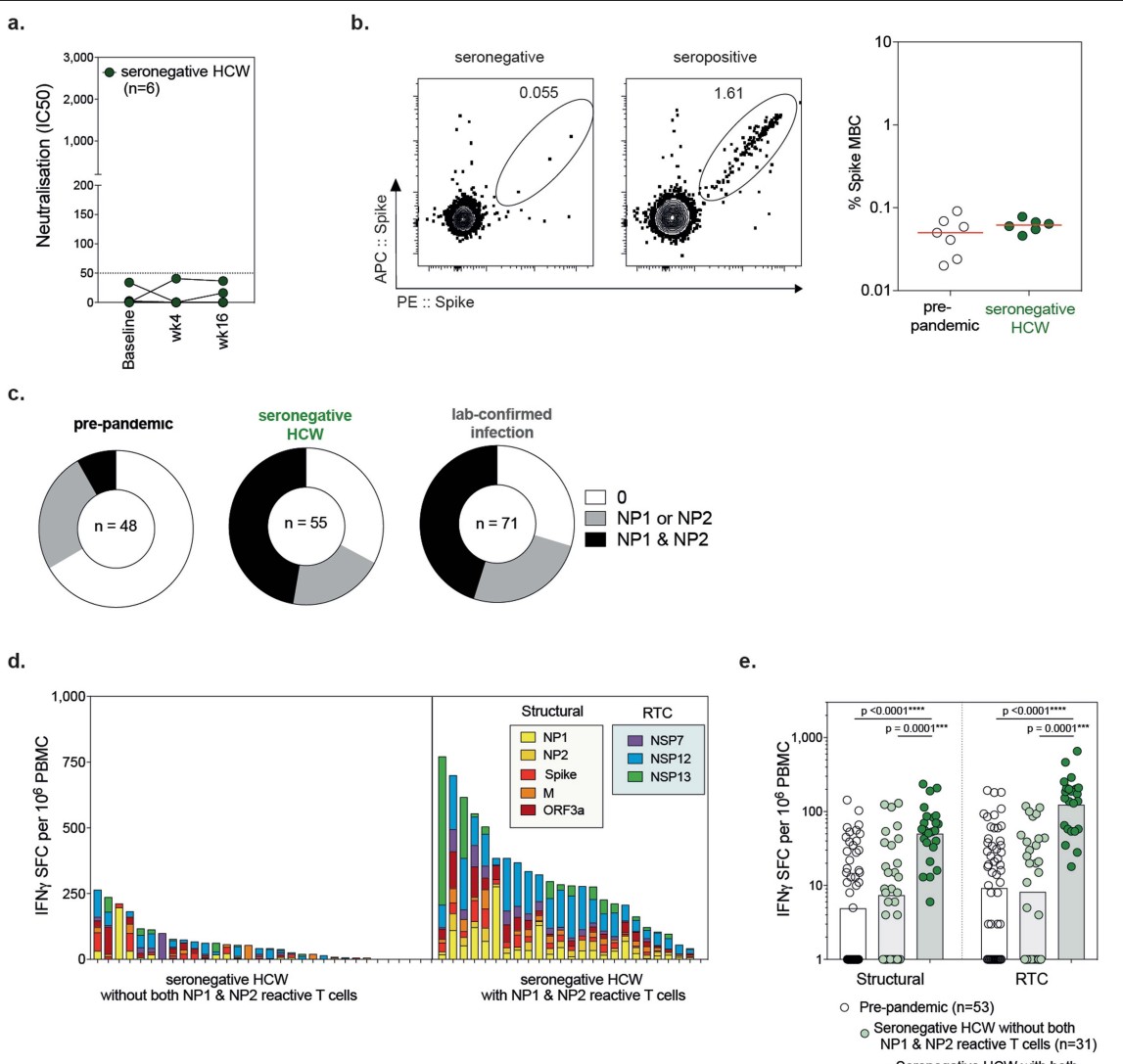

**Extended Data Fig. 1 | SARS-CoV-2 immunity in seronegative HCW – authentic virus neutralization and T cell response in those with NP1+NP2 responses. a**, authentic virus neutralization (Wuhan Hu-1). **b**, Example plots of SARS-CoV-2 spike memory B cell (MBC) staining (gated on: lymphocytes/singlets/Live, CD3-CD14-CD19+/CD20+, excluding CD38$^{hi}$, IgD+ and CD21+CD27- fractions) and frequency of SARS-CoV-2 spike-specific MBC in pre-pandemic or SN-HCW (wk16; as a percentage of total MBC). Bars, median.

**c**, Proportion of cohorts with T cell responses to NP1 and/or NP2 subpools. **d**, Magnitude of T cell response coloured by viral protein and **e**, summed response to RTC and structural regions in SN-HCW with T cells reactive against both NP1 and NP2 and against one of or neither NP1 or NP2 pools at wk16. Kruskal-Wallis with Dunn's correction. Bars, geomean. **a**-**e**, COVIDsortium HCW cohort. NP, nucleoprotein; RTC, replication-transcription complex.

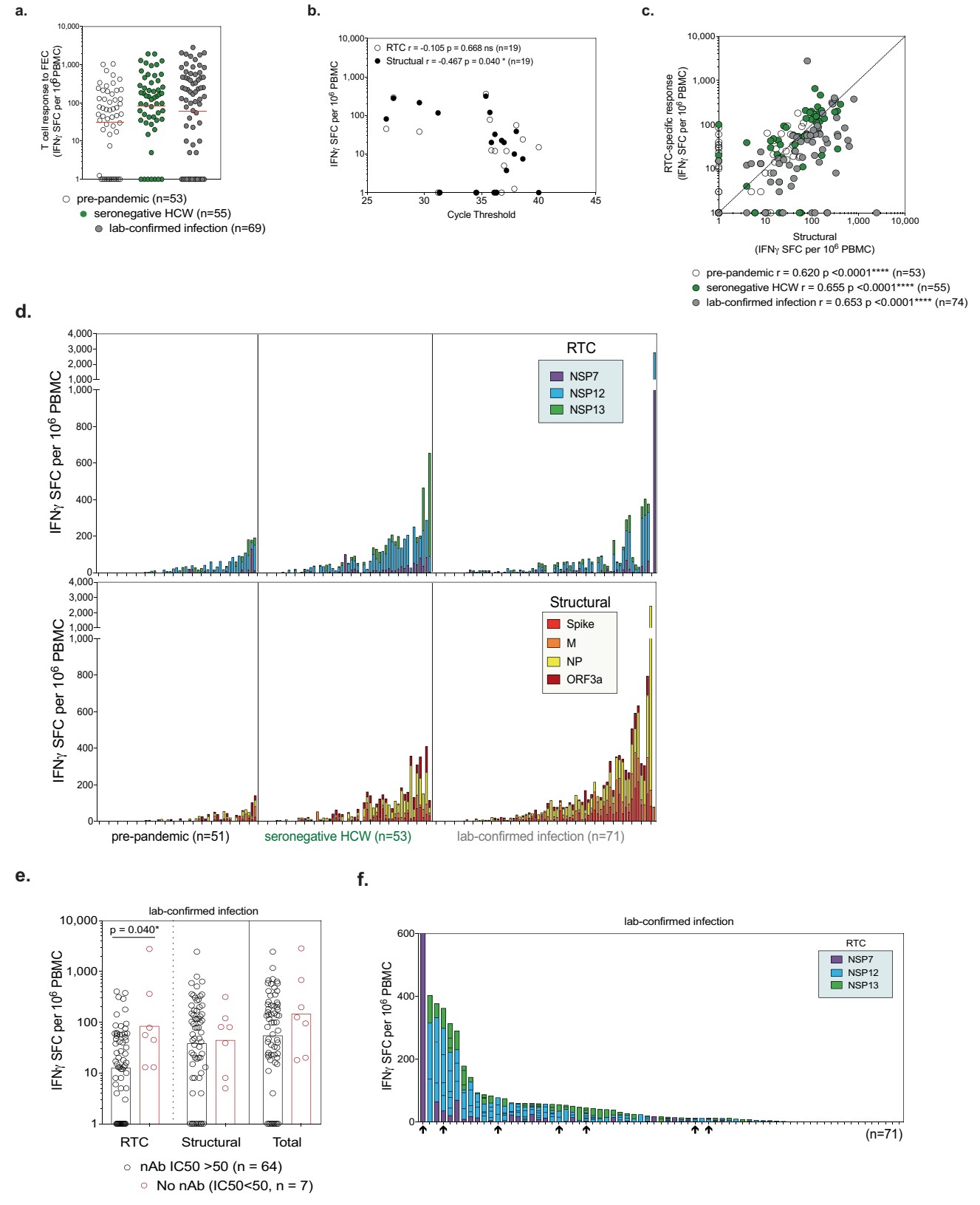

**Extended Data Fig. 2 | T cell responses to RTC and structural regions of SARS-CoV-2 by cohort. a**, T cell response to Flu, EBV and CMV (FEC) MHC class I restricted peptide pool. **b**, E gene RT-PCR cycle threshold value vs. magnitude of T cell response to RTC or structural proteins in HCW with laboratory-confirmed infection. **c**, Magnitude of T cell response to RTC vs. structural regions. **d**, Magnitude of T cell response to RTC (top) and structural regions (bottom) coloured by specificity. **e**, Magnitude of T cell response in laboratory-confirmed infection group in HCW with or without detectable neutralizing antibodies at wk16. **f**, T cell response to RTC coloured by protein in laboratory-confirmed infection group ordered by magnitude. HCW lacking neutralizing antibodies highlighted by arrows below. **a-f**, IFNγ ELISpot wk16. **a**, Red lines, geomean. **e**, Bars, geomean. **b-c** Spearman r. **a,d** Kruskal-Wallis ANOVA with Dunn's correction. **a-f**, COVIDsortium HCW cohort.

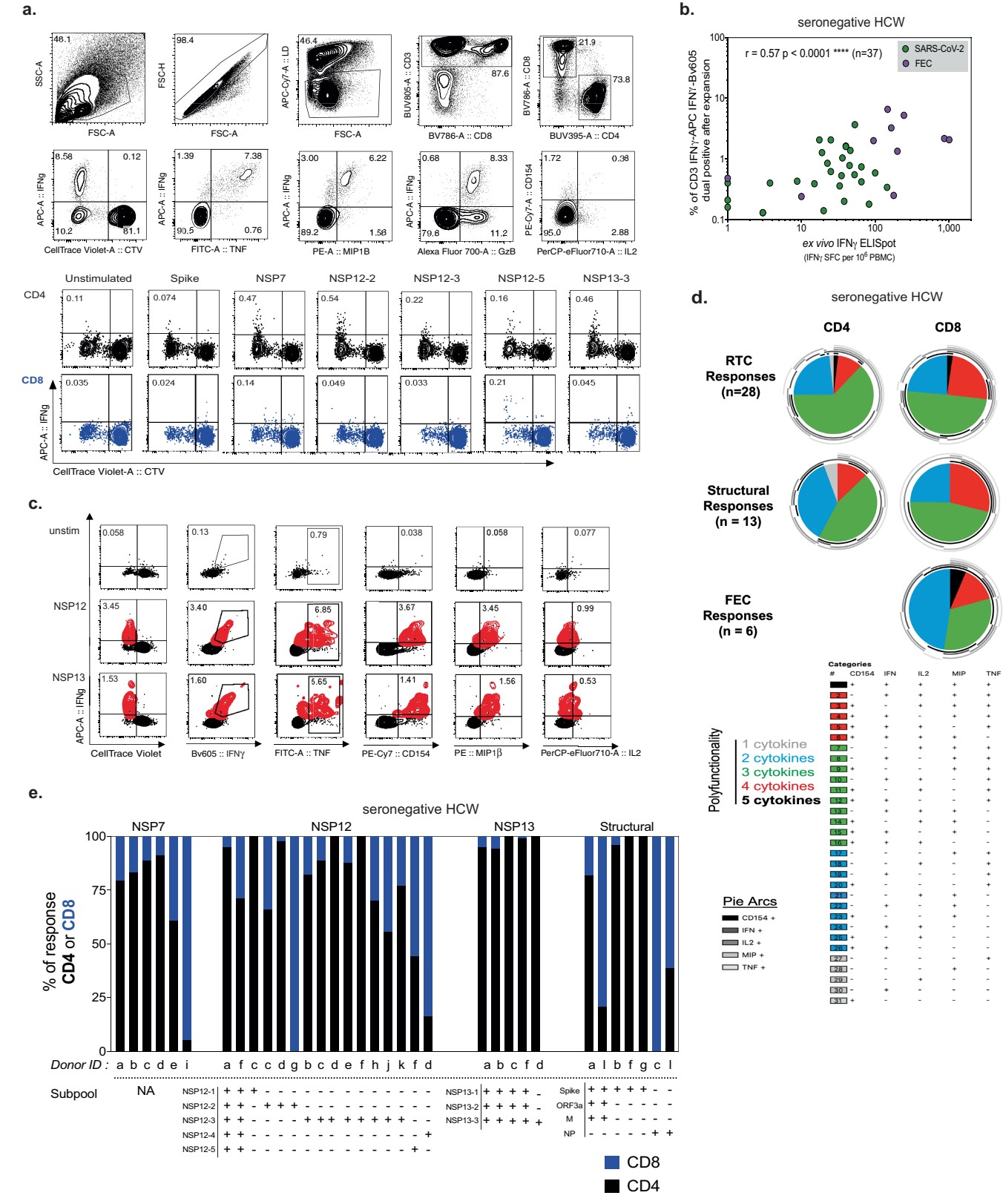

**Extended Data Fig. 3** | See next page for caption.

**Extended Data Fig. 3 | Functional and proliferative SARS-CoV-2 specific T cells in seronegative HCWs. a**, (Upper) Example gating of CTV stained PBMC after 10-day peptide stimulation: Lymphocytes (SSC-A vs. FSC-A), single cells (FSC-H vs. FSC-A), Live cells (fixable live/dead-), CD3+, CD4+ or CD8+. Second row: Gated on CD8+ showing cytokine/intracellular protein combinations. Response to immunodominant MHC class I-restricted peptide pool against Flu, EBV, CMV (FEC) in SN-HCW. (Lower) example CTV and IFNγ staining in a SN-HCW (gated on CD4+ [black] or CD8+ [blue] T cells, percentage CTV$^{lo}$IFNγ$^{+}$ shown). **b**, Correlation between the magnitude of T cells responses to SARS-CoV-2 pools or FEC after 10-day in vitro expansion (% dual staining for two anti-human IFNγ mAb clones, responses <0.1% of CD3 post-expansion excluded) and ex vivo IFNγ ELISpot in SN-HCWs. Spearman r. **c**, Example plots of dual cytokine or activation marker staining of SARS-CoV-2-specific T cells in an SN-HCW after 10-day expansion with peptide pools (proliferating T cells become CTV$^{lo}$ as they divide and dilute out marker). SARS-CoV-2-specific T cells highlighted in red (CD4+ CTV$^{lo}$IFNγ$^{+}$). Percentage of CD4+ shown. **d**, polyfunctionality of CD4+ and CD8+ T cells targeting the RTC or structural regions of SARS-CoV-2 or FEC peptide pool (proportion of cytokine producing T cells that co-producing a given number of cytokines after 10-day peptide stimulation). Pie base, mean. Pie arcs show proportion of cells producing a given cytokine. **e**, Proportion of SARS-CoV-2-specific T cells (CTV$^{lo}$IFNγ+) that are CD4+ or CD8+ after 10-day expansion (the protein specificity is listed above, donor ID (**a-l**, corresponding to raw data in Extended Data Table 2) and peptide subpools used for stimulation listed below). **a-e**, SN-HCW at wk16 COVIDsortium HCW cohort.

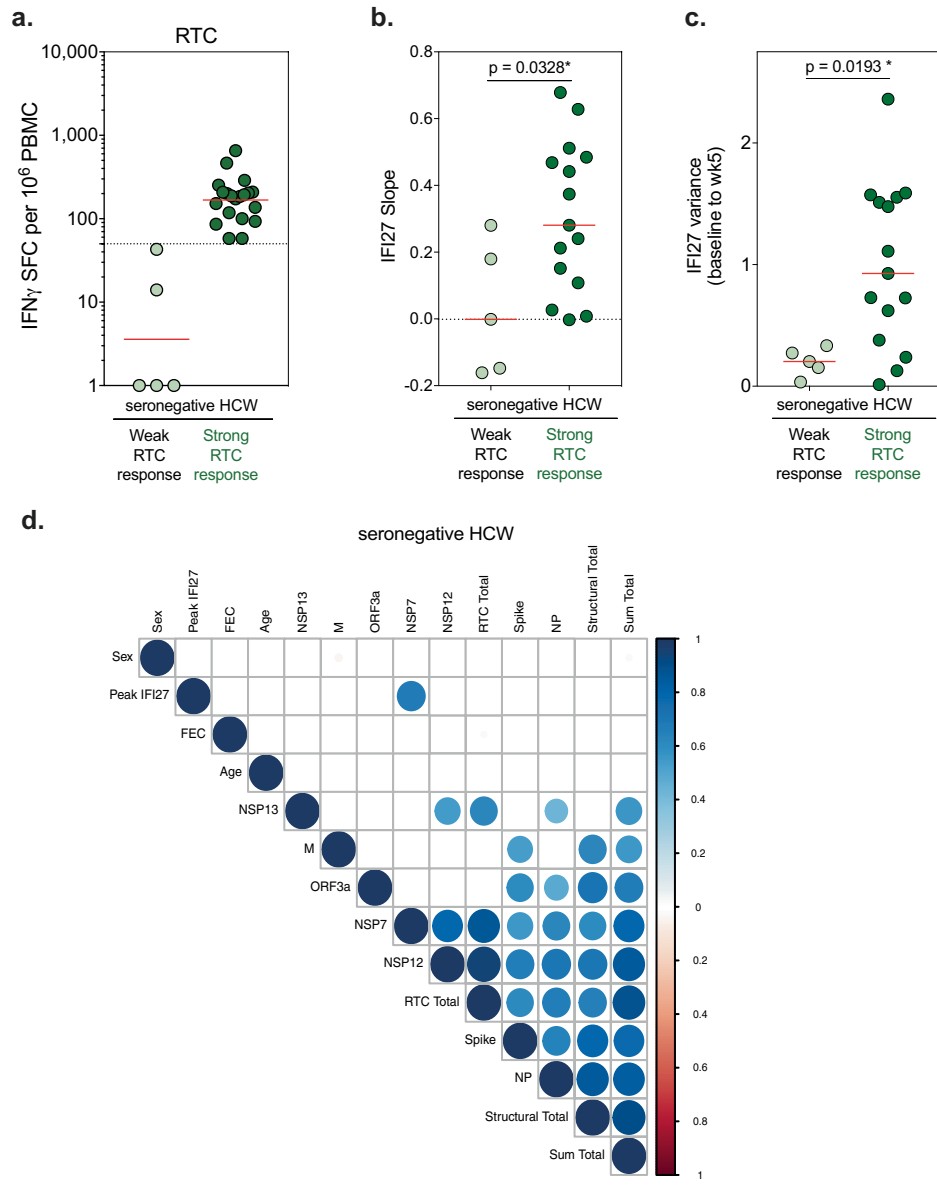

**Extended Data Fig. 4 | Slope and variance of IFI27 signal in seronegative HCWs. a**, Subsetting SN-HCW group into those with weak (n=5, <50 SFCs per 10[6] PBMCs) or strong (n=20, >50 SFCs per 10[6] PBMCs) RTC-specific T cell responses at wk16. **b**, Slope and **c**, variance of IFI27 signal (wk0-5) in SN-HCW with weak (n=5) or strong (n=15) RTC-specific T cell responses at wk16.

**d**, Correlation matrix of variables for SN-HCW (colour and size of dots represent spearman's r, only correlations p<0.05 shown; peak IFI27 signal from wk0-5, T cell responses at wk16 to proteins, regions [RTC or structural], or total SARS-CoV-2 response). **b**,**c**, Mann-Whitney test, Red lines at median. **a-d**, COVIDsortium HCW cohort.

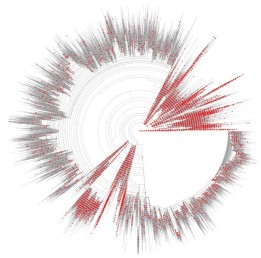

a.

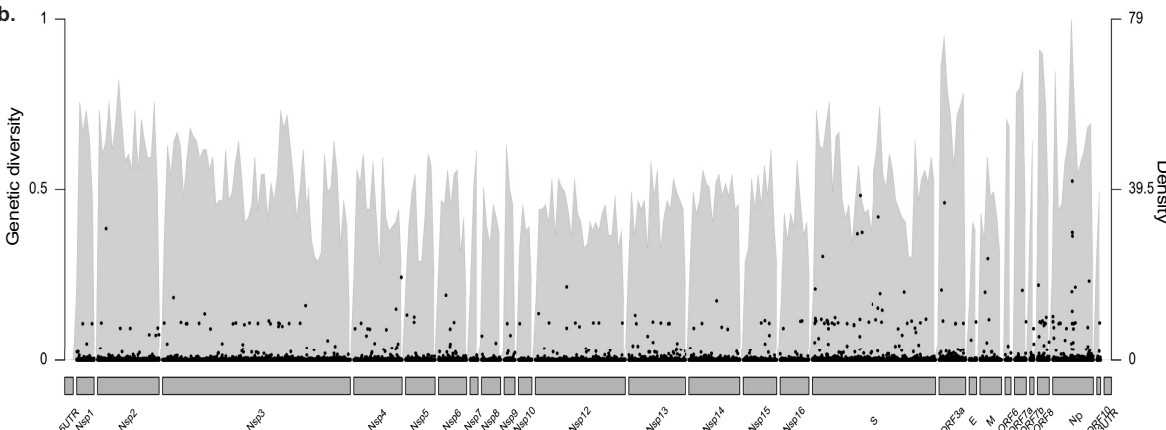

b.

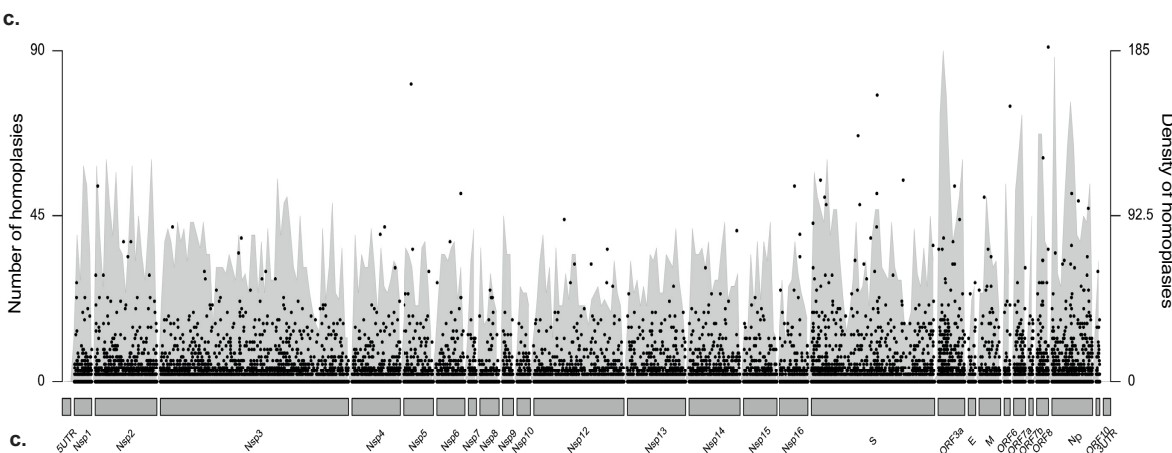

c.

c.

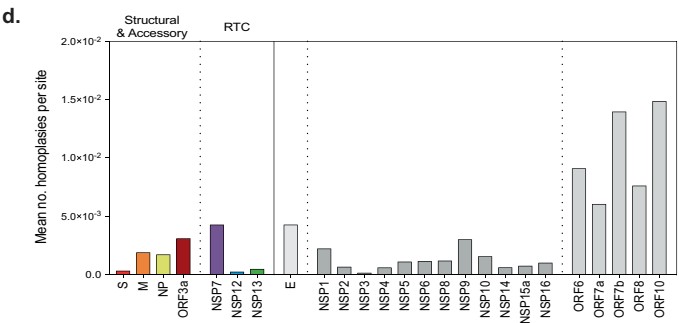

d.

**Extended Data Fig. 5 | Diversity along SARS-CoV-2 genome. a**, Radial phylogeny of SARS-CoV-2 sequence diversity (611,893 genomes) with the 13,785 accessions subsampled for diversity analysis shown in red. **b**, Genetic diversity (Nei's genetic diversity index) at individual nucleotides along the SARS-CoV-2 genomes, together with the density of polymorphic nucleotides over an 100-nucleotide sliding window shown in grey shading (right y-axis) and

**c**, Homoplasies (recurrent mutational emergences) at individual nucleotides, together with the density of the number of homoplasies recorded over an 100-nucleotide sliding window shown in grey shading (right y-axis). **d**, Mean number of homoplasies across a given protein. Viral proteins not assayed for T cell responses are shown in grey.

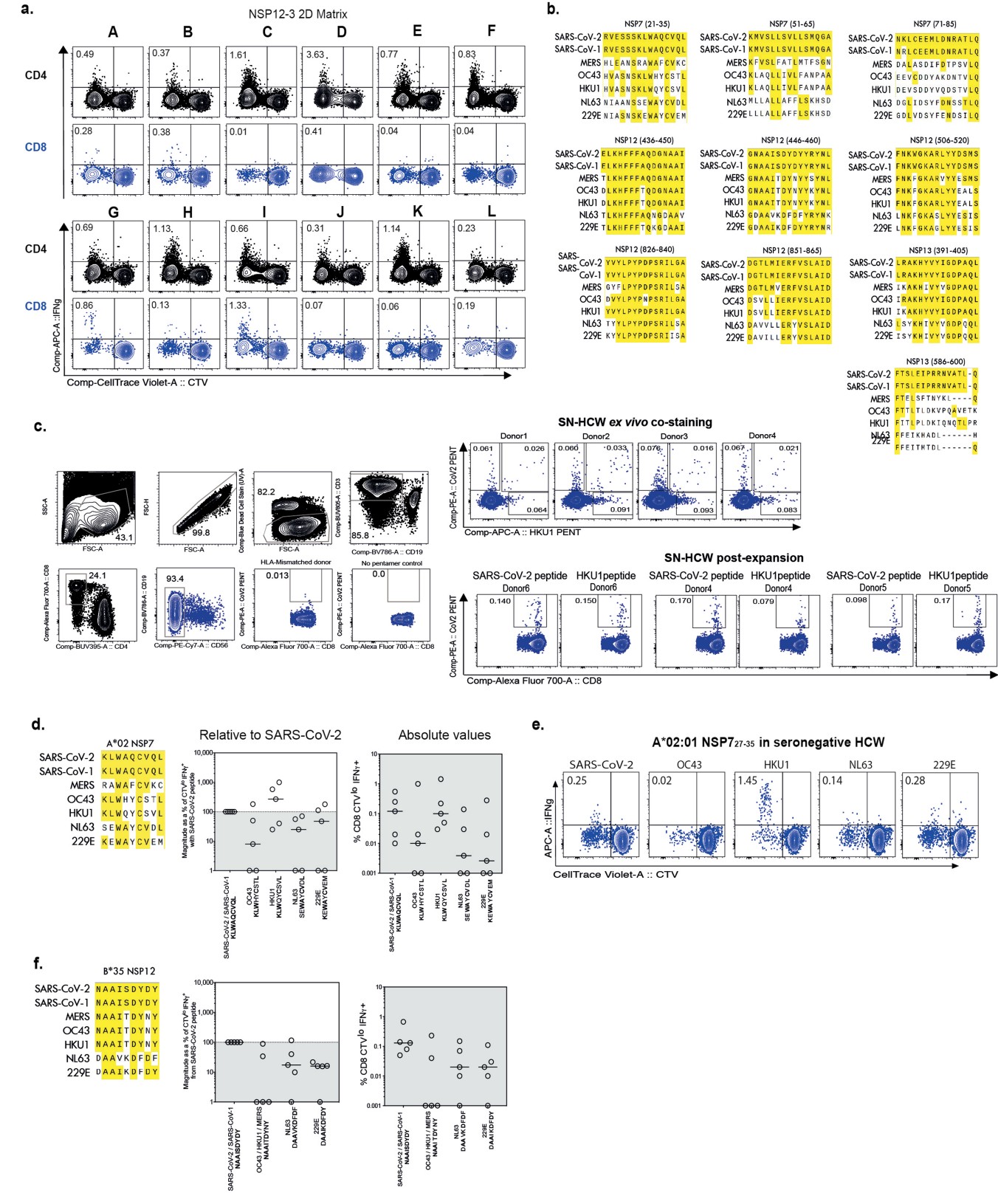

**Extended Data Fig 6** | See next page for caption.

**Extended Data Fig 6 | Cross-reactive coronavirus-specific T cells in seronegative HCWs. a**, Example 2D-mapping matrix after 10-day expansion with NSP12-3 peptide pool in an SN-HCW (antigen-specific, CTV$^{lo}$IFNγ+; percentage of CD4+ or CD8+ shown). **b**, Alignment of *Coronaviridae* consensus sequences at immunogenic 15mers peptides (Extended Data Table 4). Conserved amino acids in yellow. **c**, (left) Example gating (lymphocytes (SSC-A,FSC-A)/singlets(FSC-A,FSC-H)/Live(Live-dead$^-$)/CD3+CD19-/CD8+CD4-/CD56-; example of staining in HLA-mismatched donor and fluorescence minus one for pentamer shown) and (right above) pentamer stains of PBMC from SN-HCW at wk16-26 ex vivo (co-staining of pentamers loaded with SARS-CoV-2 peptide KLWAQCVQL and HKU1 peptide KLWQYCSVL) and (right below) after 10-day expansion with SARS-CoV-2 peptide or HKU1 peptide (stained with SARS-CoV-2 peptide loaded pentamer). Percentage of CD8+ shown.

**d**, Alignment of *Coronaviridae* sequences at HLA-A*02-restricted epitope in NSP7 (left) and magnitude of CD8+ T cell response (CTV$^{lo}$IFNγ+) after 10-day expansion with HCoV variant sequence peptides as a percentage of response with SARS-CoV-2 sequence peptide (middle) or absolute percentage of total CD8+ (right). **e**, Example plot of CTV vs. IFNγ after 10-day expansion with SARS-CoV-2 or HCoV sequence 9-mer peptides (gated on lymphocytes/singlets/live cells/CD3+/CD56-CD4-/CD8+). **f**, Alignment of *Coronaviridae* sequences at B*035-restricted epitope in NSP12 (left), magnitude of CD8+ T cell response (CTV$^{lo}$IFNγ+) after 10-day expansion with HCoV variant sequence peptides as a percentage of response with SARS-CoV-2 sequence peptide (middle) or absolute percentage of total CD8+ (right). **d,f**, Conserved amino acids in yellow. **d-f**, SN-HCW wk16. **a, c-f**, COVIDsortium HCW cohort. **d-f**, SN-HCW wk16. **d,f**, Lines, median.

## β - coronavirus

### OC43 Spike

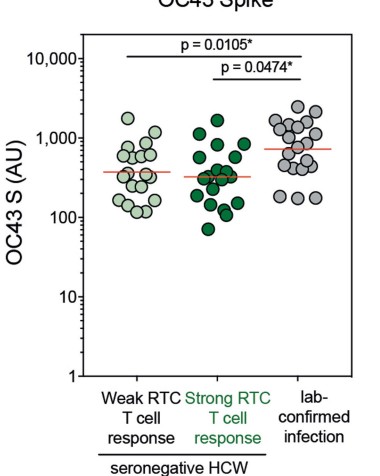

### HKU1 Spike

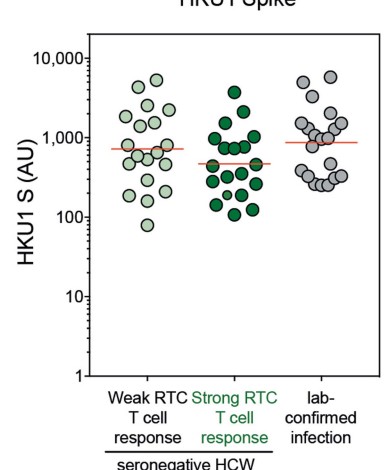

## α - coronavirus

### 229E Spike

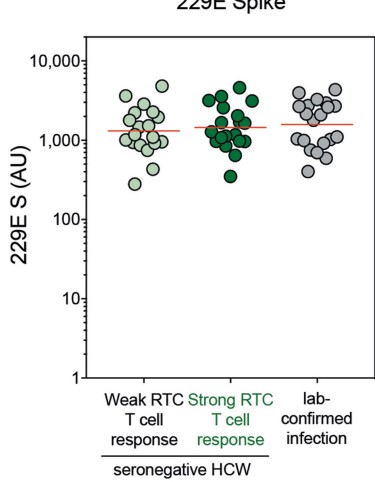

### NL63 Spike

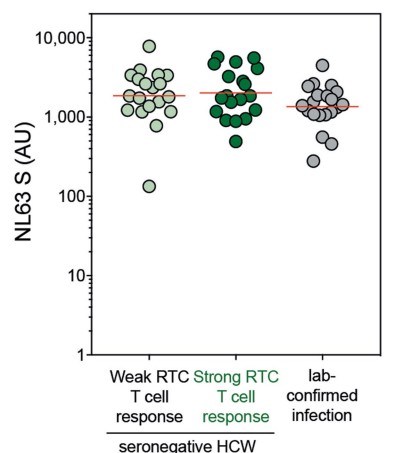

**Extended Data Fig. 7 | Anti-spike IgG to human endemic coronaviruses.** Anti-spike IgG titres were measured post-infection (wk8, time of peak SARS-CoV-2 S1 IgG seropositivity in COVIDsortium HCW cohort) in HCW with laboratory-confirmed infection (n=20), and post-exposure (wk8) in SN-HCW with weak (<50 SFCs per $10^6$ PBMCs, n=19) or strong RTC-specific T cell response at wk16 (n=19, >50 SFCs per $10^6$ PBMCs). Red lines, geomean.

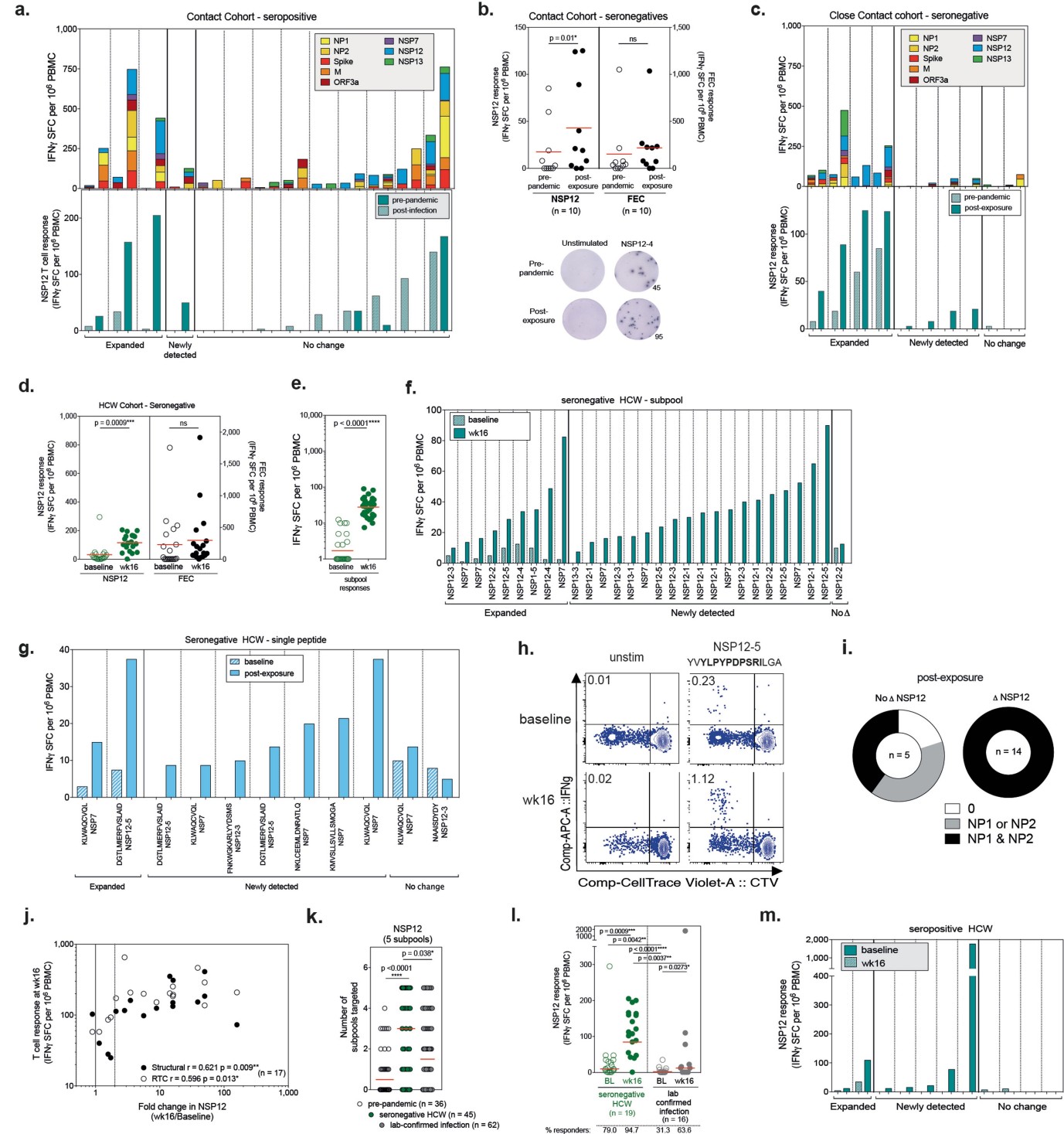

**Extended Data Fig. 8** | See next page for caption.

**Extended Data Fig. 8 | In vivo expansion of pre-existing SARS-CoV-2-reactive T cells post-infection or post-exposure. a**, Change in magnitude of T cell response between pre-pandemic and post-infection (upper panel: all proteins, lower panel: NSP12) in seropositive close contacts of cases. **b**, Summary data for paired pre-pandemic and post-exposure NSP12 and Flu/EBV/CMV (FEC) responses in seronegative close contacts of infections. Below; example ELISpot well images from a seronegative close contact (NSP12-4: pre-pandemic 45 and post-exposure 95 SFCs per $10^6$ PBMCs). **c**, Change in magnitude of T cell response between pre-pandemic and post-exposure samples (upper panel: all proteins, lower panel: NSP12) from seronegative close contacts of cases. **d**, Summary data for NSP12 and FEC responses in SN-HCW (sub-group with the top RTC response at wk16, n=19, Extended Data Fig. 4a). **e**, Summary data and **f**, change in magnitude of T cell responses for individual HCW to NSP7 (15 peptide pool) or a single subpool from NSP12 and NSP13 between baseline and post-exposure in SN-HCW (wk16-26, 29 responses from 13 SN-HCW). **g**, Change in magnitude of T cell response to individual 9-15mer peptides pre- and post-exposure in SN-HCW (wk16-26, 11 responses from 9 SN-HCW). **h**, Example plots of CTV^lo IFNγ^+ SARS-CoV-2-specific T cells after 10day expansion of PBMC from baseline and wk16 with peptide #166 (YVYLPYPDPSRILGA) or unstimulated in an HLA-B*51+ SN-HCW (gated on CD8+, percentage of CD8+ shown, gating strategy Extended Data Fig. 3a). **i**, Proportion of SN-HCW with NP1 + NP2-reactive T cells grouped by those with and without newly detected or expanded NSP12 responses at wk16, Fig. 4b. **j**, Correlation between the fold-change in NSP12 between recruitment and wk16 and total response to RTC or structural proteins at wk16 in SN-HCWs. Dotted line at 2-fold increase. **k**, The breadth of the NSP12-specific T cell response (number of subpools recognized, pre-pandemic or wk16). **l** Change in magnitude of the T cell response to NSP12 between baseline (open circles) and wk16 (closed circles) in SN-HCW and HCW with laboratory-confirmed infection. Percentage of responders shown below. **m**, Change in magnitude of NSP12-specific T cell response between pre-pandemic and post-infection in HCW with laboratory-confirmed infection. **a,c,f-g,m**, Expanded, >2-fold increase or >35 SFCs per $10^6$ PBMCs increase. **a**, Red line mean, **d-e,l**, Red line/bars, geomean. **k**, red line, median. **b,d,e** Wilcoxon test. **l**, Mann-Whitney (unpaired) and Wilcoxon (paired) tests. **k**, Kruskal-Wallis with Dunn's correction. **j**, Spearman r. **a-c**, Contact cohort, Extended Data Table 5. **d-m**, COVIDsortium cohort Extended Data Table 1.

# Extended Data Table 1 | Cohort Demographics

| | COVIDsortium HCW cohort | | | Contact Cohort | | Pre-pandemic cohorts | |
|---|---|---|---|---|---|---|---|
| | Total Cohort | Lab-confirmed infection | Seronegatives | Seropositive | Seronegatives | London | Singapore |
| Number of subjects | 731 | 76 | 58 | 13 | 10 | 53 | 12/16* |
| Mean age (range) | 38.1 (18-71) | 41.7 (25-62) | 37.9 (21-62) | 24.9 (20-65) | 26.4 (20-32) | 26.5 (20-59) | 40 (30-59) |
| **Sex:** | | | | | | | |
| Female, n (%) | 486 (66.76) | 50 (65.79) | 36 (62.07) | 9 (69.2) | 5 (50) | 35 (66.04) | 4 (33.33) |
| Male, n (%) | 242 (33.34) | 26 (34.21) | 22 (37.93) | 4 (30.8) | 5 (50) | 18 (33.96) | 8 (66.67) |
| **Ethnicity:** | | | | | | | |
| White, n (%) | 479 (65.80) | 55 (72.37) | 41 (70.68) | 8 (61.5) | 9 (90) | 39 (73.58) | 2 (16.67) |
| non-White, n (%) | 252 (34.62) | 21 (27.63) | 17 (29.31) | 5 (38.5) | 1 (10) | 14 (26.42) | 10 (83.33) |
| **Recent Travel (pre-March 2020):** | | | | | | | |
| Yes | 303 (41.51) | 33 (43.42) | 26 (44.83) | | | | |
| **COVID-19 Patient Exposure:** | | | | | | | |
| Yes | 315 (43.09) | 33 (43.42) | 17 (29.31) | | | | |
| **COVID-19 Colleague Exposure:** | | | | | | | |
| Yes | 218 (29.86) | 19 (25.00) | 10 (17.24) | | | | |
| **COVID-19 Household Exposure:** | | | | | | | |
| Yes | 8 (1.09) | 5 (6.58) | 0 (0) | | | | |
| **Role:** | | | | | | | |
| Laboratory | 12 (1.64) | 2 (2.63) | 2 (3.45) | | | | |
| Nurse | 231 (31.60) | 25 (32.89) | 22 (37.93) | | | | |
| Doctor | 150 (20.52) | 20 (26.32) | 12 (20.69) | | | | |
| Administration | 32 (4.38) | 3 (3.95) | 2 (3.45) | | | | |
| Allied Healthcare Professional | 185 (25.31) | 15 (19.74) | 14 (24.14) | | | | |
| Health Care Assistant | 43 (5.88) | 2 (2.63) | 1 (1.72) | | | | |
| Other | 78 (10.67) | 9 (11.85) | 5 (8.62) | | | | |
| **Location:** | | | | | | | |
| Laboratory | 42 (5.75) | 3 (3.95) | 6 (10.34) | | | | |
| Cardiac | 108 (14.77) | 18 (23.68) | 12 (20.69) | | | | |
| ICU | 126 (17.24) | 8 (10.53) | 7 (12.07) | | | | |
| Other medical | 58 (7.93) | 10 (13.16) | 6 (10.34) | | | | |
| Anaesthesia | 5 (0.68) | 1 (1.32) | 1 (1.72) | | | | |
| A&E | 24 (3.28) | 2 (2.63) | 0 (0) | | | | |
| Other | 361 (49.39) | 30 (39.47) | 26 (44.83) | | | | |
| Unspecified | 7 (0.96) | 4 (5.26) | 0 (0) | | | | |
| **Early PPE usage:** | | | | | | | |
| Yes | 584 (80.11) | 62 (81.58) | 39 (67.24) | | | | |
| **Aerosol creating procedures:** | | | | | | | |
| Yes | 189 (25.89) | 15 (19.74) | 13 (22.42) | | | | |

| | COVIDsortium SN-HCW divided by post-exposure T cell response | |
|---|---|---|
| | Seronegatives Weak RTC T cell response (<50 SFU/10$^6$ PBMC) | Seronegatives Strong RTC T cell response (>50 SFU/10$^6$ PBMC) |
| Number of subjects | 20 | 20 |
| Mean age (range) | 36.5 (21-58) | 43.6 (27-60) |
| **Sex:** | | |
| Female, n (%) | 13 (65.00) | 14 (70.00) |
| Male, n (%) | 7 (35.00) | 6 (30.00) |
| **Ethnicity:** | | |
| White, n (%) | 14 (70.00) | 15 (75.00) |
| non-White, n (%) | 6 (30.00) | 5 (25.00) |
| **Recent Travel (pre-March 2020):** | | |
| Yes | 8 (40.00) | 10 (50.00) |
| **COVID-19 Patient Exposure:** | | |
| Yes | 5 (25.00) | 6 (30.00) |
| **COVID-19 Colleague Exposure:** | | |
| Yes | 4 (20.00) | 6 (30.00) |
| **COVID-19 Household Exposure:** | | |
| Yes | 0 (0) | 0 (0) |
| **Role:** | | |
| Laboratory | 0 (0) | 2 (10.00) |
| Nurse | 7 (35.00) | 8 (40.00) |
| Doctor | 3 (15.00) | 5 (25.00) |
| Administration | 0 (0) | 1 (5.00) |
| Allied Healthcare Professional | 8 (40.00) | 3 (15.00) |
| Health Care Assistant | 0 (0) | 0 (0) |
| Other | 2 (10.00) | 1 (5.00) |
| **Location:** | | |
| Laboratory | 2 (10.00) | 3 (15.00) |
| Cardiac | 5 (25.00) | 3 (15.00) |
| ICU | 1 (5.00) | 4 (20.00) |
| Other medical | 3 (15.00) | 3 (15.00) |
| Anaesthesia | 1 (5.00) | 0 (0) |
| A&E | 0 (0) | 0 (0) |
| Other | 2 (10.00) | 7 (35.00) |
| Unspecified | 0 (0) | 0 |
| **Early PPE usage:** | | |
| Yes | 12 (60.00) | 14 (70.00) |
| **Aerosol creating procedures:** | | |
| Yes | 5 (25.00) | 4 (20.00) |

A&E, accident and emergency department; ICU, intensive care unit; PPE, personal protective equipment; RTC, replication-transcription complex. * demographics for 4 pre-pandemic samples unknown.

## Extended Data Table 2 | T cell proliferation assay in seronegative HCW

| Donor | Week | Antigen | Subpool | Minipool | Peptide | CD4 % CTV-IFNγ+ | CD8 % CTV-IFNγ+ | % CD4 |
|---|---|---|---|---|---|---|---|---|
| a | 16 | Spike/M/ORF3a | - | - | - | 0.11% | 0.02% | 81.78 |
| a | 16 | FEC | - | - | - | 0.00% | 10.99% | 0.00 |
| a | 16 | NSP7 | - | - | - | 0.81% | 0.19% | 80.96 |
| a | 16 | NSP12 | - | - | - | 0.23% | 0.01% | 95.00 |
| a | 16 | NSP13 | - | - | - | 1.04% | 0.06% | 94.79 |
| a | 16 | NSP7 | - | A | - | 0.36% | 0.01% | 97.81 |
| a | 16 | NSP7 | - | B | - | 0.49% | 0.01% | 98.39 |
| a | 16 | NSP7 | - | C | - | 1.23% | 0.00% | 100.00 |
| a | 16 | NSP7 | - | D | - | 0.53% | 0.00% | 100.00 |
| a | 16 | NSP7 | - | E | - | 0.68% | 0.00% | 99.66 |
| a | 16 | NSP7 | - | F | - | 0.49% | 0.01% | 98.39 |
| a | 16 | NSP7 | - | G | - | 1.12% | 0.02% | 98.50 |
| a | 16 | NSP7 | - | H | - | 1.57% | 0.01% | 99.43 |
| a | 16 | NSP7 | - | C&G | 11 | 0.30% | 0.00% | 100.00 |
| a | 16 | NSP7 | - | C&H | 15 | 0.11% | 0.01% | 91.67 |
| a | 16 | NSP7 | - | A&F | 5 | 0.16% | 0.06% | 72.41 |
| a | 16 | NSP12 | 3 | B&I | 88 | 0.07% | 0.00% | 100.00 |
| a | 16 | NSP7 | - | - | 5a | - | 0.02% | - |
| a | 16 | NSP7 | - | - | 5b | - | 0.00% | - |
| a | 16 | NSP7 | - | - | 5c | - | 0.05% | - |
| a | 16 | NSP7 | - | - | 5d | - | 0.00% | - |
| a | 16 | NSP7 | - | - | 5e | - | 0.00% | - |
| | | | | | | | | |
| b | 16 | NSP7 | - | - | - | 1.14% | 0.23% | 83.06 |
| b | 16 | NSP12 | 3 | - | - | 2.07% | 0.45% | 82.06 |
| b | 16 | NSP13 | - | - | - | 0.58% | 0.04% | 94.42 |
| b | 16 | Spike | - | - | - | 0.20% | 0.01% | 95.92 |
| b | 16 | NSP12 | 3 | A | - | 1.68% | 0.31% | 84.32 |
| b | 16 | NSP12 | 3 | B | - | 0.38% | 0.13% | 74.15 |
| b | 16 | NSP12 | 3 | C | - | 0.84% | 0.23% | 78.32 |
| b | 16 | NSP12 | 3 | D | - | 1.12% | 0.26% | 81.01 |
| b | 16 | NSP12 | 3 | E | - | 0.77% | 0.34% | 69.21 |
| b | 16 | NSP12 | 3 | F | - | 1.36% | 0.40% | 77.16 |
| b | 16 | NSP12 | 3 | G | - | 1.26% | 0.20% | 86.15 |
| b | 16 | NSP12 | 3 | H | - | 0.67% | 0.07% | 90.24 |
| b | 16 | NSP12 | 3 | I | - | 1.81% | 0.23% | 88.62 |
| b | 16 | NSP12 | 3 | J | - | 0.37% | 0.29% | 55.85 |
| b | 16 | NSP12 | 3 | K | - | 0.60% | 0.29% | 67.23 |
| b | 16 | NSP12 | 3 | L | - | 0.65% | 0.44% | 59.50 |
| | | | | | | | | |
| c | 16 | FEC | - | - | - | 0.07% | 8.56% | 0.81 |
| c | 16 | NP | 1&2 | - | - | 0.00% | 0.19% | 0.00 |
| c | 16 | NSP7 | - | - | - | 0.61% | 0.08% | 88.72 |
| c | 16 | NSP13 | - | - | - | 0.30% | 0.00% | 100.00 |
| c | 16 | NSP12 | 1 | - | - | 0.81% | 0.00% | 100.00 |
| c | 16 | NSP12 | 2 | - | - | 0.11% | 0.06% | 65.96 |
| c | 16 | NSP12 | 3 | - | - | 0.62% | 0.08% | 88.50 |
| c | 16 | NSP12 | 3 | - | - | 0.03% | 0.14% | 19.54 |
| c | 16 | NSP12 | 3 | A | - | 0.36% | 0.26% | 57.98 |
| c | 16 | NSP12 | 3 | B | - | 0.24% | 0.36% | 39.92 |
| c | 16 | NSP12 | 3 | C | - | 1.48% | 0.00% | 100.00 |
| c | 16 | NSP12 | 3 | D | - | 3.50% | 0.39% | 89.96 |
| c | 16 | NSP12 | 3 | E | - | 0.64% | 0.02% | 97.63 |
| c | 16 | NSP12 | 3 | F | - | 0.70% | 0.02% | 97.02 |
| c | 16 | NSP12 | 3 | G | - | 0.56% | 0.84% | 39.96 |
| c | 16 | NSP12 | 3 | H | - | 1.00% | 0.11% | 90.05 |
| c | 16 | NSP12 | 3 | I | - | 0.53% | 1.31% | 28.78 |
| c | 16 | NSP12 | 3 | J | - | 0.18% | 0.05% | 78.38 |
| c | 16 | NSP12 | 3 | K | - | 1.01% | 0.04% | 96.23 |
| c | 16 | NSP12 | 3 | L | - | 0.10% | 0.17% | 36.85 |
| c | 16 | NSP12 | 3 | B&G | 76 | 0.05% | 0.00% | 100.00 |
| c | 16 | NSP12 | 3 | D&H | 84 | 0.05% | 0.00% | 100.00 |
| c | 16 | NSP12 | 3 | B&I | 88 | 0.23% | 0.11% | 67.84 |
| c | 16 | NSP12 | 3 | D&K | 102 | 0.08% | 0.00% | 100.00 |
| c | 16 | NSP12 | 5 | E&J | 171 | 0.11% | 0.03% | 76.39 |
| c | 16 | NSP13 | 3 | D&I | 90a | - | 0.08% | - |
| c | 16 | NSP13 | 3 | D&I | 90b | - | 0.00% | - |
| c | 16 | NSP13 | 3 | D&I | 90c | - | 0.01% | - |
| c | 16 | NSP13 | 3 | D&I | 90d | - | 0.00% | - |
| | | | | | | | | |
| d | 16 | FEC | - | - | - | 0.06% | 17.90% | 0.33 |
| d | 16 | NSP7 | - | - | - | 0.34% | 0.03% | 90.90 |
| d | 16 | NSP12 | 2 | - | - | 0.41% | 0.01% | 97.61 |
| d | 16 | NSP12 | 3 | - | - | 0.09% | 0.00% | 100.00 |
| d | 16 | NSP12 | 5 | - | - | 0.03% | 0.16% | 16.13 |
| d | 16 | NSP13 | 3 | - | - | 0.33% | 0.00% | 100.00 |
| d | 16 | NSP7 | - | C&G | 11 | 0.00% | 0.03% | 0.00 |
| d | 16 | NSP7 | - | C&H | 15 | 0.24% | 0.04% | 86.02 |
| d | 16 | NSP12 | 3 | B&I | 88 | 0.00% | 0.00% | 0.00 |
| d | 16 | NSP12 | 3 | D&I | 90 | 0.17% | 0.00% | 97.84 |
| | | | | | | | | |
| e | 16 | NSP7 | - | A&F | 5 | 0.01% | 0.06% | 17.65 |
| e | 16 | NSP7 | - | - | - | 0.18% | 0.12% | 60.53 |
| e | 16 | NSP12 | 3 | I | - | 0.01% | 0.01% | 46.59 |
| e | 16 | NSP12 | 3 | - | - | 0.16% | 0.02% | 87.70 |
| e | 16 | NSP12 | 3 | B&I | 88 | 0.02% | 0.00% | 100.00 |
| e | 16 | NSP12 | 3 | D&I | 90 | 0.01% | 0.00% | 100.00 |
| e | 16 | NSP13 | 3 | D&I | 90a | - | 0.13% | - |
| e | 16 | NSP14 | 3 | D&I | 90b | - | 0.00% | - |
| e | 16 | NSP15 | 3 | D&I | 90c | - | 0.15% | - |
| e | 16 | NSP16 | 3 | D&I | 90d | - | 0.02% | - |

| Donor | Week | Antigen | Subpool | Minipool | Peptide | 4 % CTV-IFNγ+ | 8 % CTV-IFNγ+ | % CD4 |
|---|---|---|---|---|---|---|---|---|
| f | 29 | NSP12 | - | - | - | 1.47% | 0.60% | 71.04 |
| f | 29 | NSP12 | 5 | - | - | 0.21% | 0.27% | 43.98 |
| f | 29 | NSP13 | - | - | - | 3.40% | 0.04% | 98.84 |
| f | 29 | Spike | - | - | - | 0.10% | 0.00% | 100.00 |
| f | 29 | NSP12 | 3 | A | - | 0.16% | 0.41% | 28.32 |
| f | 29 | NSP12 | 3 | B | - | 0.18% | 0.10% | 64.54 |
| f | 29 | NSP12 | 3 | C | - | 0.08% | 0.12% | 40.59 |
| f | 29 | NSP12 | 3 | D | - | 0.27% | 0.25% | 52.11 |
| f | 29 | NSP12 | 3 | E | - | 0.28% | 0.30% | 48.45 |
| f | 29 | NSP12 | 3 | F | - | 0.09% | 0.28% | 24.73 |
| f | 29 | NSP12 | 3 | G | - | 0.32% | 0.00% | 100.00 |
| f | 29 | NSP12 | 3 | H | - | 0.17% | 0.15% | 53.42 |
| f | 29 | NSP12 | 3 | I | - | 0.21% | 1.71% | 11.03 |
| f | 29 | NSP12 | 3 | J | - | 0.16% | 0.06% | 72.97 |
| f | 29 | NSP12 | 3 | K | - | 0.27% | 0.18% | 60.18 |
| f | 29 | NSP12 | 3 | L | - | 0.10% | 0.37% | 21.61 |
| f | 29 | NSP12 | 3 | - | - | 1.61% | 0.00% | 100.00 |
| f | 29 | NSP12 | 3 | I | - | 0.13% | 0.03% | 81.99 |
| f | 29 | NSP12 | 3 | A&I | 87 | 0.00% | 0.00% | 100.00 |
| f | 29 | NSP12 | 3 | B&I | 88 | 0.10% | 0.02% | 81.30 |
| f | 29 | NSP12 | 3 | D&I | 90 | 0.17% | 0.08% | 68.27 |
| f | 29 | NSP12 | 3 | F&I | 92 | 0.00% | 0.00% | 0.00 |
| f | 29 | NSP7 | - | A&F | 5 | 0.78% | 0.00% | 100.00 |
| | | | | | | | | |
| g | 16 | Spike | - | - | - | 0.19% | 0.00% | 100.00 |
| g | 16 | NSP12 | 2 | - | - | 0.00% | 0.12% | 0.00 |
| | | | | | | | | |
| h | 16 | NSP12 | 5 | E&J | 171 | 0.17% | 0.01% | 94.44 |
| h | 16 | NSP12 | 3 | - | - | 0.25% | 0.11% | 69.83 |
| h | 16 | NSP13 | 3 | D&I | 90a | - | 0.14% | - |
| h | 16 | NSP13 | 3 | D&I | 90b | - | 0.00% | - |
| h | 16 | NSP13 | 3 | D&I | 90c | - | 0.00% | - |
| h | 16 | NSP13 | 3 | D&I | 90d | - | 0.03% | - |
| | | | | | | | | |
| i | 16 | NSP7 | - | - | - | 0.01% | 0.13% | 7.14 |
| i | 16 | NSP7 | - | A&F | 5 | 0 | 0.06% | 0.00 |
| i | 16 | NSP7 | - | - | 5a | - | 0.01% | - |
| i | 16 | NSP7 | - | - | 5b | - | 0.01% | - |
| i | 16 | NSP7 | - | - | 5c | - | 0.10% | - |
| i | 16 | NSP7 | - | - | 5d | - | 0.00% | - |
| i | 16 | NSP7 | - | - | 5e | - | 0.02% | - |
| | | | | | | | | |
| j | 16 | NSP12 | 3 | - | - | 0.59% | 0.48% | 55.35 |
| j | 16 | NSP12 | 3 | D&I | 90 | - | 0.62% | - |
| j | 16 | NSP12 | 3 | D&I | 90a | - | 0.67% | - |
| j | 16 | NSP12 | 3 | D&I | 90b | - | 0.23% | - |
| j | 16 | NSP12 | 3 | D&I | 90c | - | 0.07% | - |
| j | 16 | NSP12 | 3 | D&I | 90d | - | 0.11% | - |
| j | 16 | NSP12 | 4 | - | 148 | 0.14% | 0.10% | 58.33 |
| | | | | | | | | |
| k | 4 | NSP12 | 4 | - | - | 0.09 | 0.03 | 75.00 |
| | | | | | | | | |
| l | 16 | FEC | - | - | - | 0.76% | 4.48% | 14.50 |
| l | 16 | NP | 1&2 | - | - | 0.01% | 0.01% | 38.46 |
| l | 16 | ORF3a | - | - | - | 0.07% | 0.52% | 11.66 |
| l | 16 | Spike/M/ORF3 | - | - | - | 0.04% | 0.14% | 20.56 |
| l | 16 | NSP7 | - | - | 5a | - | 0.55% | - |
| l | 16 | NSP7 | - | - | 5b | - | 1.00% | - |
| l | 16 | NSP7 | - | - | 5c | - | 0.22% | - |
| l | 16 | NSP7 | - | - | 5d | - | 0.0039 | - |
| l | 16 | NSP7 | - | - | 5e | - | 0.0026 | - |
| | | | | | | | | |
| m | 16 | NSP12 | 4 | - | 148 | 0.10% | 0.02% | 83.02 |
| m | 16 | NSP13 | 3 | D&I | 90a | - | 0.05% | - |
| m | 16 | NSP13 | 3 | D&I | 90b | - | 0.04% | - |
| m | 16 | NSP13 | 3 | D&I | 90c | - | 0.02% | - |
| m | 16 | NSP13 | 3 | D&I | 90d | - | 0.01% | - |
| | | | | | | | | |
| n | 16 | FEC | - | - | - | 0.14% | 5.33% | 2.56 |
| | | | | | | | | |
| o | 16 | FEC | - | - | - | 5.00% | 12.73% | 28.20 |
| | | | | | | | | |
| p | 16 | NSP7 | - | - | 5a | - | 0.25% | - |
| p | 16 | NSP7 | - | - | 5b | - | 0.02% | - |
| p | 16 | NSP7 | - | - | 5c | - | 1.45% | - |
| p | 16 | NSP7 | - | - | 5d | - | 0.14% | - |
| p | 16 | NSP7 | - | - | 5e | - | 0.28% | - |
| p | 16 | NSP7 | - | - | 5a | - | 0.21% | - |
| p | 16 | NSP7 | - | - | 5b | - | 0.30% | - |
| p | 16 | NSP7 | - | - | 5c | - | 0.86% | - |
| p | 16 | NSP7 | - | - | 5d | - | 0.36% | - |
| p | 16 | NSP7 | - | - | 5e | - | 0.16% | - |
| p | 16 | NSP7 | - | - | 5a | - | 0.11% | - |
| p | 16 | NSP7 | - | - | 5b | - | 0.01% | - |
| p | 16 | NSP7 | - | - | 5c | - | 0.19% | - |
| p | 16 | NSP7 | - | - | 5d | - | 0.00% | - |
| | | | | | | | | |
| q | 16 | NSP7 | - | - | 5a | - | 0.12% | - |
| q | 16 | NSP7 | - | - | 5b | - | 0.00% | - |
| q | 16 | NSP7 | - | - | 5c | - | 0.03% | - |
| q | 16 | NSP7 | - | - | 5d | - | 0.03% | - |
| q | 16 | NSP7 | - | - | 5e | - | 0 | - |
| | | | | | | | | |
| r | 0 | NSP12 | 5 | F&I | 166 | - | 0.22% | - |
| r | 16 | NSP12 | 5 | F&I | 166 | - | 1.10% | - |

**Extended Data Table 3 | SARS-CoV-2 Neis genetic diversity and number of homoplasies per site per gene region**

| Neis genetic diversity per site, by region | | | |
|---|---|---|---|
| **Region** | **Mean** | **Median** | **SD** |
| Nsp3 | 1.57E-07 | 0 | 1.29E-06 |
| **Nsp12** | **2.69E-07** | **0** | **2.67E-06** |
| Nsp14 | 5.34E-07 | 0 | 4.38E-06 |
| Nsp2 | 5.54E-07 | 7.58E-08 | 5.49E-06 |
| Nsp13 | 5.83E-07 | 0 | 4.17E-06 |
| S | 5.98E-07 | 0 | 4.95E-06 |
| Nsp15 | 7.88E-07 | 0 | 6.60E-06 |
| Nsp4 | 7.92E-07 | 0 | 6.60E-06 |
| Nsp16 | 8.13E-07 | 0 | 7.30E-06 |
| Nsp8 | 8.50E-07 | 0 | 6.42E-06 |
| Nsp5 | 8.64E-07 | 0 | 7.95E-06 |
| Nsp10 | 1.21E-06 | 0 | 1.30E-05 |
| Nsp6 | 1.32E-06 | 0 | 1.00E-05 |
| Nsp7 | 1.80E-06 | 0 | 5.78E-06 |
| Nsp1 | 1.95E-06 | 2.69E-07 | 1.33E-05 |
| M | 2.20E-06 | 0 | 2.23E-05 |
| ORF3a | 3.03E-06 | 1.75E-07 | 2.34E-05 |
| NP | 3.30E-06 | 1.15E-07 | 2.25E-05 |
| Nsp9 | 3.32E-06 | 0 | 2.22E-05 |
| ORF6 | 3.92E-06 | 7.80E-07 | 1.96E-05 |
| E | 4.40E-06 | 0 | 3.76E-05 |
| ORF7a | 4.63E-06 | 4.02E-07 | 3.46E-05 |
| ORF8 | 1.25E-05 | 3.97E-07 | 5.71E-05 |
| ORF10 | 1.55E-05 | 1.25E-06 | 9.08E-05 |
| ORF7b | 1.59E-05 | 1.10E-06 | 7.32E-05 |

| No. Homoplasies per site, by region | | | |
|---|---|---|---|
| **Region** | **Mean** | **Median** | **SD** |
| Nsp3 | 0.000175932 | 0 | 0.000519326 |
| **Nsp12** | **0.000258192** | **0** | **0.000952715** |
| S | 0.000353033 | 0 | 0.001182173 |
| Nsp13 | 0.000502952 | 0 | 0.001535093 |
| Nsp4 | 0.000630667 | 0 | 0.002027798 |
| Nsp14 | 0.000635312 | 0 | 0.001918301 |
| Nsp2 | 0.000687067 | 0 | 0.001911003 |
| Nsp15 | 0.000774054 | 0 | 0.002450595 |
| Nsp16 | 0.001038492 | 0 | 0.004017116 |
| Nsp5 | 0.001132043 | 0 | 0.004471706 |
| Nsp6 | 0.001173207 | 0 | 0.003966735 |
| Nsp8 | 0.001218697 | 0 | 0.004520017 |
| Nsp10 | 0.001598721 | 0 | 0.004885641 |
| NP | 0.001747292 | 0 | 0.004215111 |
| M | 0.00192376 | 0 | 0.006486963 |
| Nsp1 | 0.002253086 | 0 | 0.005686097 |
| Nsp9 | 0.00304557 | 0 | 0.0080396 |
| ORF3a | 0.003140388 | 0 | 0.00685446 |
| Nsp7 | 0.004306382 | 0 | 0.011129816 |
| E | 0.004309018 | 0 | 0.01490037 |
| ORF7a | 0.006074296 | 0 | 0.012362245 |
| ORF8 | 0.007651766 | 0 | 0.021193027 |
| ORF6 | 0.009134004 | 0 | 0.034665426 |
| ORF7b | 0.014003673 | 0 | 0.030034154 |
| ORF10 | 0.014902476 | 0 | 0.036781834 |

| Neis genetic diversity per site, pairwise comparison to Nsp12 | | | | |
|---|---|---|---|---|
| **Region** | **p** | **p.adj** | **p.format** | **Significance** |
| ORF3a | 5.29E-63 | 1.60E-60 | <2E-16 | **** |
| NP | 1.17E-46 | 3.50E-44 | <2E-16 | **** |
| ORF8 | 2.74E-45 | 8.20E-43 | <2E-16 | **** |
| ORF7a | 2.88E-38 | 8.50E-36 | <2E-16 | **** |
| S | 1.78E-18 | 4.40E-16 | <2E-16 | **** |
| Nsp2 | 2.45E-40 | 7.30E-38 | <2E-16 | **** |
| Nsp1 | 1.75E-20 | 4.50E-18 | <2E-16 | **** |
| Nsp3 | 5.96E-19 | 1.50E-16 | <2E-16 | **** |
| ORF6 | 2.76E-14 | 6.50E-12 | 2.80E-14 | **** |
| ORF7b | 1.22E-12 | 2.80E-10 | 1.20E-12 | **** |
| ORF10 | 1.15E-08 | 2.30E-06 | 1.10E-08 | **** |
| Nsp14 | 3.29E-05 | 5.10E-03 | 3.30E-05 | **** |
| Nsp4 | 1.45E-04 | 2.10E-02 | 0.00014 | *** |
| Nsp15 | 4.16E-04 | 5.90E-02 | 0.00042 | *** |
| Nsp6 | 4.87E-04 | 6.80E-02 | 0.00049 | *** |
| Nsp13 | 3.76E-03 | 4.80E-01 | 0.00376 | ** |
| Nsp7 | 5.89E-03 | 7.10E-01 | 0.00589 | ** |
| M | 1.39E-02 | 1.00E+00 | 0.01388 | * |
| Nsp9 | 1.40E-02 | 1.00E+00 | 0.01396 | * |
| Nsp5 | 3.12E-02 | 1.00E+00 | 0.03125 | * |

| No. Homoplasies per site, pairwise comparison to Nsp12 | | | | |
|---|---|---|---|---|
| **Region** | **p** | **p.adj** | **p.format** | **Significance** |
| NP | 2.69E-08 | 8.10E-06 | 2.70E-08 | **** |
| ORF3a | 4.84E-08 | 1.40E-05 | 4.80E-08 | **** |
| ORF8 | 1.11E-04 | 3.20E-02 | 0.00011 | *** |
| S | 1.41E-04 | 4.10E-02 | 0.00014 | *** |
| ORF7a | 4.22E-03 | 1.00E+00 | 0.00422 | ** |
| ORF7b | 1.57E-02 | 1.00E+00 | 0.01565 | * |
| Nsp14 | 2.51E-02 | 1.00E+00 | 0.02505 | * |

Pairwise differences are assessed following wilcoxon test.

**Extended Data Table 4 | Immunogenic peptides recognised by CD4+ or CD8+ T cells in seronegative HCW**

| | Protein (amino acid residues) | SARS-CoV-2 amino acid sequence | MHC restriction (predicted) |
|---|---|---|---|
| **CD4** | NSP7 (21-35) #5 | RVESSSKLWAQCVQL | - |
| | NSP7 (51-65) #11 | KMVSLLSVLLSMQGA | - |
| | NSP7 (71-85) #15 | NKLCEEMLDNRATLQ | - |
| | NSP12 (436-450) #88 | ELKHFFFAQDGNAAI | - |
| | NSP12 (446-460) #90 | GNAAISDYDYYRYNL | - |
| | NSP12 (506-520) #102 | FNKWGKARLYYDSMS | - |
| | NSP12 (851-865) #171 | DGTLMIERFVSLAID | - |
| | NSP13 (391-405) #79 | LRAKHYVYIGDPAQL | - |
| | NSP13 (586-600) #118 | FTSLEIPRRNVATLQ | - |
| **CD8** | NSP7 (21-35) #5 | RVESSS**KLWAQCVQL** | A*02:01 |
| | NSP12 (436-450) #88 | ELKHFF**FAQDGNAAI** | (A*24:02) |
| | NSP12 (446-460) #90 | G**NAAISDYDY**YRYNL | (B*35:01) |
| | NSP12 (826-840) #166 | YV**YLPYPDPSRI**LGA | B*51:01 |
| | NSP13 (391-405) #79 | LRAKHY**VYIGDPAQL** | A*24:02, C*07:01 |
| | NSP13 (586-600) #118 | FTSLEI**PRRNVATL**Q | B*07:02, B*08:01 |

**Extended Data Table 5 | Demographics and sampling of Close Contact medical student/laboratory staff cohort**

| | Serostatus/PCR | Gender | Age (post exposure sample) | Exposure | NSP12 Response | Post-exposure sample: Months since known exposure/PCR+ or symptoms |
|---|---|---|---|---|---|---|
| **Close contact** | Seronegative | F | 20-24 | Household Contact | - | 8-9 |
| | Seronegative | M | 20-24 | Suspected household contact* | Expanded | 8-9 |
| | Seronegative | M | 20-24 | Close Contact | Newly Detected | 7-8 |
| | Seronegative | F | 20-24 | Suspected household contact* | Newly Detected | 6-7 |
| | Seronegative | M | 20-24 | Suspected household contact* | Newly Detected | 9-10 |
| | Seronegative | F | 20-24 | Close Contact | Expanded | 9-10 |
| | Seronegative | F | 20-24 | Household Contact | Newly Detected | 9-10 |
| | Seronegative | M | 20-24 | Close Contact | Expanded | 8-9 |
| | Seronegative | M | 65-69 | Household Contact | Expanded | 7-8 |
| | Seronegative | F | 25-29 | Household Contact | - | 6-7 |
| **Lab-confirmed infection** | Seropositive | M | 20-24 | not known | - | 9-10 |
| | Seropositive | F | 20-24 | Household Contact | - | 2-3 |
| | Seropositive | F | 20-24 | Close Contact | - | 9-10 |
| | Seropositive | F | 20-24 | Suspected household contact* | - | 9-10 |
| | PCR+ Seropositive | F | 20-24 | Close Contact | Newly Detected | 2-3 |
| | PCR+ Seropositive | M | 20-24 | Close Contact | Expanded | 1-2 |
| | Seropositive | F | 20-24 | Close Contact | Expanded | 9-10 |
| | PCR+ Seropositive | F | 20-24 | Household Contact | - | 1-2 |
| | Seropositive | F | 25-29 | not known | Expanded | 4-5 |
| | Seropositive | M | 25-29 | Household Contact | - | 5-6 |
| | Seropositive | F | 25-29 | not known | - | 4-5 |
| | PCR+ Seropositive | F | 30-34 | Occupational | - | 1-2 |
| | PCR+ Seropositive | M | 25-29 | not known | - | 1-2 |

*Household contact with case-defining symptoms but no PCR confirmation available early 2020. Expanded = >2 fold or >35 SFU/$10^6$ PBMC increase in NSP12 response from pre-pandemic to post-exposure or infection time point.

# nature research

# Reporting Summary

Nature Research wishes to improve the reproducibility of the work that we publish. This form provides structure for consistency and transparency in reporting. For further information on Nature Research policies, see Authors & Referees and the Editorial Policy Checklist.

## Statistics

For all statistical analyses, confirm that the following items are present in the figure legend, table legend, main text, or Methods section.

| n/a | Confirmed | |
|---|---|---|
| ☐ | ☒ | The exact sample size (*n*) for each experimental group/condition, given as a discrete number and unit of measurement |
| ☐ | ☒ | A statement on whether measurements were taken from distinct samples or whether the same sample was measured repeatedly |
| ☐ | ☒ | The statistical test(s) used AND whether they are one- or two-sided<br>*Only common tests should be described solely by name; describe more complex techniques in the Methods section.* |
| ☒ | ☐ | A description of all covariates tested |
| ☐ | ☒ | A description of any assumptions or corrections, such as tests of normality and adjustment for multiple comparisons |
| ☐ | ☒ | A full description of the statistical parameters including central tendency (e.g. means) or other basic estimates (e.g. regression coefficient) AND variation (e.g. standard deviation) or associated estimates of uncertainty (e.g. confidence intervals) |
| ☐ | ☒ | For null hypothesis testing, the test statistic (e.g. *F*, *t*, *r*) with confidence intervals, effect sizes, degrees of freedom and *P* value noted<br>*Give P values as exact values whenever suitable.* |
| ☒ | ☐ | For Bayesian analysis, information on the choice of priors and Markov chain Monte Carlo settings |
| ☒ | ☐ | For hierarchical and complex designs, identification of the appropriate level for tests and full reporting of outcomes |
| ☒ | ☐ | Estimates of effect sizes (e.g. Cohen's *d*, Pearson's *r*), indicating how they were calculated |

*Our web collection on statistics for biologists contains articles on many of the points above.*

## Software and code

Policy information about availability of computer code

| | |
|---|---|
| Data collection | A complete masked alignment was downloaded from the GISAID EpiCoV database on 26/7/2021 together with a GISAID Audacity phylogeny comprising 611,893 accessions. The alignment was subsampled to include 800 of each defined NextStrain phylogenetic clade, as provided by GISAID metadata. For clades containing less than 800 accessions all representatives of that clade were included resulting in a comprehensive sampling over the global phylogeny of 13,785 accessions encompassing the genomic diversity of SARS-CoV-2 to date (Supplementary Table 4, Extended Data Fig. 5). |
| Data analysis | Software used for data/statistical analysis: FlowJo v.10.7.1; FACSDIVA v9.0; Prism 7.0e and 9.0; Excel v.16.16.09; R version 3.5.3 with RStudio Version 1.0.153 for Mac.<br><br>Custom scripts used to perform the homology searches, heatmap visualisation and permutation testing are hosted on GitHub (https://github.com/cednotsed/tcell_cross_reactivity_covid.git). Correlogram was produced using corrplot in R (https://github.com/taiyun/corrplot). Polyfunctionality was visualised using SPICE (version 6.0) and pestle (version 2.0), available at https://niaid.github.io/spice/. MUSCLE algorithm with default parameters and percentage identity was calculated in Geneious Prime 2020.1.2. Alignment figures were made in Snapgene 5.1 (GSL Biotech). |

For manuscripts utilizing custom algorithms or software that are central to the research but not yet described in published literature, software must be made available to editors/reviewers. We strongly encourage code deposition in a community repository (e.g. GitHub). See the Nature Research guidelines for submitting code & software for further information.

## Data

Policy information about availability of data

All manuscripts must include a data availability statement. This statement should provide the following information, where applicable:

- Accession codes, unique identifiers, or web links for publicly available datasets
- A list of figures that have associated raw data
- A description of any restrictions on data availability

All data analysed during this study are included in this published article (and its supplementary information files). Genomic data analysed was obtained from the publicly available NCBI Virus database and, following registration, from the GISAID EpiCoV repository (full list and metadata available at: 10.6084/m9.figshare.16607423). The datasets generated during and/or analysed during the current study are available from the corresponding author on reasonable request. Correspondence and requests for materials should be addressed to MKM or LS.

Protein sequences for SARS-CoV-2 ORF1ab (accession numbers: QHD43415.1, NP_828849.2, YP_009047202.1, YP_009555238.1, YP_173236.1, YP_003766.2 and NP_073549.1) and for HCoV (accessions listed in Supplementary Table 1, NCBI Virus using 245 the taxid: 1118 together with accompanying metadata) were downloaded from the NCBI database (https://www.ncbi.nlm.nih.gov/).

# Field-specific reporting

Please select the one below that is the best fit for your research. If you are not sure, read the appropriate sections before making your selection.

☒ Life sciences ☐ Behavioural & social sciences ☐ Ecological, evolutionary & environmental sciences

For a reference copy of the document with all sections, see nature.com/documents/nr-reporting-summary-flat.pdf

# Life sciences study design

All studies must disclose on these points even when the disclosure is negative.

| | |
|---|---|
| Sample size | Sample sizes are given for each figure throughout the paper when individual dots are not shown. Power calculations were performed prior to week 16 sub-study sampling to determine the sample size needed to test the hypothesis that HCW with pre-existing T cell responses are enriched in exposed uninfected group at a range of incidence of infection, assuming 50% of cohort had pre-existing T cell responses. Sample sizes of 18-64 per group were estimated. An age, sex and ethnicity matched nested substudy was designed within the larger (n=731) parent study and 129 attended for 16 week sampling including high volume PBMC isolation. Sample size can vary across figure panels depending on which stimulations were performed (limited by number of PBMC recovered). Cohort sizes given in Figure 1a. |
| Data exclusions | Classification of HCW and study participants into cohorts is defined in methods as are any specific exclusions of data points from individual graphs. Two HCW in the seronegative cohort (negative for NP and S1 antibodies wk 0-16) had nAb titres just above the threshold IC50 of 50 were excluded from further analyses (exclusion criteria not pre-established, determined using unexposed pre-pandemic and PCR+ samples). No other HCW or individual samples were excluded after data was generated. |
| Replication | Replication for each assay are described in the methods. Briefly, per sample unstimulated controls were run in duplicate for ELISpot data with no data excluded due to outliers. Due to limited sample availability ELISpots were only repeated on a small number of pre-pandemic samples. Replicates were successful. CTV proliferation assays were repeated and experimental replicates performed on a subset of individuals successfully. Duplicates were used for S1 ELISAs with no outliers excluded. qPCR was repeated on a subset of individuals successfully. Neutralization assays were preformed over a wide range of dilutions in duplicate. |
| Randomization | Experiments were performed with protocols optimised to reduce batch variation and to ensure mixing of experimental groups across batches e.g Flow cytometer parameters were consistent between runs (No MFI comparisons were performed, only gating and percentage of parent). Samples from pre-pandemic, seronegative HCW and seropositive HCW were ran in parallel on ELISpot plates.<br><br>Laboratory-confirmed infection was determined by weekly nasopharyngeal RNA stabilizing swabs and reverse transcriptase polymerase chain reaction (RT-PCR; Roche cobas SARS-CoV-2 test, Envelope [E] gene) and antibody assay positivity (Spike protein 1 IgG Ab assay, EUROIMMUN) and anti-nucleocapsid total antibody assay (ROCHE). The seronegative health care worker group were matched for demographics and exposure to the laboratory-confirmed infected group and was defined by negativity by these three tests at all 16 time points as well as negative for neutralising antibodies at week 16 and at selected prior time points as indicated. Unexposed pre-pandemic samples were not matched for demographics (Demographics given in Extended Data Table 1). 'Close-contact cohort' self-identified as having had close contact (household contact or alert by NHS test-and-trace app of close contact with a confirmed case) were divided into seropositive or seronegative (determined by S1 ELISA), Extended Data Table 4. |
| Blinding | IFNg-ELISpot assays were performed on HCW cohorts prior to unblinding of group (laboratory-confirmed-infection or seronegative). Other experiments were not randomized and the investigators were not blinded to allocation during experiments and outcome assessment, however, experimental set-up and controls ensured accurate replication across technical replicates (see above and methods). |

# Reporting for specific materials, systems and methods

We require information from authors about some types of materials, experimental systems and methods used in many studies. Here, indicate whether each material, system or method listed is relevant to your study. If you are not sure if a list item applies to your research, read the appropriate section before selecting a response.

## Materials & experimental systems

| n/a | Involved in the study |
|---|---|
| ☐ | ☒ Antibodies |
| ☒ | ☐ Eukaryotic cell lines |
| ☒ | ☐ Palaeontology |
| ☒ | ☐ Animals and other organisms |
| ☐ | ☒ Human research participants |
| ☐ | ☒ Clinical data |

## Methods

| n/a | Involved in the study |
|---|---|
| ☒ | ☐ ChIP-seq |
| ☐ | ☒ Flow cytometry |
| ☒ | ☐ MRI-based neuroimaging |

# Antibodies

**Antibodies used**

Detailed information regarding all antibodies and other fluorescent agents used in this study are listed in the methods with manufacturer, clone, and dilution used.
ELISpot - human anti-IFNγ Ab (1-D1K, Mabtech; 10 µg/ml), biotinylated IFN-γ detection antibody (7-B6-1, Mabtech; 1µg/ml).

FACS: Memory B cell Panel: CD3 Bv510 (Biolegend, clone OKT3, 1:200), CD11c FITC (BD Biosciences, clone B-ly6, 1:100), CD14 Bv510 (Biolegend, clone M5E2, 1:200), CD19 Bv786 (BD bioscience, clone HIB19, 1:50), CD20 AlexFluor700 (BD biosciences 2H7, 1:100), CD21 Bv711 (BD biosciences, clone B-ly4, 1:100), CD27 BUV395 (BD biosciences, clone L128, 1:100), CD38 Pe-CF594 (BD biosciences, clone HIT2, 1:200), IgD Pe-Cy7 (BD biosciences, clone IA6-2, 1:100).

FACS: CTV assay: IL-2 PerCp-eFluor710 (Invitrogen, clone MQ1-17H12, 1:50), TNFα FITC (BD bioscience, clone MAb11, 1:100), CD8α BV785 (Biolegend, clone RPA-T8, 1:200), IFN? BV605 (BD biosciences, clone B27, 1:100), IFN? APC (Biolegend, clone 4S.B3, 1:50), CD3 BUV805 (BD biosciences, clone UCHT1, 1:200), CD4 BUV395 (BD biosciences, clone SK3, 1:200), CD154 (CD40L) Pe-Cy7 (Biolegend, clone 24-31, 1:50), MIP-1-β PE (BD biosciences, clone D21-1351, 1:100).

FACS:MHC class I pentamer panel: CD3 BUV805 (BD biosciences, clone UCHT1, 1:200), CD4 BUV395 (BD biosciences, clone SK3, 1:200), CD56 Pe-Cy7 (BD biosciences, NCAM16.2, 1:100), CD8α Alexa700 (Biolegend, RPA-78, 1:200), post-expansion CD19 Bv786 (BD biosciences, HIB19, 1:100).

**Validation**

All antibodies and MHC class I pentamers were purchased from well established manufacturers and were validated by the vendor for species and target. e.g. BD biosciences, Biolegend, and Invitrogen antibodies are tested in Knock-out/knock-in primary model systems to ensure biological accuracy in ISO 9001 certified facilities. Side-by-side lot comparisons are performed. Details of antibody clones have been included for cross-referencing of manufacturing company specification/validation processes. We further validated antibodies by titration to optimal concentrations and by using positive controls where possible (e.g. using populations known to express a certain marker or by polyclonal stimulation). MHC class I pentamers were tested in HLA-mismatched individuals to assess background staining and on T cell clones expanded with cognate peptide.

Fluorescence minus one stains were used to define gates in Flowjo for all FACS assays . Positive (SARS-CoV-2 laboratory-confirmer infected) and negative controls (unexposed pre-pandemic samples) were included in each run for memory B cell staining.  Positive control wells were used in CTV stains to ensure accurate staining of cytokines and CTV staining was checked on day 0 before stimulation. Unstimulated control wells treated as peptide wells (e.g. addition of DMSO) were run per biological sample for CTV proliferation assays and ELISpots (in duplicate for ELISpots) and all data is presented as background subtracted as described in the methods.

# Human research participants

Policy information about studies involving human research participants

**Population characteristics**

Age, sex and ethnicity of cohorts are provided in Extended Data Table 1 and 4, and in detail for the COVIDsortium in Augusto et al Wellcome Open Research 2020. Substudy recruitment for all wk16 data presented was performed on a cohort of seronegative HCW matched for age, sex, and ethnicity with a group of laboratory-confirmed infected HCW.

**Recruitment**

Recruitment is described in details in Augusto et al Wellcome Open Research and in the methods section. Adult (>18 years) hospital HCWs who were fit and well to attend work in any role and across a range of clinical areas, were invited to participate via hospital email, posters, staff meetings, training sessions and participant information leaflets (see https://covid-consortium.com). No other inclusion or exclusion criteria were considered.The "COVID-19 Immune Protection and Pathogenesis in Healthcare Worker Bioresource" (NCT04318314) uses a prospective cohort design (Figure 1). The study consists of questionnaires and biological samples (blood samples, nasal swabs ± saliva) performed at all visits: baseline, weekly follow-ups for 15 weeks, and visits at 6 and 12 months. An age, sex and ethnicity matched nested sub-study was designed within the larger (n=731) parent study and 129 attended for 16-week sampling including high volume PBMC isolation. For the 'close-contact cohort' medical students previously enrolled in a BCG vaccine trial (UCL Ethics Project ID Number: 13545/001) were invited to participate by email and were re-consented.

**Ethics oversight**

The COVIDsortium bioresource was approved by the ethical committee of UK National Research Ethics Service (20/SC/0149) and registered on ClinicalTrials.gov (NCT04318314). The cohort of medical students and laboratory staff was approved by UCL Ethics (Project ID Number: 13545/001) and pre-pandemic healthy donor samples were collected and cryopreserved before August 2019 under ethics numbers 11/LO/0421. All subjects gave written informed consent and the study conformed to the principles

of the Helsinki Declaration.

Note that full information on the approval of the study protocol must also be provided in the manuscript.

# Clinical data

Policy information about clinical studies

All manuscripts should comply with the ICMJE guidelines for publication of clinical research and a completed CONSORT checklist must be included with all submissions.

| Clinical trial registration | ClinicalTrials.gov (NCT04318314) |
| --- | --- |
| Study protocol | ClinicalTrials.gov (NCT04318314), Augusto et al Wellcome Open Research 2020. |
| Data collection | Data collection is described in detail in Augusto et al Wellcome Open Research and in the methods section. The "COVID-19 Immune Protection and Pathogenesis in Healthcare Worker Bioresource" (NCT04318314) uses a prospective cohort design. The study consists of questionnaires and biological samples (blood samples, nasal swabs ± saliva) performed at all visits: baseline, weekly follow-ups for 15 weeks, and visits at 6 and 12 months. Recruitment was initially at St Bartholomew's Hospital, London, UK (400 HCWs recruited between 23rd and 31st March 2020, just before the peak of new daily cases in London, which happened on the 2nd April, with 1,022 new cases confirmed). To improve statistical power for downstream analyses, we expanded the target sample size to n=1,000 and extended recruitment on 17th April 2020 to other local sites: Royal Free NHS Hospital Trust (large teaching hospital with specialist expertise in infectious diseases). Baseline: Participants complete a baseline questionnaire including standard variables related to demographics and exposures. These included occupation, household details, smoking status, physical activity, anthropometry, medical history (including vaccination history, current medication and dietary supplements), occupational exposure (including specific clinical areas and access to/use of personal protective equipment [PPE]), travel history, previous COVID-19 symptoms, proven contact with SARS-CoV-2 infected individuals, and any prior testing for SARS-CoV-2 infection. Follow-up: Following recruitment (baseline visit), if fit and well to attend work, participants would undertake in-person weekly questionnaires using research electronic data infrastructure (REDCap v8.5.22)16 to capture occupational metadata, new SARS-CoV-2 exposure, symptoms and test results, and biosample collection. |
| Outcomes | Prospective HCW study. Not applicable |

# Flow Cytometry

## Plots

Confirm that:

☒ The axis labels state the marker and fluorochrome used (e.g. CD4-FITC).

☒ The axis scales are clearly visible. Include numbers along axes only for bottom left plot of group (a 'group' is an analysis of identical markers).

☒ All plots are contour plots with outliers or pseudocolor plots.

☒ A numerical value for number of cells or percentage (with statistics) is provided.

## Methodology

| Sample preparation | Detailed sample preparation is given in methods. All FACS was performed on frozen and thawed PBMC isolated by density gradient separation. Peripheral blood mononuclear cells (PBMC) were isolated from heparinized blood samples using Pancoll (Pan Biotech) or Histopaque®-1077 Hybri-MaxTM (Sigma-Aldrich) density gradient centrifugation in SepMate tubes (StemCell) according to the manufacturer's specifications. Isolated PBMCs were cryopreserved in fetal calf serum (FCS) containing 10% DMSO and stored in liquid nitrogen. |
| --- | --- |
| Instrument | BD biosciences LSRII and Fortessa-X20 flow cytometers. |
| Software | FACS DIVA version 9.0 was used on instrument and exporting .fcs files were analysed in FlowJo version 10.7.1 (TreeStar) |
| Cell population abundance | PBMC were stained and run without sorting or enrichment. |
| Gating strategy | Example gating strategy for CTV proliferation and mapping FACS experiments is given in Extended Data Figure 3a. Example plots are given in Fig. 2c, Fig 3g, and Extended Data Fig 3d and Extended Data Fig. 6a. Data is reported as a percentage of lymphocytes/singlets/live/CD3+/CD4+ or CD8+ defining antigen specificity by production of IFNg and CTV dilution. For memory B cell stains example plots are given in Extended Data Figure 1b and details of cutoff and assay validation are given in Jeffery-Smith et al BioRxiv 2021. Gating is described in legends: MBC expressed as a percentage of lymphocytes, singlets, Live, CD3-CD14-CD19 +, CD20+, excluding CD38hi, IgD+ and CD21+CD27- fractions. MHC class I Pentamer gating and example plots in Extended Data Figure 6c. |

☒ Tick this box to confirm that a figure exemplifying the gating strategy is provided in the Supplementary Information.

