## [Peer Review File · Nature]

Manuscript Title: Pre-existing polymerase-specific T cells expand in abortive seronegative SARS-CoV-2

Editorial Notes:

Reviewer Comments & Author Rebuttals

Reviewer Reports on the Initial Version:

Referee #1 (Remarks to the Author):

“Pre-existing polymerase-specific T cells expand in abortive seronegative SARS-CoV-2 infection” primarily focuses on a cohort of health care workers in England who were monitored weekly by PCR for SARS-CoV-2 infection for 16 weeks, but did not report a positive test or seroconvert over that period, despite known likely exposures. These subjects are compared to a cohort of infected individuals from the same surveillance, and two pre-pandemic cohorts from London and Singapore. The authors report evidence of T cell expansion against SARS-CoV-2 over the study period, particularly against non-structural proteins that constitute the replication complex, which they suggest might be an early target of protective, cross-reactive T cells due to its early expression kinetics. They also find a correlation in subjects with the highest RTC complex T cell response and the expression of IFI27, an IFN response gene that might indicate an innate response to low levels of virus. They map individual peptides that have cross-reactive activity in the polymerase between HCoV and SARS-CoV-2, and examine specific responses to the polymerase in matched samples pre- and post-pandemic from both the original cohort and a new cohort. Here they see evidence for an increase in a subset of subjects and a similar pattern is found in their primary HCW cohort, with NSP12 consistently the most informative and apparently sensitive target.

There have been other reports of potential T cell expansion in non-seroconverters/non-positive individuals, but this is particularly thoroughly done in many respects. Understanding cryptic or subclinical infections is important to understanding the dynamics of SARS-CoV-2, particularly as the pandemic matures and is complicated by varying level and types of vaccination. Some of the results, particularly in Figure 4 are compelling. I do have some specific suggestions and concerns:

1) The authors never really state a definition of what they consider a bona fide abortive infection. To put it another way, at the end of the analysis, what proportion of their cohort(s) do they think actually had exposure to the virus and responded immunologically? It might be difficult to draw a firm line but in some places they seem to be arguing for “nearly everyone” and in others (e.g. 2f) they split the cohort into low responders and high responders, but they don’t tell us the logic behind the split. In the end I think some sense of how widespread a phenomenon this could be would be a

useful discussion point. Some discussion also of whether nsp12 would be worth targeting in a vaccine construct would be a good addition too.

2) The cohorts need much better definition—the ext table 1 is relatively sparse and doesn't include the 23 person cohort I think that appears in figure 4.

3) The figure legends need more detail about what cohort is being used and what things mean—extended figure 3 for instance—I have no clue how many people are represented or what the numbers mean in ext fig 3c. It's said to be something about sub pools, and I'm not sure what that means either.

4) For many of the analyses in figure 1 and 2 (until 2f), there would be a lot of value in seeing correlation matrices of the results. They try to get at this by doing ratios but it would be helpful to see how people were related to themselves across the different parameters measured.

5) How many people were analyzed in fig 2 and ext fig 3 for the cytokine production? Only representative plots are shown and no summary data? It seems like it might just be a handful, in which case it should be significantly expanded (or another technique used, see below).

6) Fig 2f is an important panel and definitely shows that there is “structure” in the cohort, but the correlation between IFI27 and the RTC response is apparent already at day 0. Is the argument that a large (almost half) portion of the cohort already “infected” at day 0? Is that plausible? We are shown stats on group averages, but there's longitudinal data here that could be modeled much more informatively—how many people actually rise? In the whole cohort is there any signal from someone who suddenly both expands RTC responses and upregulated IFI27?

7) Figure 3 doesn't really formally map an epitope, which needs to be confirmed ideally with tetramer binding. Further, the responses are reported in a relativized manner in 3e and should be shown as an absolute value as all the other ELISPOT data are shown in the paper. It would greatly increase the support for the cross-reactive argument if they could show binding by tetramer to two variant epitopes. As it is, this is mostly suggestive.

8) The comparison to the other pre-pandemic London and Singapore cohorts may be confounded by a number of factors—it's suggestive, but the paired sample sets in figure 4 are more powerful. The 23 person cohort in the beginning of figure 4 should also have CEF peptide control run on it, and for both cohorts, we should see the paired behavior across the time points (e.g. in 4d)

Referee #2 (Remarks to the Author):

The paper by Swalding et al. presents data related to the hypothesis that preexisting T cell responses are associated with abortive SARS CoV2 infection. This issue is of considerable interest for our understanding of SARS CoV2 infection in general, and immune mechanisms of protection from and resolution of infection. The data presented, while intriguing and suggestive, does not directly prove the hypothesis, and several interpretations are not adequately supported.

In Fig 4, data is presented related to paired PBMC samples taken pre-pandemic and sampled them again after known close contact with infected cases. This is a very interesting, unusual and exciting cohort. The Exposed Seronegative Health Care Workers (HCW) cohort was monitored for a period of 16 weeks by state-of-the-art assays and this is a point of strength. Of great interest, it is shown that “NSP12-specific T cells already detectable at baseline in 74% of ES with the strongest NSP12 responses post-exposure (Fig. 4c). NSP12 responses expanded in vivo on average 8.4-fold between recruitment and wk16 of follow-up, with no corresponding change in Flu/EBV/CMV responses (Fig. 4d)”. Conversely, responses against Spike are not expanded. This intriguing observation, is indeed consistent with (but does not prove) potential exposure and abortive infection.

The data in Fig4 lacks a crucial control, namely the analysis of T cell responses from seropositive HCW, and their comparison to T cell responses of ES. It is crucial to show that the level of T cell pre-pandemic is correlated to becoming seropositive vs seronegative upon the hypothesized exposure during the pandemic.

Fig1 relates to “T cell reactivity of HCW who did not develop documented SARS-CoV-2 infection despite likely exposure during the first UK pandemic wave”, as compared to those HCW exposed and infected. This appears to be an overinterpretation, since the mere fact of being a HCW and being in contact with COVID does not provide evidence of exposure to the virus. Presumably the individuals implemented protective measures; while the immune system of some might have been exposed to the virus, to simply assume so is a serious overinterpretation.

A control cohort of non-HCW, non-exposed subjects is not presented. Data from a pre-pandemic cohort is presented, but lack of a contemporary cohort is still a major limitation, since seasonal exposure to common cold coronaviruses could change the level and patterns of reactivity.

The next section analyses RTC-specific T cell in ES. It is stated that “The infected group had memory T cells dominated by more responses to structural proteins (Spike, membrane, NP, and ORF3a) than to RTC (NSP7, NSP12, NSP13) (Fig. 2a-b). By contrast, pre-existing T cell responses predominantly

targeted RTC proteins, whilst ES recognized both regions.” This interpretation is not supported by the data shown in Fig 2a-b; which by contrast shows that while ES responses are overall higher, the relative balance of responses is similar in ES and pre-pandemic samples.

In the same section is stated that” T cells generated by early, transient viral exposure preferentially focus on key components of the RTC “. As mentioned above, this is overstated, since no evidence of early transient viral exposure is presented.

In the following section it is stated that T cell responses are focused on RTC, and it is shown that RTC most conserved in CCC. However, it can't be concluded T cell responses target RTC (NSP7, NSP12, NSP13) because many NSP are not analyzed and some of them are just as conserved. NSP 5,8,10,14,16 have higher homology than NSP7; NSP 16 has higher homology than NSP12.

Furthermore, the fact that the RTC is conserved provides an alternative explanation for the data presented; e.g. responses are directed to RTC because of conservation and preexisting CCC responses, rather than abortive infection, as also shown by the data demonstrating similar levels of NSP12 recognition in pre-pandemic samples.

Referee #3 (Remarks to the Author):

In this report, Swadling et al study healthcare workers (HCWs) with intense exposure to SARS-CoV-2. These HCWs remain seronegative but develop T cell responses to nonstructural proteins, especially the polymerase, nsp12, that are similar in magnitude to those detected in seropositive HCWs. These HCWs were monitored weekly for 16 weeks evidence of COVID-19 by PCR and serology. T cell assays were performed at week 16. The authors then postulate that these cells expand in the seronegative HCWs in response to subclinical infection, contributing to rapid clearance of the virus, thereby providing protection from infection. Most of these conclusions have been reached previously, although the data in this manuscript are especially compelling.

PBMCs from students and laboratory workers obtained prior to the pandemic were also studied. These individuals also had responses to the same viral proteins, although at lower levels. They were exposed to SARS-CoV-2, but mostly in the context of sharing a household with a SARS-CoV-2 member. A unique aspect of the study are the data shown in Figure 4 in which paired prepandemic and postpandemic PBMC samples from these medical students and laboratory staff were analyzed. Household exposure is likely not as prolonged as occurred in the hospital but increases in virus-specific T cells were observed. If these samples or those from seronegative HCWs also shown in Figure 4 (4c-e) could be used to demonstrate expansion of epitope-specific (as opposed to protein-

specific T cells), the conclusions about expansion of pre-existing cells would be strengthened considerably.

Specific comments.

1. Lines 96, 399-409-As the authors point out, the 5' end of the genome is translated after cell entry to produce ORF1a/b proteins involved in transcription/replication. However, these proteins are believed to be immediately incorporated into membranous structures, initiating the production of subgenomic negative and positive strand RNA. The latter are used as the template for producing proteins, including structural proteins which are made in larger amounts than are the nonstructural proteins. It is unlikely that this small difference in time plays any role in the rapidity in which a T cell response is activated. This should be modified since it is not correct and does not change the importance of the results.

2. Line 328-339, Figures 4a, 4b-These data are the most novel in the report and most strongly support the conclusion that pre-existing virus-specific T cells expanded in response to infection. Since the responses could also be de novo, the argument would be strengthened considerably if the pre-pandemic and postexposure responses were shown to be directed at the same epitopes or subpools of nsp-specific epitopes.

3. Lines 341-388, Figures 4c-e-PBMCs were available at the time of enrollment and at 16 weeks for some seronegative HCWs. In addition to presenting the composite data shown in figure 4e, paired data should be shown for individual subjects in a separate figure or Extended Data Figure, to support the conclusion that pre-existing T cells responding to specific subpools expanded in response to infection. While it may not be possible to show that single epitope-specific cells expanded because of PBMC availability, this approach could at least demonstrate concordance between the early and later responses to nsp-specific peptide pools since these data are in hand.

4. Line 394-6-This conclusion is premature since it is equally likely that the infection is controlled by the innate or another aspect of the immune response and that T cell expansion is a secondary phenomenon (as the authors mention on lines 447-449). It should be toned down.

Minor comments.

1. Figure 2c-What stimulator was used in this assay?

2. Line 306-HKU1 is a seasonal CoV, so this sentence should be corrected.

Author Rebuttals to Initial Comments:

Referees' comments:

Referee #1 (Remarks to the Author):

.....There have been other reports of potential T cell expansion in non-seroconverters/non-positive individuals, but this is particularly thoroughly done in many respects. Understanding cryptic or subclinical infections is important to understanding the dynamics of SARS-CoV-2, particularly as the pandemic matures and is complicated by varying level and types of vaccination. Some of the results, particularly in Figure 4 are compelling.

We are glad you found our study thorough, the topic of subclinical infection important and results in Fig. 4 particularly compelling.

I do have some specific suggestions and concerns:

1) The authors never really state a definition of what they consider a bona fide abortive infection. To put it another way, at the end of the analysis, what proportion of their cohort(s) do they think actually had exposure to the virus and responded immunologically? It might be difficult to draw a firm line but in some places they seem to be arguing for “nearly everyone” and in others (e.g. 2f) they split the cohort into low responders and high responders, but they don't tell us the logic behind the split. In the end I think some sense of how widespread a phenomenon this could be would be a useful discussion point.

We agree that the distinction between the whole cohort of seronegative, potentially exposed, healthcare workers (HCW), and the subset with evidence supporting likely exposure resulting in abortive infection, needed clarification in our manuscript. To address this point, and an overlapping comment from Reviewer 2, we have re-named the 'ES (exposed seronegative) HCW' cohort as simply 'seronegative HCW (SN-HCW)'. We have also now clarified how we have defined the subset with evidence for exposure. Distinguishing pre-existing responses from those expanded by exposure is difficult and we have not examined all possible T cell specificities in this study. We therefore focused on the 34% (20 out of 58) SN-HCW with the strongest RTC-specific T cells ($>50\text{SFU}/106\text{ PBMC}$, Fig. 2d) at 16 weeks and sought corroborative evidence for exposure in these subjects. Not only did these SN-HCW tend to have stronger NSP12 responses than pre-pandemics, most (74%) also had evidence for expansion of NSP12-specific T cells from baseline to week 16 (Fig. 4c). However, we consider those who also had an interferon-induced gene signal (a robust marker of early SARS-CoV-2 infection, Gupta et al Lancet Microbe 2021) to have the most convincing evidence for abortive infection. To assess this more accurately, we first established the normal range for IFI27 (Fig. 2e-f, greyed area) using samples from an asymptomatic pre-pandemic HCW cohort ($n=99$, Pollara et al STM 2021). Using a cut-off of $>2\text{SD}$ above the pre-pandemic mean (standardised Z score >2), 40% of those with the highest RTC-specific T cells had an IFI27 signal, i.e. 10.3% of the 58 SN-HCW originally examined (Fig. 2e-f).

To further corroborate this estimate, we extended the analysis of IFI27 to a larger cohort of 99 SN-HCW (not pre-selected by T cell responses), and found that 9.1% had levels above 2SD of the pre-pandemic mean at recruitment. Our longitudinal IFI27 analysis suggest this proportion would be increase to around 20% by week 5, in line with likely cumulative exposure. Thus, two independent analyses give an estimate of abortive infection in around 10% of this cohort at the time of recruitment. We have addressed this in the revised manuscript in Results lines 227-230. We have also now discussed the possibility that such abortive infection may primarily occur in the presence of low-level exposure such as HCW wearing PPE and may not be generalisable to viral variants that have since emerged (Conclusions line 536-539).

Some discussion also of whether nsp12 would be worth targeting in a vaccine construct would be a good addition too.

This is an important discussion point which we have now emphasised more strongly. To strengthen the rationale for the inclusion of NSP12 (RNA polymerase) in future vaccines, we include new data provided by co-authors van Dorp and Balloux, showing that NSP12 is one of the most highly conserved proteins, not only across all human and animal coronaviridae (previous submission and Tan et al bioRxiv 2020), but also across global circulating SARS-CoV-2 sequences (representative sub-sample of 13,785 accessions downloaded from GISAID on 27th July 2021). We employed two approaches to estimate evolutionary sequence conservation across the SARS-CoV-2 genome, Nei's classical measure of genetic diversity, which captures the amount of genetic variation within a population and a homoplasy index, which informs on the propensity of individual sites in a genome to undergo recurrent mutation. Both metrics indicate that NSP12 and NSP13 were significantly more conserved than many structural proteins. This high level of conservation, together with the potential for pre-existing NSP12-specific T cells to be boosted in vivo, support consideration of inclusion of NSP12 (and potentially other conserved regions within ORF1) in future vaccines. See new Fig.3c and Extended Data Fig. 5a-d, new results section lines 274-281 and Conclusion lines 543-545.

2) The cohorts need much better definition—the ext table 1 is relatively sparse and doesn't include the 23 person cohort I think that appears in figure 4.

We have now referred more clearly to Extended Data Table 4 which detailed the 23 person cohort (now referred to as the Contact Cohort). We have also included them in Extended Data Table 1 as requested.

We have added more demographic data to Extended Data Table 1 (previously published in Reynolds et al, Sci Immunol 2020, Augusto et al Wellcome Open Research 2020) including: recent travel, COVID-19 patient/colleague/household contact exposure, job role, job location, early PPE usage and use of aerosol creating procedure. In Extended Data Table 1 we also compare these demographics in SN-HCW with or without strong RTC-specific T cells. No clear differences in demographics were noted between those individuals that have laboratory-confirmed infection or those that remain

seronegative, or between seronegative individuals with and without strong RTC-specific T cell responses (Results section lines 237-241).

3) The figure legends need more detail about what cohort is being used and what things mean—extended figure 3 for instance—I have no clue how many people are represented or what the numbers mean in ext fig 3c. It's said to be something about sub pools, and I'm not sure what that means either.

All figure legends have been updated to now state whether they refer to "COVIDsortium" HCW data or the "Contact cohort" (Fig. 4 and Extended Data Fig. 8 only).

We apologise for lack of clarity in Extended Data Fig. 3c; we have improved the figure labelling and the legend as follows (now Extended Data Fig. 3e, results page 41):

"e, Proportion of SARS-CoV-2-specific T cells (CTV1oIFN γ +) that are CD4+ or CD8+ after 10-day expansion (the protein specificity is listed above, donor ID (a-l, corresponding to raw data in Extended Data Table 2) and peptide sub-pools used for stimulation listed below)."

Figure 1. legend has been updated to explain the meaning of 'sub-pools':

"g, SARS-CoV-2 proteome with RTC and structural regions assayed for T cell responses highlighted. When proteins are divided into subpools of peptides for stimulation, these are identified by numbered boxes, with the number of overlapping 15mer peptides (or mapped epitope peptides for spike) used in brackets below."

4) For many of the analyses in figure 1 and 2 (until 2f), there would be a lot of value in seeing correlation matrices of the results. They try to get at this by doing ratios but it would be helpful to see how people were related to themselves across the different parameters measured.

Thank you for this suggestion; we have produced a correlation matrix for SN-HCW of T cell responses to individual proteins, summed responses to RTC or structural regions, total summed SARS-CoV-2 T cell responses, age, sex, and peak IFI27 (new Extended Data Fig. 4c, results page 42). This confirms the weak positive correlations we had already shown in Extended Data Fig. 2c between the magnitude of structural and RTC-specific T cell responses but reveals that RTC responses tend to correlate more strongly with each other, as do structural responses. NSP7 responses correlated with IFI27 but there were no significant correlations with gender or age (Results section lines 237-241).

5) How many people were analyzed in fig 2 and ext fig 3 for the cytokine production? Only representative plots are shown and no summary data? It seems like it might just be a handful, in which case it should be significantly expanded (or another technique used, see below).

We have now included summary data (new Extended Data Fig. 3d, results page 41) showing comparable degrees of polyfunctionality for T cell responses directed against RTC (n=28 CD4+ and CD8+), structural (n=13 CD4+ and CD8+) and flu/EBV/CMV (FEC, n=6; CD8+) peptides.

6) Fig 2f is an important panel and definitely shows that there is “structure” in the cohort, but the correlation between IFI27 and the RTC response is apparent already at day 0. Is the argument that a large (almost half) portion of the cohort already “infected” at day 0? Is that plausible? We are shown stats on group averages, but there’s longitudinal data here that could be modeled much more informatively—how many people actually rise? In the whole cohort is there any signal from someone who suddenly both expands RTC responses and upregulated IFI27?

Yes, the IFI27 data suggest that 40% of the group of SN-HCW selected on the basis of raised RTC-specific T cell responses had already had their exposure to SARS-CoV-2 by the time of recruitment; this is plausible given that PCR positivity in laboratory-confirmed HCW peaked at the time of recruitment (Fig. 1b), which coincided with the peak of the first pandemic wave in the UK. We have added Fig. 2d to show the cut-off of RTC-specific T cells used to pre-select these SN-HCW for IFI27 measurement, and have marked on the percentage of this group that were above the threshold for IFI27, based on the new data from an asymptomatic pre-pandemic HCW cohort, at each timepoint in Fig. 2e-f, to illustrate the kinetics of induction of IFI27. By week 3, 82% of this group had an IFI27 signal above the threshold, whilst some only peaked at wk5. This is an unexpectedly delayed peak (based on the earlier peak of PCR positivity in the parallel laboratory-confirmed cohort of HCW, Fig. 1b) and is suggestive of either delayed/repetitive SARS-CoV-2 exposure, or a slow-onset, protracted innate response, in HCW with abortive infection.

We do not have longitudinal T cell data across the first 5 weeks of follow-up to directly correlate IFI27 kinetics and T cell data. 14/15 (93.3%) of HCW with strong RTC-specific T cells had a peak IFI27 signal above the normal range threshold, whereas none of the HCW with low/undetectable RTC-specific responses at wk16 had a raised IFI27 at any time during the first 5 weeks of follow-up (Fig. 2f). As requested we have now modelled the differences between the seronegative HCW with and without strong RTC-specific T cell responses, showing that there was a significantly higher slope (linear regression) and variance of IFI27 in longitudinal data collected over the first 5 weeks of follow-up in seronegative individuals who have strong RTC-specific responses post-exposure (new Extended Data Fig. 4, results page 42).

7) Figure 3 doesn’t really formally map an epitope, which needs to be confirmed ideally with tetramer binding. Further, the responses are reported in a relativized manner in 3e and should be shown as an absolute value as all the other ELISPOT data are shown in the paper. It would greatly increase the support for the cross-reactive argument if they could show binding by tetramer to two variant epitopes. As it is, this is mostly suggestive.

Absolute data are now included in Extended Data Fig. 6d-e (page 44) and have been extended to n = 5 for both epitopes studied. The NSP7 epitope has been mapped elsewhere

(<http://www.iedb.org/epitope/32240>) and we have now included ex vivo staining of 4 SN-HCW showing binding by HLA-A2 pentamers loaded with the SARS-CoV-2 and HKU1 variants of this NSP7 epitope (new Extended Data Fig. 6c, page 44). Additionally, we show that the HLA-A2/SARS-CoV-2 pentamer can bind CD8 T cells expanded in vitro with either the SARS-CoV-2 or HKU1 variant peptide (new Extended Data Fig.6c).

8) The comparison to the other pre-pandemic London and Singapore cohorts may be confounded by a number of factors—it's suggestive, but the paired sample sets in figure 4 are more powerful. The 23 person cohort in the beginning of figure 4 should also have CEF peptide control run on it, and for both cohorts, we should see the paired behavior across the time points (e.g. in 4d)

We agree that the HCW cohorts with laboratory-confirmed infection, T cell and innate signals of exposure, or neither are the most powerful contemporaneous comparator cohorts. We have added a sentence to stress that the pre-pandemic cohorts are not directly comparable to the HCW because of potential confounders but it is of interest that, nevertheless, they show the same dominance of NSP12 responses.

Referee #2 (Remarks to the Author):

The paper by Swalding et al. presents data related to the hypothesis that preexisting T cell responses are associated with abortive SARS CoV2 infection. This issue is of considerable interest for our understanding of SARS CoV2 infection in general, and immune mechanisms of protection from and resolution of infection. The data presented, while intriguing and suggestive, does not directly prove the hypothesis, and several interpretations are not adequately supported. In Fig 4, data is presented related to paired PBMC samples taken pre-pandemic and sampled them again after known close contact with infected cases. This is a very interesting, unusual and exciting cohort. The Exposed Seronegative Health Care Workers (HCW) cohort was monitored for a period of 16 weeks by state-of-the-art assays and this is a point of strength. Of great interest, it is shown that “NSP12-specific T cells already detectable at baseline in 74% of ES with the strongest NSP12 responses post-exposure (Fig. 4c). NSP12 responses expanded in vivo on average 8.4-fold between recruitment and wk16 of follow-up, with no corresponding change in Flu/EBV/CMV responses (Fig. 4d)”. Conversely, responses against Spike are not expanded. This intriguing observation, is indeed consistent with (but does not prove) potential exposure and abortive infection.

We are pleased you find our data intriguing and suggestive, the cohort exciting and the issue of abortive infection of great interest. We agree that our findings are consistent with exposure and abortive infection but that formally proving this is beyond the scope of this study and would require a human challenge trial.

The data in Fig4 lacks a crucial control, namely the analysis of T cell responses from seropositive HCW, and their comparison to T cell responses of ES. It is crucial to show that the level of T cell pre-pandemic is correlated to becoming seropositive vs seronegative upon the hypothesized exposure during the pandemic.

Thank you for this suggestion; we agree that the frequency of pre-existing NSP12 responses in individuals that go on to become infected and seroconvert is an important control. The difference in frequency of NSP12 responses we already demonstrated between those with laboratory-confirmed infection and SN-HCW at wk16 was suggestive of differences in pre-existing T cell responses but could also be due to differential responses following exposure. We have now been able to access valuable baseline samples from HCW taken either prior to becoming PCR+ or at least 4 weeks before seroconversion. Baseline samples were limited to 1-3 million PBMC so we concentrated on analysing NSP12 responses. We now show that pre-existing NSP12 responses are significantly less frequent in those that go on to become seropositive than those that remain seronegative but who have expanded or newly detected NSP12 T cell responses at wk16 (new Fig. 4i, new Extended Data Fig. 8i-j, results page 46).

Fig1 relates to “T cell reactivity of HCW who did not develop documented SARS-CoV-2 infection despite likely exposure during the first UK pandemic wave”, as compared to those HCW exposed and infected. This appears to be an overinterpretation, since the mere fact of being a HCW and being in contact with COVID does not provide evidence of exposure to the virus. Presumably the individuals implemented protective measures; while the immune system of some might have been exposed to the virus, to simply assume so is a serious overinterpretation.

Please see response to Reviewer 1, point 1. We agree it would be incorrect to imply that all HCW without laboratory-confirmed infection were exposed to the virus; we had used the term ‘exposed uninfected, ES’ as a shorthand to distinguish this cohort and to emphasise that they were at increased risk compared to the general population, since they were matched for demographics and exposure risk within the hospital with a seropositive cohort that had 21.5% infection rate (highly exposed relative to the general public, Treibel et al Lancet 2020). PPE was not routinely used by HCW at the start of the first wave of infections in the UK and mask wearing was not mandated until 24th July. In fact our data support only 10-20% of seronegative HCW as having had exposure leading to abortive infection. As discussed above, we have therefore tried to make it clearer that we are not assuming exposure by now referring to this cohort throughout as ‘seronegative healthcare workers (SN-HCW)’ rather than ‘exposed seronegative’. We have amended the sentence quoted above to refer to ‘risk of exposure’ rather than ‘likely exposure’.

A control cohort of non-HCW, non-exposed subjects is not presented. Data from a pre-pandemic cohort is presented, but lack of a contemporary cohort is still a major limitation, since seasonal exposure to common cold coronaviruses could change the level and patterns of reactivity.

Due to the nature of abortive infections (unidentifiable by routine PCR/serology) it is difficult to identify a truly unexposed contemporary cohort. However, using longitudinal T cell monitoring from

before PCR+ infections peaked in London and blood transcriptomic monitoring we were able to identify subsets of seronegative individuals who did or did not show evidence of exposure, and therefore have made within cohort comparisons. Using comparisons with unexposed pre-pandemic cohorts and a seropositive cohort we were also able to highlight differences in post-exposure T cell responses in a subset of seronegative HCWs and to show that raised IFI27 signal was seen selectively only in those with strong post-exposure T cell responses. We agree that the pre-pandemic cohorts likely differ from HCW on various confounders and we have therefore relied primarily on comparisons within our HCW cohort, between those with laboratory-confirmed infection, or T cell and innate signals of exposure, or neither. We have now included serology showing that all HCW had been previously exposed to seasonal coronaviruses, with no differences in titres for spike from the four endemic human coronaviruses between those with or without expanded RTC-specific T cells (new Extended Data Fig. 7 page 45).

The next section analyses RTC-specific T cell in ES. It is stated that “The infected group had memory T cells dominated by more responses to structural proteins (Spike, membrane, NP, and ORF3a) than to RTC (NSP7, NSP12, NSP13) (Fig. 2a-b). By contrast, pre-existing T cell responses predominantly targeted RTC proteins, whilst ES recognized both regions.” This interpretation is not supported by the data shown in Fig 2a-b; which by contrast shows that while ES responses are overall higher, the relative balance of responses is similar in ES and pre-pandemic samples.

We have re-phrased the interpretation of Fig. 2a-b as follows: “...By contrast, SN-HCW targeted both structural and RTC regions, with significantly more RTC-specific T cells than either the infected or pre-pandemic groups (Fig. 2a, Extended Data Fig. 2c-d). Pre-pandemic samples had a ratio of RTC to structural T cell responses that did not differ significantly from that in SN-HCW (Fig. 2b), pointing to a possible influence of pre-existing responses on the pool of T cells expanding in SN-HCW.” Results sections lines 178-185.

In the same section is stated that” T cells generated by early, transient viral exposure preferentially focus on key components of the RTC “. As mentioned above, this is overstated, since no evidence of early transient viral exposure is presented.

Thank you for pointing out this overstatement; we have changed this sentence to simply state “...T cells seen in SN-HCW preferentially focus on key components of the RTC.” Results section line 203. In the following sections we then go on to provide evidence supporting viral exposure in a subset of SN-HCW, showing in vivo expansion of RTC-specific T cells and induction of an innate transcriptional signature of SARS-CoV-2 infection.

In the following section it is stated that T cell responses are focused on RTC, and it is shown that RTC most conserved in CCC. However, it can't be concluded T cell responses target RTC (NSP7, NSP12, NSP13) because many NSP are not analyzed and some of them are just as conserved. NSP 5,8,10,14,16 have higher homology than NSP7; NSP 16 has higher homology than NSP12.

This is a good point, we have not studied T cells against other highly conserved regions in ORF1 such as NSP16 but have now added the need for future studies addressing this to the Discussion. We have amended the interpretation to clarify that NSP12 was the most commonly targeted amongst the proteins tested. Additionally, as mentioned above, we have estimated the sequence conservation of viral proteins across SARS-CoV-2 clades, showing that NSP12 and NSP13 are again among the most conserved proteins.

Furthermore, the fact that the RTC is conserved provides an alternative explanation for the data presented; e.g. responses are directed to RTC because of conservation and preexisting CCC responses, rather than abortive infection, as also shown by the data demonstrating similar levels of NSP12 recognition in pre-pandemic samples.

We agree that the conservation of RTC proteins, and their targeting by pre-existing HCoV cross-reactive responses, is a likely factor driving their expansion upon encountering SARS-CoV-2. However, the selective enrichment and expansion of such responses in a subset of SN-HCW who also have induction of an innate signature of SARS-CoV-2 infection, supports the interpretation that RTC-specific T cells also associate with, and perhaps contribute to, abortive infection.

Referee #3 (Remarks to the Author):

....Most of these conclusions have been reached previously, although the data in this manuscript are especially compelling.

....If these samples or those from seronegative HCWs also shown in Figure 4 (4c-e) could be used to demonstrate expansion of epitope-specific (as opposed to protein-specific T cells), the conclusions about expansion of pre-existing cells would be strengthened considerably.

We are pleased you find our data compelling; as described below, we are able to provide the requested data on epitope-specific expansion to strengthen our conclusions.

1. Lines 96, 399-409-As the authors point out, the 5' end of the genome is translated after cell entry to produce ORF1a/b proteins involved in transcription/replication. However, these proteins are believed to be immediately incorporated into membranous structures, initiating the production of subgenomic negative and positive strand RNA. The latter are used as the template for producing proteins, including structural proteins which are made in larger amounts than are the nonstructural proteins. It is unlikely that this small difference in time plays any role in the rapidity in which a T cell response is activated. This should be modified since it is not correct and does not change the importance of the results.

We acknowledge that SARS-CoV-2 typically has a very rapid life-cycle and, as requested, have therefore removed many of the sentences we had included stressing the potential relevance of the

earlier translation of ORF1a/b compared to structural proteins. However, it is not yet known if this rapid lifecycle applies in all target cell types and in all circumstances. RIG-I has been reported to restrain SARS-CoV-2 replication by abrogating viral RNA-dependent-RNA polymerase mediation of the first step of replication (Yamada et al Nature Immunol. 2021). We therefore think it is plausible that innate restriction of SARS-CoV-2 in selected cells might terminate infection at a stage when only RTC proteins are present in the cell for presentation to T cells, because sub-genomic replication is blocked. In this scenario, any-pre-existing T cells targeting RNA polymerase (NSP12) would be uniquely able to recognise cells with such restricted infection and help to rapidly clear them. We have speculated about this in lines 523-525 of the Conclusions.

2. Line 328-339, Figures 4a, 4b-These data are the most novel in the report and most strongly support the conclusion that pre-existing virus-specific T cells expanded in response to infection. Since the responses could also be de novo, the argument would be strengthened considerably if the prepandemic and postexposure responses were shown to be directed at the same epitopes or subpools of nsp-specific epitopes.

For the Control Cohort shown in Fig. 4a-b we now include a representative example ELISpot showing a pre-existing sub-pool (NSP12-4) response expanding after putative subclinical exposure. Below we detail further sub-pool and epitope-specific expansion data from COVIDsortium.

3. Lines 341-388, Figures 4c-e-PBMCs were available at the time of enrollment and at 16 weeks for some seronegative HCWs. In addition to presenting the composite data shown in figure 4e, paired data should be shown for individual subjects in a separate figure or Extended Data Figure, to support the conclusion that pre-existing T cells responding to specific subpools expanded in response to infection. While it may not be possible to show that single epitope-specific cells expanded because of PBMC availability, this approach could at least demonstrate concordance between the early and later responses to nsp-specific peptide pools since these data are in hand.

Paired data showing expansion for total NSP12 responses pre-post exposure are shown in Fig. 4b for the close contact cohort and Fig. 4c for the COVIDsortium HCW cohort. Because we only had 1-3 million PBMC available from baseline samples, we stimulated with a single NSP12 pool. However, we have now obtained baseline and 16wk vials from additional SN-HCW so have been able to add the requested demonstration of pre/post exposure expansion at the level of sub-pool and individual peptide (new Fig. 4e-g, Extended Data Fig. 8c-d).

4. Line 394-6-This conclusion is premature since it is equally likely that the infection is controlled by the innate or another aspect of the immune response and that T cell expansion is a secondary phenomenon (as the authors mention on lines 447-449). It should be toned down.

We have toned this conclusion down as follows: "Taken together, these data highlight a role for the expansion of pre-existing and de novo induction of RTC-specific T cells in aborting early viral infection without the development of an antibody response." Lines 469-472.

Minor comments.

1. Figure 2c—What stimulator was used in this assay?

The labels for the stimulations in Fig. 2c were omitted in error and have now been added back to this panel.

2. Line 306—HKU1 is a seasonal CoV, so this sentence should be corrected.

Thank you for pointing out this typo which we have now corrected.

Reviewer Reports on the First Revision:

Referee #1 (Remarks to the Author):

The authors have made some substantial revisions and improved the clarity of many key figures. They have also added valuable data. There are a few relatively minor points that could be clarified.

1. Lines 309–312—what data do these sentences refer to? Are we supposed to be comparing the magnitude of responses in 3d to those in 3e? If so this should be done statistically—were the experiments conducted in a way that allow us to make that comparison?

2. Figure 4f—this is described as “single SARS-CoV-2 peptide responses, but I cannot find what peptide is being used here. The main text just says “and individual peptide (Fig. 4f-g)” and the legend says “responses to individual 9-15 mer peptides from RTC”

3. Lines 496–497—“in line with the dominance of CD4+ responses in SN-HCW shown here”—I think this is referring to extended Figure 3e. This could use a bit more comment—do the authors think this is maybe just variation in the sensitivity of assays to detect CD4 vs. CD8 (in general CD4s are easy to see in functional assays)? Or do they think that the CD4s might be playing a protective role (in an apparently antibody-independent manner)?

4. Line 62, I would add “putative” before “abortive”

Referee #2 (Remarks to the Author):

I have reviewed the revised manuscript draft. The authors have improved the manuscript by softening some of the statements, clarifying some issues and including additional data. I believe that as a result the authors have addressed most of my concerns and I would therefore recommend acceptance.

Referee #3 (Remarks to the Author):

The authors did an excellent job at responding to the comments of the reviewers. This reviewer has no additional comments or questions.